# Unified Algorithms for RL with Decision-Estimation Coefficients: No-Regret, PAC, and Reward-Free Learning

## Abstract

Finding unified complexity measures and algorithms for sample-efficient learning is a central topic of research in reinforcement learning (RL). The Decision-Estimation Coefficient (DEC) is recently proposed by Foster et al. (2021) as a governing complexity measure for sample-efficient no-regret RL. This paper makes progress towards a unified theory for RL with the DEC framework. First, we propose two new DEC-type complexity measures: Explorative DEC (EDEC), and Reward-Free DEC (RFDEC). We show that they are necessary and sufficient for sample-efficient PAC learning and reward-free learning, thereby extending the original DEC which only captures no-regret learning. Next, we design new unified sample-efficient algorithms for all three learning goals. Our algorithms instantiate variants of the Estimation-To-Decisions (E2D) meta-algorithm with a strong and general model estimation subroutine. Even in the no-regret setting, our algorithm E2D-TA improves upon the algorithms of Foster et al. (2021) which require either bounding a variant of the DEC which may be prohibitively large, or designing problem-specific estimation subroutines. As applications, we recover existing and obtain new sample-efficient learning results for a wide range of tractable RL problems using essentially a single algorithm. Finally, as a connection, we re-analyze two existing optimistic model-based algorithms based on Posterior Sampling or Maximum Likelihood Estimation, showing that they enjoy similar regret bounds as E2D-TA under similar structural conditions as the DEC.

## 1 Introduction

Reinforcement Learning (RL) has achieved immense success in modern artificial intelligence. As RL agents typically require an enormous number of samples to train in practice (Mnih et al., 2015; Silver et al., 2016), *sample-efficiency* has been an important question in RL research. This question has been studied extensively in theory, with provably sample-efficient algorithms established for many concrete RL problems starting with tabular Markov Decision Processes (MDPs) (Brafman & Tennenholtz, 2002; Azar et al., 2017; Agrawal & Jia, 2017; Jin et al., 2018; Dann et al., 2019; Zhang et al., 2020b), and later MDPs with various types of linear structures (Yang & Wang, 2019; Jin et al., 2020b; Zanette et al., 2020b; Ayoub et al., 2020; Zhou et al., 2021; Wang et al., 2021).

Towards a more unifying theory, a recent line of work seeks general structural conditions and unified algorithms that encompass as many as possible known sample-efficient RL problems. Many such structural conditions have been identified, such as Bellman rank (Jiang et al., 2017), Witness rank (Sun et al., 2019), Eluder dimension (Russo & Van Roy, 2013; Wang et al., 2020b), Bilinear Class (Du et al., 2019), and Bellman-Eluder dimension (Jin et al., 2021). The recent work of Foster et al. (2021) proposes the Decision-Estimation Coefficient (DEC) as a quantitative complexity measure that governs the statistical complexity of model-based RL with a model class. Roughly speaking, the DEC measures the optimal trade-off—achieved by any policy—between exploration (gaining information) and exploitation (being a near-optimal policy itself) when the true model could be any model within the model class. Foster et al. (2021) establish regret lower bounds for online RL in terms of the DEC, and upper bounds in terms of (a variant of) the DEC and model class capacity, showing that the DEC is necessary and (in the above sense) sufficient for online RL with low regret. This constitutes a significant step towards a unified understanding of sample-efficient RL.

Despite this progress, several important questions remain open within the DEC framework. First, in Foster et al. (2021), regret upper bounds for low-DEC problems are achieved by the Estimation-To-Decisions (E2D) meta-algorithm, which requires a subroutine for online model estimation given past observations. However, their instantiations of this algorithm either (1) use a general *improper*[1] estimation subroutine that works black-box for any model class, but results in a regret bound that scales with a (potentially significantly) larger variant of the DEC that does not admit known polynomial bounds, or (2) require a *proper* estimation subroutine, which typically requires problem-specific designs and unclear how to construct for general model classes. These additional bottlenecks prevent their instantiations from being a unified sample-efficient algorithm for any low-DEC problem. Second, while the DEC captures the complexity of no-regret learning, there are alternative learning goals that are widely studied in the RL literature such as PAC learning (Dann et al., 2017) and reward-free learning (Jin et al., 2020a), and it is unclear whether they can be characterized using a similar framework. Finally, several other optimistic model-based algorithms such as Optimistic Posterior Sampling (Zhang, 2022; Agarwal & Zhang, 2022a) or Optimistic Maximum Likelihood Estimation (Mete et al., 2021; Liu et al., 2022a;b) have been proposed in recent work, whereas the E2D algorithm does not explicitly use optimism in its algorithm design. It is unclear whether E2D actually bears any similarities or connections to the aforementioned optimistic algorithms.

In this paper, we resolve the above open questions positively by developing new complexity measures and unified algorithms for RL with Decision-Estimation Coefficients. Our contributions can be summarized as follows.

- We design E2D-TA, the first unified algorithm that achieves low regret for any problem with bounded DEC and low-capacity model class (Section 3). E2D-TA instantiates the E2D meta-algorithm with *Tempered Aggregation*, a general improper online estimation subroutine that achieves stronger guarantees than variants used in existing work.
- We establish connections between E2D-TA and two existing model-based algorithms: Optimistic Model-Based Posterior Sampling, and Optimistic Maximum-Likelihood Estimation. We show that these two algorithms enjoy similar regret bounds as E2D-TA under similar structural conditions as the DEC (Appendix E).
- We extend the DEC framework to two new learning goals: PAC learning and reward-free learning. We define variants of the DEC, which we term as Explorative DEC (EDEC) and Reward-Free DEC (RFDEC), and show that they give upper and lower bounds for sample-efficient learning in the two settings respectively (Section 4).
- We instantiate our results to give sample complexity guarantees for the broad problem class of RL with low-complexity Bellman representations. Our results recover many existing and yield new guarantees when specialized to concrete RL problems (Section 5).

**Related work** Our work is closely related to the long lines of work on sample-efficient RL (both no-regret/PAC and reward-free), and problems/algorithms in general interactive decision making. We review these related work in Appendix A due to the space limit.

## 2 PRELIMINARIES

**RL as Decision Making with Structured Observations** We adopt the general framework of Decision Making with Structured Observations (DMSO) (Foster et al., 2021), which captures broad classes of problems such as bandits and reinforcement learning.

In DMSO, the environment is described by a *model* $M = (\mathsf{P}^M, R^M)$, where $\mathsf{P}^M$ specifies the distribution of the *observation* $o \in \mathcal{O}$, and $R^M$ specifies the conditional means[2] of the *reward vector* $\mathbf{r} \in [0,1]^H$, where $H$ is the horizon length. The learner interacts with a model using a *policy* $\pi \in \Pi$. Upon executing $\pi$ in $M$, they observe an (observation, reward) tuple $(o, \mathbf{r}) \sim M(\pi)$ as follows:

1. The learner first observes an observation $o \sim \mathsf{P}^M(\pi)$ (also denoted as $\mathbb{P}^{M,\pi}(\cdot) \in \Delta(\mathcal{O})$).
2. Then, the learner receives a (random) reward vector $\mathbf{r} = (r_h)_{h=1}^H$, with conditional mean $\mathbf{R}^M(o) = (R_h^M(o))_{h=1}^H := \mathbb{E}_{\mathbf{r} \sim R^M(\cdot|o)}[\mathbf{r}] \in [0,1]^H$ and independent entries conditioned on $o$.

---

[1]Which outputs in general a distribution of models within class rather than a single model.
[2]Note that $R^M$ (and thus $M$) only specifies the conditional mean rewards instead of the reward distributions.

Let $f^M(\pi) := \mathbb{E}^{M,\pi}[\sum_{h=1}^{H} r_h]$ denote the *value* (expected cumulative reward) of $\pi$ under $M$, and let $\pi_M := \arg\max_{\pi \in \Pi} f^M(\pi)$ and $f^M(\pi_M)$ denote the optimal policy and optimal value for $M$, respectively.

In this paper, we focus on RL in episodic Markov Decision Processes (MDPs) using the DMSO framework. An MDP $M = (H, \mathcal{S}, \mathcal{A}, \mathbb{P}^M, r^M)$ can be cast as a DMSO problem as follows. The observation $o = (s_1, a_1, \ldots, s_H, a_H)$ is the full state-action trajectory (so that the observation space is $\mathcal{O} = (\mathcal{S} \times \mathcal{A})^H$). Upon executing policy $\pi = \{\pi_h : \mathcal{S} \to \Delta(\mathcal{A})\}_{h \in [H]}$ in $M$, the learner observes $o = (s_1, a_1 \ldots, s_H, a_H) \sim \mathsf{P}^M(\pi)$, which sequentially samples $s_1 \sim \mathbb{P}_0^M(\cdot)$, $a_h \sim \pi_h(\cdot|s_h)$, and $s_{h+1} \sim \mathbb{P}_h^M(\cdot|s_h, a_h)$ for all $h \in [H]$. The learner then receives a reward vector $\mathbf{r} = (r_h)_{h \in [H]} \in [0, 1]^H$, where $r_h = r_h^M(s_h, a_h)$ is the (possibly random) instantaneous reward for the $h$-th step with conditional mean $\mathbb{E}^M[r_h|o] = R_h^M(o) =: R_h^M(s_h, a_h)$ depending only on $(s_h, a_h)$. We assume that $\sum_{h=1}^{H} R_h^M(s_h, a_h) \in [0, 1]$ for all $M$ and all $o \in \mathcal{O}$.

**Learning goals** We consider the online learning setting, where the learner interacts with a fixed (unknown) ground truth model $M^\star$ for $T$ episodes. Let $\pi^t \in \Pi$ denote the policy executed within the $t$-th episode. In general, $\pi^t$ may be sampled by the learner from a distribution $p^t \in \Delta(\Pi)$ before the $t$-th episode starts. One main learning goal in this paper is to minimize the standard notion of regret that measures the cumulative suboptimality of $\{\pi^t\}_{t \in [T]}$:

$$\mathbf{Reg_{DM}} := \sum_{t=1}^{T} \mathbb{E}_{\pi^t \sim p^t}\Big[ f^{M^\star}(\pi_{M^\star}) - f^{M^\star}(\pi^t)\Big].$$

To achieve low regret, this paper focuses on model-based approaches, where we are given a *model class* $\mathcal{M}$, and we assume *realizability*: $M^\star \in \mathcal{M}$. Additionally, throughout the majority of the paper, we assume that the model class is finite: $|\mathcal{M}| < \infty$ (or $|\mathcal{P}| < \infty$ for the reward-free setting in Section 4.2) for simplicity of the presentation; both can be relaxed using standard covering arguments (see e.g. Appendix D.2), which we do when we instantiate our results to concrete RL problems in Example 13-15.

## 2.1 DEC WITH RANDOMIZED REFERENCE MODELS

The Decision-Estimation Coefficient (DEC) is proposed by Foster et al. (2021) as a key quantity characterizing the statistical complexity of sequential decision making. We consider the following definition of DEC with randomized reference models (henceforth "DEC"):

**Definition 1** (DEC with randomized reference models). *The DEC of $\mathcal{M}$ with respect to distribution $\overline{\mu} \in \Delta(\mathcal{M})$ (with policy class $\Pi$ and parameter $\gamma > 0$) is defined as*

$$\mathrm{dec}_\gamma(\mathcal{M}, \overline{\mu}) := \inf_{p \in \Delta(\Pi)} \sup_{M \in \mathcal{M}} \mathbb{E}_{\pi \sim p} \mathbb{E}_{\overline{M} \sim \overline{\mu}}\big[f^M(\pi_M) - f^M(\pi) - \gamma D_{\mathrm{RL}}^2(M(\pi), \overline{M}(\pi))\big],$$

*Further define $\overline{\mathrm{dec}}_\gamma(\mathcal{M}) := \sup_{\overline{\mu} \in \Delta(\mathcal{M})} \mathrm{dec}_\gamma(\mathcal{M}, \overline{\mu})$. Above, $D_{\mathrm{RL}}^2$ is the following squared divergence function*

$$D_{\mathrm{RL}}^2(M(\pi), \overline{M}(\pi)) := D_{\mathrm{H}}^2(\mathsf{P}^M(\pi), \mathsf{P}^{\overline{M}}(\pi)) + \mathbb{E}_{o \sim \mathsf{P}^M(\pi)}\Big[\big\|\mathbf{R}^M(o) - \mathbf{R}^{\overline{M}}(o)\big\|_2^2\Big], \qquad (1)$$

*where $D_{\mathrm{H}}^2(\mathbb{P}, \mathbb{Q}) := \int(\sqrt{d\mathbb{P}/d\mu} - \sqrt{d\mathbb{Q}/d\mu})^2 d\mu$ denotes the standard Hellinger distance between probability distributions $\mathbb{P}, \mathbb{Q}$.*

Definition 1 instantiates the general definition of DECs in Foster et al. (2021, Section 4.3) with divergence function chosen as $D_{\mathrm{RL}}^2$. The quantity $\mathrm{dec}_\gamma(\mathcal{M}, \overline{\mu})$ measures the optimal trade-off of a policy distribution $p \in \Delta(\Pi)$ between two terms: low suboptimality $f^M(\pi_M) - f^M(\pi)$, and high information gain $D_{\mathrm{RL}}^2(M(\pi), \overline{M}(\pi))$ with respect to the *randomized* reference model $\overline{M} \sim \overline{\mu}$.

The main feature of $D_{\mathrm{RL}}^2$ is that it treats the estimation of observations and rewards separately: It requires the observation distribution to be estimated accurately in Hellinger distance between the *full distributions*, but the reward only accurately in the squared $L_2$ error between the *conditional means*. Such a treatment is particularly suitable for RL problems, where estimating mean rewards is easier than estimating full reward distributions[3] and is also sufficient in most scenarios.

---

[3]Foster et al. (2021) mostly use the standard Hellinger distance (in the tuple $(o, \mathbf{r})$) in their definition of the DEC, which cares about full reward distributions (cf. Appendix C.1 for detailed discussions).

---

**Algorithm 1** E2D-TA: ESTIMATION-TO-DECISIONS WITH TEMPERED AGGREGATION

---

**Input:** Parameter $\gamma > 0$; Learning rate $\eta_{\mathrm{p}} \in (0, \frac{1}{2})$, $\eta_{\mathrm{r}} > 0$.

1: Initialize $\mu^1 \leftarrow \mathrm{Unif}(\mathcal{M})$.
2: **for** $t = 1, \ldots, T$ **do**
3:     Set $p^t \leftarrow \arg\min_{p \in \Delta(\Pi)} \widehat{V}_{\gamma}^{\mu^t}(p)$, where $\widehat{V}_{\gamma}^{\mu^t}$ is defined in (2).
4:     Sample $\pi^t \sim p^t$. Execute $\pi^t$ and observe $(o^t, \mathbf{r}^t)$.
5:     Update randomized model estimator by Tempered Aggregation:

$$\mu^{t+1}(M) \propto_M \mu^t(M) \cdot \exp\left(\eta_{\mathrm{p}} \log \mathbb{P}^{M, \pi^t}(o^t) - \eta_{\mathrm{r}} \left\| \mathbf{r}^t - \mathbf{R}^M(o^t) \right\|_2^2\right). \tag{3}$$

---

## 3  E2D WITH TEMPERED AGGREGATION

We begin by presenting our algorithm *Estimation-to-Decisions with Tempered Aggregation* (E2D-TA; Algorithm 1), a unified sample-efficient algorithm for any problem with bounded DEC.

**Algorithm description**  In each episode $t$, Algorithm 1 maintains a randomized model estimator $\mu^t \in \Delta(\mathcal{M})$, and uses it to obtain a distribution of policies $p^t \in \Delta(\Pi)$ by minimizing the following risk function (cf. Line 3):

$$\widehat{V}_{\gamma}^{\mu^t}(p) := \sup_{M \in \mathcal{M}} \mathbb{E}_{\pi \sim p} \mathbb{E}_{\overline{M} \sim \mu^t}\left[ f^M(\pi_M) - f^M(\pi) - \gamma D_{\mathrm{RL}}^2(M(\pi), \overline{M}(\pi)) \right]. \tag{2}$$

This risk function instantiates the E2D meta-algorithm with randomized estimators (Foster et al., 2021, Algorithm 3) with divergence $D_{\mathrm{RL}}^2$. The algorithm then samples a policy $\pi^t \sim p^t$, executes $\pi^t$, and observes $(o^t, r^t)$ from the environment (Line 4).

Core to our algorithm is the subroutine for updating our randomized model estimator $\mu^t$: Inspired by Agarwal & Zhang (2022a), we use a *Tempered Aggregation* subroutine that performs an exponential weights update on $\mu^t(M)$ using a linear combination of the log-likelihood $\log \mathbb{P}^{M, \pi^t}(o^t)$ for the observation, and the negative squared $L_2$ loss $-\left\| \mathbf{r}^t - \mathbf{R}^M(o^t) \right\|_2^2$ for the reward (cf. Line 5). An important feature of this subroutine is the learning rate $\eta_{\mathrm{p}} \leqslant 1/2$, which is smaller compared to e.g. Vovk's aggregating algorithm (Vovk, 1995) which uses $\eta_{\mathrm{p}} = 1$. As we will see shortly, this difference is crucial and allows a stronger estimation guarantee that is suitable for our purpose. As an intuition, exponential weights with $\exp(\eta_{\mathrm{p}} \log \mathbb{P}^{M, \pi^t}(o^t)) = (\mathbb{P}^{M, \pi^t}(o^t))^{\eta_{\mathrm{p}}}$ with $\eta_{\mathrm{p}} \leqslant 1/2$ is equivalent to computing the *tempered posterior* in a Bayesian setting (Bhattacharya et al., 2019; Alquier & Ridgway, 2020) (hence our name "tempered"), whereas $\eta_{\mathrm{p}} = 1$ computes the *exact* posterior (see Appendix C.2 for a derivation).

We are now ready to present the main theoretical guarantee for Algorithm 1.

**Theorem 2** (E2D with Tempered Aggregation). *Choosing $\eta_{\mathrm{p}} = \eta_{\mathrm{r}} = 1/3$, Algorithm 1 achieves the following with probability at least $1 - \delta$:*

$$\mathbf{Reg_{DM}} \leqslant T \overline{\mathrm{dec}}_{\gamma}(\mathcal{M}) + 10\gamma \cdot \log(|\mathcal{M}|/\delta).$$

By choosing the optimal $\gamma > 0$, we get $\mathbf{Reg_{DM}} \asymp \inf_{\gamma > 0}\left\{ T \overline{\mathrm{dec}}_{\gamma}(\mathcal{M}) + \gamma \log(|\mathcal{M}|/\delta) \right\}$, which scales as $\sqrt{dT \log(|\mathcal{M}|/\delta)}$ if the model class satisfies $\overline{\mathrm{dec}}_{\gamma}(\mathcal{M}) \lesssim d/\gamma$ for some complexity measure $d$. To our best knowledge, this is the first unified sample-efficient algorithm for general problems with low DEC, and resolves a subtle but important technical challenge in Foster et al. (2021) which prevented them from obtaining such a unified algorithm.

Concretely, Foster et al. (2021, Theorem 3.3 & 4.1) show that E2D with Vovk's aggregating algorithm as the estimation subroutine achieves the following regret bound with high probability:

$$T \cdot \sup_{\overline{M} \in \mathrm{co}(\mathcal{M})} \mathrm{dec}_{\gamma}(\mathcal{M}; \delta_{\overline{M}}) + \mathcal{O}(\gamma \log(|\mathcal{M}|/\delta)),$$

where $\delta_{\overline{M}}$ denotes point mass at $\overline{M}$, and $\mathrm{co}(\mathcal{M})$ denotes the set of all possible *mixtures* of models in $\mathcal{M}$. Unfortunately, this mixture causes $\sup_{\overline{M} \in \mathrm{co}(\mathcal{M})} \mathrm{dec}_{\gamma}(\mathcal{M}; \delta_{\overline{M}})$ to be potentially intractable for

most RL problems—Even when $\mathcal{M}=\{$tabular MDPs$\}$, $\sup_{\overline{M}\in\mathrm{co}(\mathcal{M})}\mathrm{dec}_\gamma(\mathcal{M};\delta_{\overline{M}})$ does not admit known bounds of the form $\mathrm{poly}(H, S, A, 1/\gamma)$. Our Theorem 2 removes this bottleneck and only scales with $\overline{\mathrm{dec}}_\gamma(\mathcal{M})$, which is much milder and admits tractable bounds for most known tractable RL problems (Section 5); For example, $\overline{\mathrm{dec}}_\gamma(\mathcal{M}) \lesssim H^2 SA/\gamma$ for tabular MDPs (Appendix K.3.1). See Appendix C.1 for additional details on the comparison between these two DECs.

**Proof overview** The proof of Theorem 2 (deferred to Appendix D.2) builds upon the analysis of E2D meta-algorithms (Foster et al., 2021). The main new ingredient in the proof is the following online estimation guarantee for the Tempered Aggregation subroutine (proof in Appendix D.1).

**Lemma 3** (Online estimation guarantee for Tempered Aggregation). *Subroutine (3) with* $4\eta_\mathrm{p} + \eta_\mathrm{r} < 2$ *achieves the following bound with probability at least* $1 - \delta$:

$$\mathbf{Est}_\mathrm{RL} := \sum_{t=1}^{T} \mathbb{E}_{\pi^t \sim p^t} \mathbb{E}_{\widehat{M}^t \sim \mu^t} \Big[ D_\mathrm{RL}^2(M^\star(\pi^t), \widehat{M}^t(\pi^t)) \Big] \leqslant C \cdot \log(|\mathcal{M}|/\delta), \qquad (4)$$

*where $C$ depends only on* $(\eta_\mathrm{p}, \eta_\mathrm{r})$. *Specifically, we can choose* $\eta_\mathrm{p} = \eta_\mathrm{r} = 1/3$ *and* $C = 10$.

Bound (4) is stronger than the estimation bound for Vovk's aggregating algorithm (e.g. Foster et al. (2021, Lemma A.15), adapted to $D_\mathrm{RL}^2$), which only achieves

$$\sum_{t=1}^{T} \mathbb{E}_{\pi^t \sim p^t} \Big[ D_\mathrm{RL}^2\Big(M^\star(\pi^t), \mathbb{E}_{\widehat{M}^t \sim \mu^t}\big[\widehat{M}^t(\pi^t)\big]\Big) \Big] \leqslant C \cdot \log(|\mathcal{M}|/\delta), \qquad (5)$$

where $\mathbb{E}_{\widehat{M}^t \sim \mu^t}\big[\widehat{M}^t(\pi^t)\big]$ denotes the mixture model of $\widehat{M}^t(\pi^t)$ where $\widehat{M}^t \sim \mu^t$. Note that (4) is stronger than (5) by convexity of $D_\mathrm{RL}^2$ in the second argument and Jensen's inequality. Therefore, while both algorithms yield randomized model estimates, the guarantee of Tempered Aggregation is *stronger*, which in turn allows our regret bound in Theorem 2 to scale with a *smaller* DEC.

**Connections to other optimistic algorithms** In Appendix E, we re-analyze two existing optimistic algorithms: Model-based Optimistic Posterior Sampling (MOPS), and Optimistic Maximum Likelihood Estimation (OMLE). These algorithms are similar to E2D-TA due to their use of posteriors/likelihoods, and we show that they achieve regret bounds similar as E2D-TA under structural conditions similar as the DEC, thereby establishing a connection between these three algorithms.

## 4 PAC LEARNING AND REWARD-FREE LEARNING

We now extend the DEC framework to two alternative learning goals in RL beyond no-regret: PAC learning, and reward-free learning. We propose new generalized definitions of the DEC and show that they upper and lower bound the sample complexity in both settings.

### 4.1 PAC LEARNING VIA EXPLORATIVE DEC

In *PAC learning*, we only require the learner to output a near-optimal policy after $T$ episodes are finished, and does not require the executed policies $\{\pi^t\}_{t=1}^T$ (the "exploration policies") to be high-quality. It is a standard result that any no-regret algorithm can be converted into a PAC algorithm by the online-to-batch conversion (e.g. Jin et al. (2018)), so that the DEC (and the corresponding E2D-TA algorithm) gives upper bounds for PAC learning as well. However, for certain problems, there may exist PAC algorithms that are better than converted no-regret algorithms, for which the DEC would not *tightly* capture the complexity of PAC learning.

To better capture PAC learning, we define the following Explorative DEC (EDEC):

**Definition 4** (Explorative DEC). *The Explorative Decision-Estimation Coefficient (EDEC) of a model-class $\mathcal{M}$ with respect to $\overline{\mu} \in \Delta(\mathcal{M})$ and parameter $\gamma > 0$ is defined as*

$$\mathrm{edec}_\gamma(\mathcal{M}, \overline{\mu}) := \inf_{\substack{p_\mathrm{exp} \in \Delta(\mathcal{M}) \\ p_\mathrm{out} \in \Delta(\mathcal{M})}} \sup_{M \in \mathcal{M}} \mathbb{E}_{\pi \sim p_\mathrm{out}} \big[ f^M(\pi_M) - f^M(\pi) \big] - \gamma \mathbb{E}_{\pi \sim p_\mathrm{exp}, \overline{M} \sim \overline{\mu}} \big[ D_\mathrm{RL}^2(M(\pi), \overline{M}(\pi)) \big].$$

*Further, define* $\overline{\mathrm{edec}}_\gamma(\mathcal{M}) := \sup_{\overline{\mu} \in \Delta(\mathcal{M})} \mathrm{edec}_\gamma(\mathcal{M}, \overline{\mu})$.

The main difference between the EDEC and the DEC (Definition 1) is that the inf is taken over two different policy distributions $p_{\text{exp}}$ and $p_{\text{out}}$, where $p_{\text{exp}}$ (the "exploration policy distribution") appears in the information gain term, and $p_{\text{out}}$ (the "output policy distribution") appears in the suboptimality term. In comparison, the DEC restricts the policy distribution to be the same in both terms. This accurately reflects the difference between PAC learning and no-regret learning, where in PAC learning the exploration policies are not required to be the same as the final output policy.

**Algorithm and theoretical guarantee** The EDEC naturally leads to the following EXPLORATIVE E2D algorithm for PAC learning. Define risk function $\widehat{V}_{\text{pac},\gamma}^{\mu^t} : \Delta(\Pi) \times \Delta(\Pi) \to \mathbb{R}$ as

$$\widehat{V}_{\text{pac},\gamma}^{\mu^t}(p_{\text{exp}}, p_{\text{out}}) := \sup_{M \in \mathcal{M}} \mathbb{E}_{\pi \sim p_{\text{out}}} \left[ f^M(\pi_M) - f^M(\pi) \right] - \gamma \mathbb{E}_{\pi \sim p_{\text{exp}}, \widehat{M}^t \sim \mu^t} \left[ D_{\text{RL}}^2(M(\pi), \widehat{M}^t(\pi)) \right].$$

$$(6)$$

Our algorithm (full description in Algorithm 7) is similar as E2D-TA (Algorithm 1), except that in each iteration, we find $(p_{\text{exp}}^t, p_{\text{out}}^t)$ that jointly minimizes $\widehat{V}_{\text{pac},\gamma}^{\mu^t}(\cdot, \cdot)$ (Line 3), execute $\pi^t \sim p_{\text{exp}}^t$ to collect data, and return $\widehat{p}_{\text{out}} = \frac{1}{T} \sum_{t=1}^{T} p_{\text{out}}^t$ as the output policy after $T$ episodes.

**Theorem 5** (PAC learning with EXPLORATIVE E2D). *Choosing $\eta_{\text{p}} = \eta_{\text{r}} = 1/3$, Algorithm 7 achieves the following PAC guarantee with probability at least $1 - \delta$:*

$$\mathbf{SubOpt} := f^{M^\star}(\pi_{M^\star}) - \mathbb{E}_{\pi \sim \widehat{p}_{\text{out}}} \left[ f^{M^\star}(\pi) \right] \leqslant \overline{\text{edec}}_\gamma(\mathcal{M}) + 10 \frac{\gamma \log(|\mathcal{M}|/\delta)}{T}.$$

The proof can be found in Appendix H.1. For problems with $\overline{\text{edec}}_\gamma(\mathcal{M}) \lesssim \widetilde{\mathcal{O}}(d/\gamma)$, Theorem 5 achieves $\mathbf{SubOpt} \leqslant \widetilde{\mathcal{O}}\left(\sqrt{d \log |\mathcal{M}|/T}\right)$ (by tuning $\gamma$), which implies an $\widetilde{\mathcal{O}}\left(d \log |\mathcal{M}|/\varepsilon^2\right)$ sample complexity for learning an $\varepsilon$ near-optimal policy.

In the literature, PAC RL algorithms with exploration policies different from output policies have been designed for various problems, e.g. Jiang et al. (2017); Du et al. (2021); Liu et al. (2022a). These algorithms typically design their exploration policies manually (e.g. concatenating the output policy with a uniform policy over time step $h$) using prior knowledge about the problem. By contrast, EXPLORATIVE E2D does not require such knowledge and automatically *learns* the best exploration policy $p_{\text{exp}} \in \Delta(\Pi)$ by minimizing (6), thus substantially simplifying the algorithm design.

**Lower bound** We show that a suitably localized version of the EDEC gives an information-theoretic lower bound for PAC learning. The form of this lower bound is similar as the regret lower bound in terms of the (localized) DEC (Foster et al., 2021). For any model class $\mathcal{M}$ and $\overline{M} \in \mathcal{M}$, define shorthand $\text{edec}_\gamma(\mathcal{M}, \overline{M}) := \text{edec}_\gamma(\mathcal{M}, \delta_{\overline{M}})$ where $\delta_{\overline{M}}$ denotes the point mass at $\overline{M}$.

**Proposition 6** (Lower bound for PAC learning; Informal version of Proposition H.2). *For any model class $\mathcal{M}$, $T \in \mathbb{Z}_{\geqslant 1}$, and any algorithm $\mathfrak{A}$, there exists a $M^\star \in \mathcal{M}$ such that*

$$\mathbb{E}^{M^\star, \mathfrak{A}}[\mathbf{SubOpt}] \geqslant c_0 \cdot \max_{\gamma > 0} \sup_{\overline{M} \in \mathcal{M}} \text{edec}_\gamma(\mathcal{M}_{\underline{\varepsilon}_\gamma}^\infty(\overline{M}), \overline{M}),$$

*where $c_0 > 0$ is an absolute constant, and $\mathcal{M}_{\underline{\varepsilon}_\gamma}^\infty(\overline{M})$ denotes a certain localized subset of $\mathcal{M}$ around $\overline{M}$ with radius $\underline{\varepsilon}_\gamma \asymp \gamma/T$ (formal definition in (45)).*

The upper and lower bounds in Theorem 5 and Proposition 6 together show that the EDEC governs the complexity of PAC learning, similar as the DEC for no-regret learning (Foster et al., 2021).

Proposition 6 can be used to establish PAC lower bounds for concrete problems: For example, for tabular MDPs, we show in Proposition H.3 that $\sup_{\overline{M} \in \mathcal{M}} \text{edec}_\gamma(\mathcal{M}_\varepsilon^\infty(\overline{M}), \overline{M}) \gtrsim \min\{1, HSA/\gamma\}$ as long as $\varepsilon \gtrsim HSA/\gamma$, which when plugged into Proposition 6 recovers the known $\Omega(\sqrt{HSA/T})$ PAC lower bound for tabular MDPs with $\sum_h r_h \in [0, 1]$ (Domingues et al., 2021). This implies (and is slightly stronger than) the $\Omega(\sqrt{HSAT})$ regret lower bound for the same problem implied by the DEC (Foster et al., 2021, Section 5.2.4), as no-regret is at least as hard as PAC learning.

**Relationship between DEC and EDEC** As the definition of EDEC takes the infimum over a larger set than the DEC, we directly have $\overline{\text{edec}}_\gamma(\mathcal{M}, \overline{\mu}) \leqslant \overline{\text{dec}}_\gamma(\mathcal{M}, \overline{\mu})$ for any $\mathcal{M}$ and $\overline{\mu} \in \Delta(\mathcal{M})$. The following shows that the converse also holds in an approximate sense (proof in Appendix H.3).

**Proposition 7** (Relationship between DEC and EDEC). *For any $(\alpha, \gamma) \in (0,1) \times \mathbb{R}_{>0}$ and $\overline{\mu} \in \Delta(\mathcal{M})$, we have $\operatorname{dec}_\gamma(\mathcal{M}, \overline{\mu}) \leqslant \alpha + (1-\alpha)\operatorname{edec}_{\gamma\alpha/(1-\alpha)}(\mathcal{M}, \overline{\mu})$, and thus*

$$\overline{\operatorname{edec}}_\gamma(\mathcal{M}) \overset{(i)}{\leqslant} \overline{\operatorname{dec}}_\gamma(\mathcal{M}) \overset{(ii)}{\leqslant} \inf_{\alpha > 0} \{\alpha + (1-\alpha)\overline{\operatorname{edec}}_{\gamma\alpha/(1-\alpha)}(\mathcal{M})\}.$$

Bound (i) asserts that, any problem with a bounded DEC enjoys the same bound on the EDEC, on which EXPLORATIVE E2D achieves sample complexity no worse than that of E2D-TA (Theorem 5 & Theorem 2). On the other hand, the converse bound (ii) is in general a *lossy* conversion—For a class with low EDEC, the implied DEC bound yields a slightly worse rate, similar to the standard *explore-then-commit* conversion from PAC to no-regret (cf. Appendix H.3.1 for detailed discussions). Indeed, there exist problems for which the current best sample complexity through no-regret learning and bounding the DEC is $\widetilde{\mathcal{O}}(1/\varepsilon^3)$; whereas PAC learning through bounding the EDEC gives a tighter $\widetilde{\mathcal{O}}(1/\varepsilon^2)$ (cf. Proposition 12 and the discussions thereafter).

## 4.2 REWARD-FREE LEARNING VIA REWARD-FREE DEC

In *reward-free RL* (Jin et al., 2020a), the goal is to optimally explore the environment without observing reward information, so that after the exploration phase, a near-optimal policy of *any* given reward can be computed using the collected trajectory data alone without further interacting with the environment.

We define the following Reward-Free DEC (RFDEC) to capture the complexity of reward-free learning. Let $\mathcal{R}$ denote a set of *mean* reward functions, $\mathcal{P}$ denote a set of transition dynamics, and $\mathcal{M} := \mathcal{P} \times \mathcal{R}$ denote the class of all possible models specified by $M = (\mathsf{P}, R) \in \mathcal{P} \times \mathcal{R}$. We assume the true transition dynamics $\mathsf{P}^\star \in \mathcal{P}$.

**Definition 8** (Reward-Free DEC). *The Reward-Free Decision-Estimation Coefficient (RFDEC) of model class $\mathcal{M} = \mathcal{P} \times \mathcal{R}$ with respect to $\overline{\mu} \in \Delta(\mathcal{P})$ and parameter $\gamma > 0$ is defined as*

$$\operatorname{rfdec}_\gamma(\mathcal{M}, \overline{\mu}) := \inf_{p_{\exp} \in \Delta(\Pi)} \sup_{R \in \mathcal{R}} \inf_{p_{\mathrm{out}} \in \Delta(\Pi)} \sup_{\mathsf{P} \in \mathcal{P}} \Big\{ \mathbb{E}_{\pi \sim p_{\mathrm{out}}} \big[ f^{\mathsf{P}, R}(\pi_{\mathsf{P}, R}) - f^{\mathsf{P}, R}(\pi) \big]$$
$$- \gamma \mathbb{E}_{\pi \sim p_{\exp}} \mathbb{E}_{\overline{\mathsf{P}} \sim \overline{\mu}} \big[ D_{\mathrm{H}}^2(\mathsf{P}(\pi), \overline{\mathsf{P}}(\pi)) \big] \Big\}.$$

*Further, define $\overline{\operatorname{rfdec}}_\gamma(\mathcal{M}) := \sup_{\overline{\mu} \in \Delta(\mathcal{P})} \operatorname{rfdec}_\gamma(\mathcal{M}, \overline{\mu})$.*

The RFDEC can be viewed as a modification of the EDEC, where we further insert a $\sup_{R \in \mathcal{R}}$ to reflect that we care about the complexity of learning *any* reward $R \in \mathcal{R}$, and use $D_{\mathrm{H}}^2(\mathsf{P}(\pi), \overline{\mathsf{P}}(\pi))$ as the divergence to reflect that we observe the state-action trajectories $o^t$ only and not the reward.

**Algorithm and theoretical guarantee** Our algorithm REWARD-FREE E2D (full description in Algorithm 8) is an adaptation of EXPLORATIVE E2D to the reward-free setting, and works in two phases. In the exploration phase, we find $p_{\exp}^t \in \Delta(\Pi)$ minimizing the sup-risk $\sup_{R \in \mathcal{R}} \widehat{V}_{\mathrm{rf}, \gamma}^{\mu^t}(\cdot, R)$ in the $t$-th episode, where

$$\widehat{V}_{\mathrm{rf}, \gamma}^{\mu^t}(p_{\exp}, R) := \inf_{p_{\mathrm{out}}} \sup_{\mathsf{P} \in \mathcal{P}} \mathbb{E}_{\pi \sim p_{\mathrm{out}}} \big[ f^{\mathsf{P}, R}(\pi_{\mathsf{P}, R}) - f^{\mathsf{P}, R}(\pi) \big] - \gamma \mathbb{E}_{\pi \sim p_{\exp}} \mathbb{E}_{\widehat{\mathsf{P}}^t \sim \mu^t} \big[ D_{\mathrm{H}}^2(\mathsf{P}(\pi), \widehat{\mathsf{P}}^t(\pi)) \big]. \tag{7}$$

Then, in the planning phase, for any given reward $R^\star \in \mathcal{R}$, we compute $p_{\mathrm{out}}^t(R^\star)$ as the argmin of the $\inf_{p_{\mathrm{out}}}$ in $\widehat{V}_{\mathrm{rf}, \gamma}^{\mu^t}(p_{\exp}^t, R^\star)$, and output the average policy $\widehat{p}_{\mathrm{out}}(R^\star) := \frac{1}{T} \sum_{t=1}^{T} p_{\mathrm{out}}^t(R^\star)$.

**Theorem 9** (Reward-Free E2D). *Algorithm 8 achieves the following with probability at least $1 - \delta$:*

$$\mathbf{SubOpt_{rf}} := \sup_{R^\star \in \mathcal{R}} \Big\{ f^{\mathsf{P}^\star, R^\star}(\pi_{\mathsf{P}^\star, R^\star}) - \mathbb{E}_{\pi \sim \widehat{p}_{\mathrm{out}}(R^\star)} \big[ f^{\mathsf{P}^\star, R^\star}(\pi) \big] \Big\} \leqslant \overline{\operatorname{rfdec}}_\gamma(\mathcal{M}) + \gamma \frac{3 \log(|\mathcal{P}|/\delta)}{T}.$$

The proof can be found in Appendix I.2. For problems with $\overline{\operatorname{rfdec}}_\gamma(\mathcal{M}) \lesssim \widetilde{\mathcal{O}}(d/\gamma)$, by tuning $\gamma > 0$ Theorem 9 achieves $\mathbf{SubOpt_{rf}} \leqslant \varepsilon$ within $\widetilde{\mathcal{O}}(d \log |\mathcal{P}|/\varepsilon^2)$ episodes of play. The only known such general guarantee for reward-free RL is the recently proposed RFOlive algorithm of Chen et al.

(2022) which achieves sample complexity $\widetilde{\mathcal{O}}\left(\text{poly}(H) \cdot d_{\text{BE}}^2 \log(|\mathcal{F}| |\mathcal{R}|)/\varepsilon^2\right)$ in the model-free setting[4]. Theorem 9 can be seen as a generalization of this result to the model-based setting, with a more general *form* of structural condition (RFDEC). Further, our guarantee *does not further* depend on the statistical complexity (e.g. log-cardinality) of $\mathcal{R}$ once we assume bounded RFDEC.

**Lower bound**  Similar as EDEC for PAC learning, the RFEC also gives the following lower bound for reward-free learning. For any $\mathcal{M} = \mathcal{P} \times \mathcal{R}$ and $\overline{\mathsf{P}} \in \mathcal{P}$, define shorthand $\text{rfdec}_\gamma(\mathcal{M}, \overline{\mathsf{P}}) := \text{rfdec}_\gamma(\mathcal{M}, \delta_{\overline{\mathsf{P}}})$ where $\delta_{\overline{\mathsf{P}}}$ denotes the point mass at $\overline{\mathsf{P}}$.

**Proposition 10** (Reward-free lower bound; Informal version of Proposition I.2). *For any model class $\mathcal{M} = \mathcal{P} \times \mathcal{R}$, $T \in \mathbb{Z}_{\geqslant 1}$, and any algorithm $\mathfrak{A}$, there exists a $\mathsf{P}^\star \in \mathcal{P}$ such that*

$$\mathbb{E}^{\mathsf{P}^\star, \mathfrak{A}}[\mathbf{SubOpt_{rf}}] \geqslant c_0 \cdot \max_{\gamma > 0} \sup_{\overline{\mathsf{P}} \in \mathcal{P}} \text{rfdec}_\gamma(\mathcal{M}_{\underline{\varepsilon}_\gamma}^{\infty, \text{rf}}(\overline{\mathsf{P}}), \overline{\mathsf{P}}),$$

*where $c_0 > 0$ is an absolute constant, and $\mathcal{M}_{\underline{\varepsilon}_\gamma}^{\infty, \text{rf}}(\overline{M})$ denotes a certain localized subset of $\mathcal{M}$ around $\overline{M}$ with radius $\underline{\varepsilon}_\gamma \asymp \gamma/T$ (formal definition in (50)).*

## 5 Instantiation: RL with Bellman Representability

In this section, we instantiate our theories to bound the three DEC variants and give unified sample-efficient algorithms for a broad class of problems—RL with low-complexity Bellman Representations (Foster et al., 2021). Consequently, our algorithms recover existing and obtain new sample complexity results on a wide range of concrete RL problems.

**Definition 11** (Bellman Representation). *The Bellman representation of $(\mathcal{M}, \overline{M})$ is a collection of function classes $(\mathcal{G}_h^{\overline{M}} := \{g_h^{M; \overline{M}} : \mathcal{M} \to [-1, 1]\})_{h \in [H]}$ such that:*

*(a) For all $M \in \mathcal{M}$, $\left|\mathbb{E}^{\overline{M}, \pi_M}\left[Q_h^{M, \pi_M}(s_h, a_h) - r_h - V_{h+1}^{M, \pi_M}(s_{h+1})\right]\right| \leqslant \left|g_h^{M; \overline{M}}(M)\right|.$*

*(b) There exists a family of estimation policies $\{\pi_{M,h}^{\text{est}}\}_{M \in \mathcal{M}, h \in [H]}$ and a constant $L \geqslant 1$ such that for all $M, M' \in \mathcal{M}$, $\left|g_h^{M'; \overline{M}}(M)\right| \leqslant L \cdot D_{\text{RL}}\left(\overline{M}(\pi_{M,h}^{\text{est}}), M'(\pi_{M,h}^{\text{est}})\right).$*

*We say $\mathcal{M}$ satisfies Bellman representability with Bellman representation $\mathcal{G} := (\mathcal{G}_h^{\overline{M}})_{\overline{M} \in \mathcal{M}, h \in [H]}$ if $(\mathcal{M}, \overline{M})$ admits a Bellman representation $(\mathcal{G}_h^{\overline{M}})_{h \in [H]}$ for all $\overline{M} \in \mathcal{M}$.*

It is shown in Foster et al. (2021) that problems admitting a low-complexity Bellman representation $\mathcal{G}$ (e.g. linear or low Eluder dimension) include tractable subclasses such as (model-based versions of) Bilinear classes (Du et al., 2021) and Bellman-Eluder dimension (Jin et al., 2021). We show that Bellman representability with a low complexity $\mathcal{G}$ implies bounded DEC/EDEC/RFDEC, which in turn leads to concrete rates using our E2D algorithms in Section 3 and 4.

**Proposition 12** (Rates for RL with low-complexity Bellman representations). *Suppose $\mathcal{M}$ admits a Bellman representation $\mathcal{G}$ with low complexity: $\min\{\mathfrak{e}(\mathcal{G}_h^{\overline{M}}, \Delta), \mathfrak{s}(\mathcal{G}_h^{\overline{M}}, \Delta)^2\} \leqslant \widetilde{\mathcal{O}}(d)$ for all $(h, \overline{M}) \in [H] \times \mathcal{M}$, where $\mathfrak{e}$ is the Eluder dimension, $\mathfrak{s}$ is the star number (Definition K.3 & K.4), $d > 0$ and $\widetilde{\mathcal{O}}(\cdot)$ contains possibly $\text{polylog}(1/\Delta)$ factors. Then we have*

*(1) (**No-regret learning**) If $\pi_{M,h}^{\text{est}} = \pi_M$ for all $M \in \mathcal{M}$ (the on-policy case), then $\overline{\text{dec}}_\gamma(\mathcal{M}) \leqslant \widetilde{\mathcal{O}}\left(dH^2 L^2/\gamma\right)$, and Algorithm E2D-TA achieves $\mathbf{Reg_{DM}} \leqslant \widetilde{\mathcal{O}}(HL\sqrt{dT \log |\mathcal{M}|})$.*

*(2) (**PAC learning**) For any general $\{\pi_{M,h}^{\text{est}}\}_{M \in \mathcal{M}, h \in [H]}$, we have $\overline{\text{edec}}_\gamma(\mathcal{M}) \leqslant \widetilde{\mathcal{O}}\left(dH^2 L^2/\gamma\right)$, and Algorithm EXPLORATIVE E2D achieves $\mathbf{SubOpt} \leqslant \varepsilon$ within $\widetilde{\mathcal{O}}\left(dH^2 L^2 \log |\mathcal{M}|/\varepsilon^2\right)$ episodes of play.*

*(3) (**Reward-free learning**) If $\mathcal{G}$ is a Bellman representation in a stronger sense (cf. Definition K.8), then $\overline{\text{rfdec}}_\gamma(\mathcal{M}) \leqslant \widetilde{\mathcal{O}}\left(dH^2 L^2/\gamma\right)$, and Algorithm REWARD-FREE E2D achieves $\mathbf{SubOpt_{rf}} \leqslant \varepsilon$ within $\widetilde{\mathcal{O}}\left(dH^2 L^2 \log |\mathcal{P}|/\varepsilon^2\right)$ episodes of play.*

---

[4]Where $\mathcal{F}$ denotes the value class, and $\mathcal{R}$ denotes the reward class, and $d_{\text{BE}}$ denotes the Bellman-Eluder dimension of a certain class of *reward-free Bellman errors* induced by $\mathcal{F}$.

The proof can be found in Appendix K.2. To our best knowledge, results (2)(3) above are the first such results with DEC-type structural assumptions. In particular, when the estimation policies are general, the $\widetilde{\mathcal{O}}\left(1/\varepsilon^2\right)$ PAC sample complexity in (2) through bounded EDEC improves over Foster et al. (2021, Theorem 7.1 & F.2) which uses the conversion to DEC and results in an $\widetilde{\mathcal{O}}\left(1/\varepsilon^3\right)$ sample complexity. The no-regret result in (1) also improves over the result of Foster et al. (2021) as a unified algorithm without requiring proper online estimation oracles which is often problem-specific. We remark that a similar bound as (1) also holds for $\mathrm{psc}_\gamma$ (c.f. Definition E.1), and for $\mathrm{mlec}_\gamma$ (c.f. Definition E.4) assuming bounded Eluder dimension $\mathfrak{e}(\mathcal{G}_h^{\overline{M}}, \Delta) \leqslant \widetilde{\mathcal{O}}(d_\mathfrak{e})$, and thus MOPS and OMLE algorithms achieve similar regret bounds as E2D-TA (cf. Appendix K.1).

**Examples** Proposition 12 can be specialized to a wide range of concrete RL problems, for which we give a few illustrating examples here (problem definitions and proofs in Appendix K.3). We emphasize that, except for feeding the different model class $\mathcal{M}$'s into the Tempered Aggregation subroutines, the rates below (for each learning goal) are obtained through a single unified algorithm without further problem-dependent designs.

**Example 13.** For tabular MDPs, E2D-TA achieves regret $\mathbf{Reg_{DM}} \leqslant \widetilde{\mathcal{O}}\left(\sqrt{S^3 A^2 H^3 T}\right)$, and REWARD-FREE E2D achieves reward-free guarantee $\mathbf{SubOpt_{rf}} \leqslant \widetilde{\mathcal{O}}\left(\sqrt{S^3 A^2 H^3 / T}\right)$. ◊

Both rates are worse then the optimal $\widetilde{\mathcal{O}}(\sqrt{HSAT})$ regret bound (Azar et al., 2017)[5] and $\widetilde{\mathcal{O}}(\sqrt{\mathrm{poly}(H)S^2 A/T})$ reward-free bound (Jin et al., 2020a). However, our rates are obtained through unified algorithms that is completely agnostic to the tabular structure.

**Example 14.** For linear mixture MDPs (Ayoub et al., 2020) with a $d$-dimensional feature map, E2D-TA achieves regret $\mathbf{Reg_{DM}} \leqslant \widetilde{\mathcal{O}}\left(\sqrt{d^2 H^3 T}\right)$, and REWARD-FREE E2D achieves reward-free guarantee $\mathbf{SubOpt_{rf}} \leqslant \widetilde{\mathcal{O}}\left(\sqrt{d^2 H^3 / T}\right)$. ◊

The later result implies a sample complexity of $\widetilde{\mathcal{O}}\left(d^2 H^3 / \varepsilon^2\right)$ for $\varepsilon$-near optimal reward-free learning in linear mixture MDPs, which only has an additional $H^2$ factor over the current best sample complexity $\widetilde{\mathcal{O}}\left(d^2 H / \varepsilon^2\right)$ of Chen et al. (2021)[5].

**Example 15.** For low-rank MDPs with unknown $d$-dimensional features $(\phi, \psi) \in \Phi \times \Psi$ (i.e. the FLAMBE setting (Agarwal et al., 2020)), $A$ actions, and rewards $R \in \mathcal{R}$, EXPLORATIVE E2D achieves PAC guarantee $\mathbf{SubOpt} \leqslant \widetilde{\mathcal{O}}\left(\sqrt{dAH^2 \log(|\Phi| |\Psi| |\mathcal{R}|)/T}\right)$, and E2D-RF achieves reward-free guarantee $\mathbf{SubOpt_{rf}} \leqslant \widetilde{\mathcal{O}}\left(\sqrt{dAH^2 \log(|\Phi| |\Psi|)/T}\right)$. ◊

Our PAC result matches the best known sample complexity achieved by e.g. the V-Type Golf Algorithm of Jin et al. (2021). For reward-free learning, our linear in $d$ dependence improves over the current best $d^2$ dependence achieved by the RFOlive algorithm (Chen et al., 2022), and we do not require linearity or low complexity assumptions on the class of reward functions $\mathcal{R}$ made in existing work (Wang et al., 2020a; Chen et al., 2022). However, we remark that they handle a slightly more general setting where only the $\Phi$ class is known, due to their model-free approach.

An important subclass of low-rank MDPs is Block MDPs (Du et al., 2019). In (stationary-transition) Block MDPs with $O$ states, $A$ actions, and $S$ latent states, REWARD-FREE E2D achieves $\mathbf{SubOpt_{rf}} \leqslant \widetilde{\mathcal{O}}\left(\sqrt{(SOA + S^3 A^2)H^2/T}\right)$, which improves over the result of Jedra et al. (2022) in the $S$ dependence, and does not require a certain reachability assumption made in their result.

## 6 CONCLUSION

This paper proposes unified sample-efficient algorithms for no-regret, PAC, and reward-free reinforcement learning, by developing new complexity measures and stronger algorithms within the DEC framework. We believe our work opens up many important questions, such as developing model-free analogs of this framework, extending to other learning goals (such as multi-agent RL), and computational efficiency of our algorithms.

---

[5]Rescaled to total reward within $[0, 1]$.

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

## A  RELATED WORK

**Sample-efficient reinforcement learning**  Sample-efficient RL has been extensively studied in the basic model of tabular MDPs (Kearns & Singh, 2002; Brafman & Tennenholtz, 2002; Jaksch et al., 2010; Dann & Brunskill, 2015; Azar et al., 2017; Agrawal & Jia, 2017; Jin et al., 2018; Russo, 2019; Dann et al., 2019; Zanette & Brunskill, 2019; Zhang et al., 2020b; Domingues et al., 2021). The minimax sample complexity for finite-horizon tabular MDPs has been achieved by both model-based and model-free approaches (Azar et al., 2017; Zhang et al., 2020b). When there is function approximation, the sample-efficiency of RL has been studied under concrete assumptions about the function class and/or the MDP, such as (various forms of) linear or low-rank MDPs (Yang & Wang, 2019; Du et al., 2020; Jin et al., 2020b; Zanette et al., 2020b; Cai et al., 2020; Lattimore et al., 2020; Agarwal et al., 2020; Ayoub et al., 2020; Modi et al., 2020; Zhou et al., 2021), generalized linear

function approximation (Wang et al., 2021), Block MDPs (Du et al., 2019; Misra et al., 2020), and so on. More general structural conditions and algorithms has been studied (Russo & Van Roy, 2013; Jiang et al., 2017; Sun et al., 2019; Wang et al., 2020b) and later unified by frameworks such as Bilinear Class (Du et al., 2021) and Bellman-Eluder dimension (Jin et al., 2021).

Foster et al. (2021) propose the DEC as a complexity measure for interactive decision making problems, and develop the E2D meta-algorithm as a general model-based algorithm for problems within their DMSO framework (which covers bandits and RL). The DEC framework is further generalized in (Foster et al., 2022) to capture adversarial decision making problems. The DEC has close connections to the modulus of continuity (Donoho & Liu, 1987; 1991a;b), information ratio (Russo & Van Roy, 2016; 2018; Lattimore & Gyorgy, 2021), and Exploration-by-optimization (Lattimore & Szepesvári, 2020). Our work builds on and extends the DEC framework: We propose the E2D-TA algorithm as a general and strong instantiation of the E2D meta-algorithm, and generalize the DEC to capture PAC and reward-free learning.

**Other general algorithms**  Posterior sampling (or Thompson Sampling) is another general purpose algorithm for interactive decision making (Thompson, 1933; Russo, 2019; Agrawal & Jia, 2017; Zanette et al., 2020a; Zhang, 2022; Agarwal & Zhang, 2022a;b). Frequentist regret bounds for posterior sampling are established in tabular MDPs (Agrawal & Jia, 2017; Russo, 2019) and linear MDPs (Russo, 2019; Zanette et al., 2020a). Zhang (2022) proves regret bounds of a posterior sampling algorithm for RL with general function approximation, which is then generalized in Agarwal & Zhang (2022a;b). Our Appendix E.1 discusses the connection between the MOPS algorithm of Agarwal & Zhang (2022a) and E2D-TA. The OMLE (Optimistic Maximum Likelihood Estimation) algorithm is studied in (Liu et al., 2022a;b) for Partially Observable Markov Decision Process; however, the algorithm itself is general and can be used for any problem within the DMSO framework; We provide such a generalization and discuss the connections in Appendix E.2. Maximum-likelihood based algorithms for RL are also studied in (Mete et al., 2021; Agarwal et al., 2020; Uehara et al., 2021).

**Reward-free RL**  The reward-free learning framework is proposed by (Jin et al., 2020a) and well-studied in both tabular and function approximation settings (Jin et al., 2020a; Zhang et al., 2020a; Kaufmann et al., 2021; Ménard et al., 2021; Wang et al., 2020a; Zanette et al., 2020c; Agarwal et al., 2020; Liu et al., 2021; Modi et al., 2021; Zhang et al., 2021a;b; Qiu et al., 2021; Wagenmaker et al., 2022). The recent work of Chen et al. (2022) proposes a general algorithm for problems with low (reward-free version of) Bellman-Eluder dimension. Our Reward-Free DEC framework generalizes many of these results by offering a unified structural condition and algorithm for reward-free RL with a model class.

**Other problems covered by DMSO**  Besides multi-armed bandits and RL, the DMSO framework of (Foster et al., 2021) (and thus all our theories as well) can handle other problems such as contextual bandits (Auer et al., 2002; Langford & Zhang, 2007; Chu et al., 2011; Beygelzimer et al., 2011; Agarwal et al., 2014; Foster & Rakhlin, 2020; Foster et al., 2020), contextual reinforcement learning (Abbasi-Yadkori & Neu, 2014; Modi et al., 2018; Dann et al., 2019; Modi & Tewari, 2020), online convex bandits (Kleinberg, 2004; Bubeck et al., 2015; Bubeck & Eldan, 2016; Lattimore, 2020), and non-parametric bandits (Kleinberg, 2004; Auer et al., 2007; Kleinberg et al., 2013). Instantiating our theories to these settings would be an interesting direction for future work.

## B  TECHNICAL TOOLS

### B.1  STRONG DUALITY

The following strong duality result for variational forms of bilinear functions is standard, e.g. extracted from the proof of Foster et al. (2021, Proposition 4.2).

**Theorem B.1** (Strong duality). *Suppose that $\mathcal{X}$, $\mathcal{Y}$ are two topological spaces, such that $\mathcal{X}$ is Hausdorff[6] and $\mathcal{Y}$ is finite (with discrete topology). Then for a bi-continuous function $f : \mathcal{X} \times \mathcal{Y} \to \mathbb{R}$*

---

[6]The Hausdorff space requirement of $\mathcal{X}$ is only needed to ensure that $\Delta(\mathcal{X})$ contains all finitely supported distributions on $\mathcal{X}$.

*that is uniformly bounded, it holds that*

$$\sup_{X \in \Delta(\mathcal{X})} \inf_{Y \in \Delta(\mathcal{Y})} \mathbb{E}_{x \sim X} \mathbb{E}_{y \sim Y}[f(x,y)] = \inf_{Y \in \Delta(\mathcal{Y})} \sup_{X \in \Delta(\mathcal{X})} \mathbb{E}_{x \sim X} \mathbb{E}_{y \sim Y}[f(x,y)].$$

In this paper, for most applications of Theorem B.1, we take $\mathcal{X} = \mathcal{M}$ and $\mathcal{Y} = \Pi$. We will assume that $\Pi$ is finite, which is a natural assumption. For example, in tabular MDPs, it is enough to consider deterministic Markov policies and there are only finitely many of them. Also, the finiteness assumption in Theorem B.1 can be relaxed—The strong duality holds as long as both $\mathcal{X}, \mathcal{Y}$ is Hausdorff, and the function class $\{f(x, \cdot) : \mathcal{Y} \to \mathbb{R}\}_{x \in \mathcal{X}}$ has a finite $\rho$-covering for all $\rho > 0$. Such relaxed assumption is always satisfied in our applications.

## B.2 CONCENTRATION INEQUALITIES

We will use the following standard concentration inequality in the paper.

**Lemma B.2** (Foster et al. (2021, Lemma A.4)). *For any sequence of real-valued random variables* $(X_t)_{t \leqslant T}$ *adapted to a filtration* $(\mathcal{F}_t)_{t \leqslant T}$, *it holds that with probability at least* $1 - \delta$, *for all* $t \leqslant T$,

$$\sum_{s=1}^{t} -\log \mathbb{E}\left[\exp(-X_s) \mid \mathcal{F}_{s-1}\right] \leqslant \sum_{s=1}^{t} X_s + \log\left(\delta^{-1}\right).$$

## B.3 PROPERTIES OF THE HELLINGER DISTANCE

Recall that for two distributions $\mathbb{P}, \mathbb{Q}$ that are absolutely continuous with respect to $\mu$, their squared Hellinger distance is defined as

$$D_{\mathrm{H}}^2(\mathbb{P}, \mathbb{Q}) := \int (\sqrt{d\mathbb{P}/d\mu} - \sqrt{d\mathbb{Q}/d\mu})^2 d\mu.$$

We will use the following properties of the Hellinger distance.

**Lemma B.3** (Foster et al. (2021, Lemma A.11, A.12)). *For distributions* $\mathbb{P}, \mathbb{Q}$ *defined on* $\mathcal{X}$ *and function* $h : \mathcal{X} \to [0, R]$, *we have*

$$|\mathbb{E}_{\mathbb{P}}[h(X)] - \mathbb{E}_{\mathbb{Q}}[h(X)]| \leqslant \sqrt{2R(\mathbb{E}_{\mathbb{P}}[h(X)] + \mathbb{E}_{\mathbb{Q}}[h(X)]) \cdot D_{\mathrm{H}}^2(\mathbb{P}, \mathbb{Q})}.$$

*Therefore,* $\mathbb{E}_{\mathbb{P}}[h(X)] \leqslant 3\mathbb{E}_{\mathbb{Q}}[h(X)] + 2RD_{\mathrm{H}}^2(\mathbb{P}, \mathbb{Q})$. *Also, for function* $h : \mathcal{X} \to [-R, R]$, *we have*

$$|\mathbb{E}_{\mathbb{P}}[h(X)] - \mathbb{E}_{\mathbb{Q}}[h(X)]| \leqslant \sqrt{8R(\mathbb{E}_{\mathbb{P}}[|h(X)|] + \mathbb{E}_{\mathbb{Q}}[|h(X)|]) \cdot D_{\mathrm{H}}^2(\mathbb{P}, \mathbb{Q})}.$$

**Lemma B.4.** *For any pair of random variable* $(X, Y)$, *it holds that*

$$\mathbb{E}_{X \sim \mathbb{P}_X}\left[D_{\mathrm{H}}^2\left(\mathbb{P}_{Y|X}, \mathbb{Q}_{Y|X}\right)\right] \leqslant 2D_{\mathrm{H}}^2\left(\mathbb{P}_{X,Y}, \mathbb{Q}_{X,Y}\right).$$

*Conversely, it holds that*

$$D_{\mathrm{H}}^2\left(\mathbb{P}_{X,Y}, \mathbb{Q}_{X,Y}\right) \leqslant 3D_{\mathrm{H}}^2\left(\mathbb{P}_X, \mathbb{Q}_X\right) + 2\mathbb{E}_{X \sim \mathbb{P}_X}\left[D_{\mathrm{H}}^2\left(\mathbb{P}_{Y|X}, \mathbb{Q}_{Y|X}\right)\right].$$

*Proof.* Throughout the proof, we slightly abuse notations and write a distribution $\mathbb{P}$ and its density $d\mathbb{P}/d\mu$ interchangeably. By the definition of the Hellinger distance, we have

$$\frac{1}{2}D_{\mathrm{H}}^2\left(\mathbb{P}_{X,Y}, \mathbb{Q}_{X,Y}\right) = 1 - \int \sqrt{\mathbb{P}_{X,Y}}\sqrt{\mathbb{Q}_{X,Y}}$$

$$= 1 - \int \sqrt{\mathbb{P}_X \mathbb{Q}_X}\sqrt{\mathbb{P}_{Y|X}}\sqrt{\mathbb{Q}_{Y|X}}$$

$$\geqslant 1 - \int \frac{\mathbb{P}_X + \mathbb{Q}_X}{2}\sqrt{\mathbb{P}_{Y|X}}\sqrt{\mathbb{Q}_{Y|X}}$$

$$= \int \frac{\mathbb{P}_X + \mathbb{Q}_X}{2}\left(1 - \sqrt{\mathbb{P}_{Y|X}}\sqrt{\mathbb{Q}_{Y|X}}\right)$$

$$= \frac{1}{4}\mathbb{E}_{X \sim \mathbb{P}_X}\left[D_{\mathrm{H}}^2\left(\mathbb{P}_{Y|X}, \mathbb{Q}_{Y|X}\right)\right] + \frac{1}{4}\mathbb{E}_{X \sim \mathbb{Q}_X}\left[D_{\mathrm{H}}^2\left(\mathbb{P}_{Y|X}, \mathbb{Q}_{Y|X}\right)\right].$$

Similarly,

$$\frac{1}{2} D_{\mathrm{H}}^2 \left(\mathbb{P}_{X,Y}, \mathbb{Q}_{X,Y}\right) = 1 - \int \sqrt{\mathbb{P}_X \mathbb{Q}_X} + \int \sqrt{\mathbb{P}_X \mathbb{Q}_X}(1 - \sqrt{\mathbb{P}_{Y|X} \mathbb{Q}_{Y|X}})$$
$$\leqslant \frac{1}{2} D_{\mathrm{H}}^2 \left(\mathbb{P}_X, \mathbb{Q}_X\right) + \int \frac{\mathbb{P}_X + \mathbb{Q}_X}{2} \cdot \frac{1}{2} D_{\mathrm{H}}^2 \left(\mathbb{P}_{Y|X}, \mathbb{Q}_{Y|X}\right),$$

and hence

$$D_{\mathrm{H}}^2 \left(\mathbb{P}_{X,Y}, \mathbb{Q}_{X,Y}\right) \leqslant D_{\mathrm{H}}^2 \left(\mathbb{P}_X, \mathbb{Q}_X\right) + \frac{1}{2} \mathbb{E}_{X \sim \mathbb{P}_X} \left[ D_{\mathrm{H}}^2 \left(\mathbb{P}_{Y|X}, \mathbb{Q}_{Y|X}\right) \right]$$
$$+ \frac{1}{2} \mathbb{E}_{X \sim \mathbb{Q}_X} \left[ D_{\mathrm{H}}^2 \left(\mathbb{P}_{Y|X}, \mathbb{Q}_{Y|X}\right) \right]$$
$$\leqslant 3 D_{\mathrm{H}}^2 \left(\mathbb{P}_X, \mathbb{Q}_X\right) + 2 \mathbb{E}_{X \sim \mathbb{P}_X} \left[ D_{\mathrm{H}}^2 \left(\mathbb{P}_{Y|X}, \mathbb{Q}_{Y|X}\right) \right],$$

where the last inequality is due to Lemma B.3 and $D_{\mathrm{H}}^2 \in [0, 2]$. □

Next, recall the divergence $D_{\mathrm{RL}}^2$ defined in (1):

$$D_{\mathrm{RL}}^2(M(\pi), \overline{M}(\pi)) = D_{\mathrm{H}}^2(\mathsf{P}^M(\pi), \mathsf{P}^{\overline{M}}(\pi)) + \mathbb{E}_{o \sim \mathsf{P}^M(\pi)} \left[ \left\| \mathbf{R}^M(o) - \mathbf{R}^{\overline{M}}(o) \right\|_2^2 \right].$$

**Proposition B.5.** *Recall that* $(o, \mathbf{r}) \sim M(\pi)$ *is the observation and reward vectors as described in Section 2, with* $o \sim \mathsf{P}^M(\pi)$ *and* $\mathbf{r} \sim \mathsf{R}^M(\cdot|o)$. *Suppose that* $\mathbf{r} \in [0, 1]^H$ *almost surely and* $\|\mathbf{R}^M(o) - \mathbf{R}^{\overline{M}}(o)\|_2^2 \leqslant 2$ *for all* $o \in \mathcal{O}$. *Then it holds that*

$$D_{\mathrm{RL}}^2(M(\pi), \overline{M}(\pi)) \leqslant 5 D_{\mathrm{H}}^2 \left(M(\pi), \overline{M}(\pi)\right),$$

*where* $D_{\mathrm{H}}^2 \left(M(\pi), \overline{M}(\pi)\right)$ *is the standard squared Hellinger distance between* $\mathsf{R}^M \otimes \mathsf{P}^M(\pi)$ *and* $\mathsf{R}^{\overline{M}} \otimes \mathsf{P}^{\overline{M}}(\pi)$.

*Proof.* To prove this proposition, we need to bound $\|\mathbf{R}^M(o) - \mathbf{R}^{\overline{M}}(o)\|_2^2$ in terms of $D_{\mathrm{H}}^2 \left(\mathsf{R}^M(o), \mathsf{R}^{\overline{M}}(o)\right)$. We denote by $\mathsf{R}_h^M(o)$ the distribution of $r_h$. Then by independence, we have

$$1 - \frac{1}{2} D_{\mathrm{H}}^2 \left(\mathsf{R}^M(o), \mathsf{R}^{\overline{M}}(o)\right) = \prod_h \left(1 - \frac{1}{2} D_{\mathrm{H}}^2 \left(\mathsf{R}_h^M(o), \mathsf{R}_h^{\overline{M}}(o)\right)\right)$$
$$\leqslant \prod_h \left(1 - \frac{1}{2} D_{\mathrm{TV}}^2 \left(\mathsf{R}_h^M(o), \mathsf{R}_h^{\overline{M}}(o)\right)\right)$$
$$\leqslant \prod_h \left(1 - \frac{1}{2} \left| R_h^M(o) - R_h^{\overline{M}}(o) \right|^2\right)$$
$$\leqslant \exp \left(-\frac{1}{2} \left\| \mathbf{R}^M(o) - \mathbf{R}^{\overline{M}}(o) \right\|_2^2\right)$$
$$\leqslant 1 - \frac{1}{4} \left\| \mathbf{R}^M(o) - \mathbf{R}^{\overline{M}}(o) \right\|_2^2,$$

where the last inequality use the fact that $e^{-x} \leqslant 1 - x/2$ for all $x \in [0, 1]$. Then by Lemma B.4,

$$\mathbb{E}_{o \sim \mathsf{P}^M(\pi)} \left[ \left\| \mathbf{R}^M(o) - \mathbf{R}^{\overline{M}}(o) \right\|_2^2 \right] \leqslant 2 \mathbb{E}_{o \sim \mathsf{P}^M(\pi)} \left[ D_{\mathrm{H}}(\mathsf{R}^M(o), \mathsf{R}^{\overline{M}}(o))^2 \right]$$
$$\leqslant 4 D_{\mathrm{H}}^2 \left(M(\pi), \overline{M}(\pi)\right).$$

Combining the above estimation with the fact that $D_{\mathrm{H}}^2(\mathsf{P}^M(\pi), \mathsf{P}^{\overline{M}}(\pi)) \leqslant D_{\mathrm{H}}^2 \left(M(\pi), \overline{M}(\pi)\right)$ (data-processing inequality) completes the proof. □

The following lemma shows that, although $D_{\mathrm{RL}}^2$ is not symmetric with respect to its two arguments (due to the expectation over $o \sim \mathsf{P}^M(\pi)$ in the second term), it is *almost* symmetric within a constant multiplicative factor:

**Lemma B.6.** *For any two models $M, \overline{M}$ and any policy $\pi$, we have*
$$D_{\mathrm{RL}}^2(\overline{M}(\pi), M(\pi)) \leqslant 5 D_{\mathrm{RL}}^2(M(\pi), \overline{M}(\pi)).$$

*Proof.* For any function $h : \mathcal{O} \to [0, 2]$, by Lemma B.3 we have
$$\mathbb{E}_{o \sim \mathsf{P}^{\overline{M}}(\pi)}[h(o)] \leqslant 3 \mathbb{E}_{o \sim \mathsf{P}^M(\pi)}[h(o)] + 4 D_{\mathrm{H}}^2(\mathsf{P}^M(\pi), \mathsf{P}^{\overline{M}}(\pi)).$$

Therefore, we can take $h$ as $h(o) := \left\| \mathbf{R}^M(o) - \mathbf{R}^{\overline{M}}(o) \right\|_2^2$, and the bound above gives
$$\underbrace{D_{\mathrm{H}}^2(\mathsf{P}^M(\pi), \mathsf{P}^{\overline{M}}(\pi)) + \mathbb{E}_{o \sim \mathsf{P}^{\overline{M}}(\pi)}[h(o)]}_{D_{\mathrm{RL}}^2(\overline{M}(\pi), M(\pi))} \leqslant \underbrace{5 D_{\mathrm{H}}^2(\mathsf{P}^M(\pi), \mathsf{P}^{\overline{M}}(\pi)) + 5 \mathbb{E}_{o \sim \mathsf{P}^M(\pi)}[h(o)]}_{=5 D_{\mathrm{RL}}^2(M(\pi), \overline{M}(\pi))},$$

which is the desired result. $\qquad\square$

**Lemma B.7.** *For any two models $M, \overline{M}$ and any policy $\pi$, we have*
$$\left| f^M(\pi) - f^{\overline{M}}(\pi) \right| \leqslant \sqrt{H+1} \cdot D_{\mathrm{RL}}(M(\pi), \overline{M}(\pi)).$$

*Proof.* We have
$$
\begin{aligned}
\left| f^M(\pi) - f^{\overline{M}}(\pi) \right| &= \left| \mathbb{E}_{o \sim \mathsf{P}^M(\pi)}\left[ R^M(o) \right] - \mathbb{E}_{o \sim \mathsf{P}^{\overline{M}}(\pi)}\left[ R^{\overline{M}}(o) \right] \right| \\
&\leqslant \left| \mathbb{E}_{o \sim \mathsf{P}^M(\pi)}\left[ R^M(o) - R^{\overline{M}}(o) \right] \right| + \left| \mathbb{E}_{o \sim \mathsf{P}^M(\pi)}\left[ R^{\overline{M}}(o) \right] - \mathbb{E}_{o \sim \mathsf{P}^{\overline{M}}(\pi)}\left[ R^{\overline{M}}(o) \right] \right| \\
&\stackrel{(i)}{\leqslant} \mathbb{E}_{o \sim \mathsf{P}^M(\pi)}\left[ \sqrt{H} \left\| \mathbf{R}^M(o) - \mathbf{R}^{\overline{M}}(o) \right\|_2 \right] + D_{\mathrm{H}}(\mathsf{P}^M(\pi), \mathsf{P}^{\overline{M}}(\pi)) \\
&\stackrel{(ii)}{\leqslant} \sqrt{(H+1)\left( \mathbb{E}_{o \sim \mathsf{P}^M(\pi)}\left[ \left\| \mathbf{R}^M(o) - \mathbf{R}^{\overline{M}}(o) \right\|_2^2 \right] + D_{\mathrm{H}}^2(\mathsf{P}^M(\pi), \mathsf{P}^{\overline{M}}(\pi)) \right)} \\
&= \sqrt{H+1} \cdot D_{\mathrm{RL}}(M(\pi), \overline{M}(\pi)).
\end{aligned}
$$

Above, (i) uses the fact that $R^{\overline{M}}(o) \in [0, 1]$ almost surely, and the bound
$$\left| \mathbb{E}_{o \sim \mathsf{P}^M(\pi)}\left[ R^{\overline{M}}(o) \right] - \mathbb{E}_{o \sim \mathsf{P}^{\overline{M}}(\pi)}\left[ R^{\overline{M}}(o) \right] \right| \leqslant D_{\mathrm{TV}}(\mathsf{P}^M(\pi), \mathsf{P}^{\overline{M}}(\pi)) \leqslant D_{\mathrm{H}}(\mathsf{P}^M(\pi), \mathsf{P}^{\overline{M}}(\pi));$$
(ii) uses the Cauchy inequality $\sqrt{H}a + b \leqslant \sqrt{(H+1)(a^2 + b^2)}$ and the fact that the squared mean is upper bounded by the second moment. $\qquad\square$

## C  DISCUSSIONS ABOUT DEC DEFINITIONS AND AGGREGATION ALGORITHMS

### C.1  DEC DEFINITIONS

Here we discuss the differences between the DEC definitions used within our E2D-TA and within the E2D algorithm of Foster et al. (2021, Section 4.1) which uses Vovk's aggregating algorithm as the subroutine (henceforth E2D-VA). Recall that the regret bound of E2D-TA scales with $\overline{\mathrm{dec}}_\gamma(\mathcal{M})$ defined in Definition 1 (cf. Theorem 2).

We first remark that all the following DECs considered in Foster et al. (2021) are defined in terms of the squared Hellinger distance $D_{\mathrm{H}}^2(M(\pi), \overline{M}(\pi))$ between the full distribution of $(o, \mathbf{r})$ induced by models $M$ and $\overline{M}$ under $\pi$, instead of our $D_{\mathrm{RL}}^2$ which is defined in terms of squared Hellinger distance in $o$ and squared $L_2$ loss in (the mean of) $\mathbf{r}$. However, all these results hold for $D_{\mathrm{RL}}^2$ as well with the DEC definition and algorithms changed correspondingly. For clarity, we state their results in terms of $D_{\mathrm{RL}}^2$, which will not affect the essence of the comparisons.

Foster et al. (2021, Theorem 3.3 & 4.1) show that E2D-VA achieves the following regret bound with probability at least $1 - \delta$:
$$\mathbf{Reg_{DM}} \lesssim \mathcal{O}\left( T \cdot \sup_{\overline{\mu} \in \mathrm{co}(\mathcal{M})} \mathrm{dec}_\gamma(\mathcal{M}; \delta_{\overline{\mu}}) + \gamma \log(|\mathcal{M}|/\delta) \right),$$

where

$$\sup_{\overline{\mu} \in \text{co}(\mathcal{M})} \text{dec}_\gamma(\mathcal{M}; \delta_{\overline{\mu}})$$

$$= \sup_{\overline{\mu} \in \text{co}(\mathcal{M})} \inf_{p \in \Delta(\Pi)} \sup_{M \in \mathcal{M}} \mathbb{E}_{\pi \sim p}\Big[ f^M(\pi_M) - f^M(\pi) - \gamma D_{\text{RL}}^2\Big( M(\pi), \mathbb{E}_{\overline{M} \sim \overline{\mu}}\big[\overline{M}(\pi)\big]\Big)\Big], \quad (8)$$

where $\mathbb{E}_{\overline{M} \sim \overline{\mu}}[\overline{M}(\pi)]$ denotes the mixture distribution of $\overline{M}(\pi)$ for $\overline{M} \sim \overline{\mu}$, and $\text{co}(\mathcal{M})$ denotes the set of all mixtures of models in $\mathcal{M}$ (which can be also identified with $\Delta(\mathcal{M})$, the set of all probability distributions over $\mathcal{M}$).

Compared with $\overline{\text{dec}}_\gamma$, Eq. (8) is different only in the place where the expectation $\mathbb{E}_{\overline{M} \sim \overline{\mu}}$ is taken. As $D_{\text{RL}}^2$ is convex in the second argument (by convexity of the squared Hellinger distance and linearity of $\mathbb{E}_{\overline{M} \sim \overline{\mu}} \mathbb{E}_{o \sim \mathbb{P}^M(\pi)}\Big[ \big\| \mathbf{R}^M(o) - \mathbf{R}^{\overline{M}}(o) \big\|_2^2 \Big]$ in $\overline{\mu}$), by Jensen's inequality, we have

$$\sup_{\overline{\mu} \in \text{co}(\mathcal{M})} \text{dec}_\gamma(\mathcal{M}; \delta_{\overline{\mu}}) \geqslant \overline{\text{dec}}_\gamma(\mathcal{M}).$$

Unfortunately, bounding $\sup_{\overline{\mu} \in \text{co}(\mathcal{M})} \text{dec}_\gamma(\mathcal{M}; \delta_{\overline{\mu}})$ requires handling the information gain with respect to the mixture model $\mathbb{E}_{\overline{M} \sim \overline{\mu}}[\overline{M}(\pi)]$, which is in general much harder than bounding the $\overline{\text{dec}}_\gamma(\mathcal{M})$ which only requires handling the expected information gain with respect to a proper model $\overline{M}(\pi)$ over $\overline{M} \sim \mu$. For general RL problems with $H > 1$, it is unclear whether $\sup_{\overline{\mu} \in \text{co}(\mathcal{M})} \text{dec}_\gamma(\mathcal{M}; \delta_{\overline{\mu}})$ admit bounds of the form $d/\gamma$ where $d$ is some complexity measure (Foster et al., 2021, Section 7.1.3). By contrast, our $\overline{\text{dec}}_\gamma(\mathcal{M})$ can be bounded for broad classes of problems (e.g. Section 5).

We also remark that an alternative approach considered in Foster et al. (2021, Theorem 4.1) depends on the following definition of DEC with respect to deterministic reference models:

$$\text{dec}_\gamma(\mathcal{M}, \overline{M}) := \inf_{p \in \Delta(\mathcal{M})} \sup_{M \in \mathcal{M}} \mathbb{E}_{\pi \sim p} \big[ f_M(\pi_M) - f_M(\pi) - \gamma D_{\text{RL}}^2(M(\pi), \overline{M}(\pi)) \big],$$

and the DEC of the model class $\mathcal{M}$ is simply $\text{dec}_\gamma(\mathcal{M}) := \sup_{\overline{M} \in \mathcal{M}} \text{dec}_\gamma(\mathcal{M}, \overline{M})$. As $\mathcal{M}$ (viewed as the set of all point masses) is a subset of $\text{co}(\mathcal{M})$, by definition we have

$$\text{dec}_\gamma(\mathcal{M}) \leqslant \overline{\text{dec}}_\gamma(\mathcal{M}),$$

therefore $\text{dec}_\gamma(\mathcal{M})$ can be bounded as long as $\overline{\text{dec}}_\gamma(\mathcal{M})$ can. However, that approach requires the online model estimation subroutine to output a *proper* estimator $\widehat{M}^t \in \mathcal{M}$ with bounded Hellinger error, which—unlike Tempered Aggregation (for the improper case)—requires problem-specific designs using prior knowledge about $\mathcal{M}$ and is unclear how to construct for general $\mathcal{M}$.

## C.2 AGGREGATION ALGORITHMS AS POSTERIOR COMPUTATIONS

We illustrate that Tempered Aggregation is equivalent to computing the *tempered posterior* (or *power posterior*) (Bhattacharya et al., 2019; Alquier & Ridgway, 2020) in the following vanilla Bayesian setting.

Consider a model class $\mathcal{M}$ associated with a prior $\mu^1 \in \Delta(\mathcal{M})$, and each model specifies a distribution $\mathbb{P}^M(\cdot) \in \Delta(\mathcal{O})$ of observations $o \in \mathcal{O}$. Suppose we receive observations $o^1, \dots, o^t, \dots$ in a sequential fashion. In this setting, the Tempered Aggregation updates

$$\mu^{t+1}(M) \propto_M \mu^t(M) \cdot \exp\big(\eta_{\text{p}} \log \mathbb{P}^M(o^t)\big) = \mu^t(M) \cdot \big(\mathbb{P}^M(o^t)\big)^{\eta_{\text{p}}}.$$

Therefore, for all $t \geqslant 1$,

$$\mu^{t+1}(M) \propto_M \mu^1(M) \cdot \left( \prod_{s=1}^t \mathbb{P}^M(o^s) \right)^{\eta_{\text{p}}}.$$

If $\eta_{\text{p}} = 1$ as in Vovk's aggregating algorithm (Vovk, 1995), by Bayes' rule, the above $\mu^{t+1}$ is exactly the posterior $M|o^{1:t}$. As we chose $\eta_{\text{p}} \leqslant 1/2 < 1$ in Tempered Aggregation, $\mu^{t+1}$ gives the tempered posterior, which is a slower variant of the posterior where data likelihoods are weighed less than in the exact posterior.

---

**Algorithm 2** TEMPERED AGGREGATION

---

**Input:** Learning rate $\eta_{\mathrm{p}} \in (0, \frac{1}{2}), \eta_{\mathrm{r}} > 0$.
1: Initialize $\mu^1 \leftarrow \mathrm{Unif}(\mathcal{M})$.
2: **for** $t = 1, \dots, T$ **do**
3:   Receive $(\pi^t, o^t, \mathbf{r}^t)$.
4:   Update randomized model estimator:

$$\mu^{t+1}(M) \propto_M \mu^t(M) \cdot \exp\left(\eta_{\mathrm{p}} \log \mathbb{P}^M(o^t|\pi^t) - \eta_{\mathrm{r}} \left\|\mathbf{r}^t - \mathbf{R}^M(o^t)\right\|_2^2\right). \tag{10}$$

---

## D    PROOFS FOR SECTION 3

### D.1    TEMPERED AGGREGATION

In this section, we analyze the Tempered Aggregation algorithm for finite model classes. For the sake of both generality and simplicity, we state our results in the following general setup of online model estimation. Lemma 3 (restated in Corollary D.2) then follows as a direct corollary.

**Setup: Online model estimation**    In an online model estimation problem, the learner is given a model set $\mathcal{M}$, a context space $\Pi$, an observation space $\mathcal{O}$, a family of conditional distributions $(\mathbb{P}^M(\cdot|\cdot) : \Pi \to \Delta(\mathcal{O}))_{M \in \mathcal{M}}$[7], a family of vector-valued mean reward functions $(\mathbf{R}^M : \mathcal{O} \to [0,1]^H)_{M \in \mathcal{M}}$. The environment fix a ground truth model $M^\star \in \mathcal{M}$; for shorthand, let $\mathbb{P}^\star := \mathbb{P}^{M^\star}, \mathbf{R}^\star := \mathbf{R}^{M^\star}$. For simplicity (in a measure-theoretic sense) we assume that $\mathcal{O}$ is finite[8].

At each step $t \in [T]$, the learner first determines a randomized model estimator (i.e. a distribution over models) $\mu^t \in \Delta(\mathcal{M})$. Then, the environment reveals the context $\pi^t \in \Pi$ (that is in general random and possibly depends on $\mu^t$ and history information), generates the observation $o^t \sim \mathbb{P}^\star(\cdot|\pi^t)$, and finally generates the reward $\mathbf{r}^t \in \mathbb{R}^d$ (which is a random vector) such that $\mathbb{E}\left[\mathbf{r}^t|o^t\right] = \mathbf{R}^\star(o^t)$. The information $(\pi^t, o^t, \mathbf{r}^t)$ may then be used by the learner to obtain the updated estimator $\mu^{t+1}$.

For any $M \in \mathcal{M}$, we consider the following estimation error of model $M$ with respect to the true model, at step $t$:

$$\mathrm{Err}_M^t := \mathbb{E}_t\left[D_{\mathrm{H}}^2\left(\mathbb{P}^M(\cdot|\pi^t), \mathbb{P}^\star(\cdot|\pi^t)\right) + \left\|\mathbf{R}^M(o^t) - \mathbf{R}^\star(o^t)\right\|_2^2\right], \tag{9}$$

where $\mathbb{E}_t$ is taken with respect to all randomness after prediction $\mu^t$ is made[9]—in particular it takes the expectation over $(\pi^t, o^t)$. Note that $\mathrm{Err}_{M^\star}^t = 0$ by definition.

**Algorithm and theoretical guarantee**    The Tempered Aggregation Algorithm is presented in Algorithm 2. Here we present the case with a finite model class ($|\mathcal{M}| < \infty$); In Appendix D.3 we treat the more general case of infinite model classes using covering arguments.

**Theorem D.1** (Tempered Aggregation). *Suppose $|\mathcal{M}| < \infty$, the reward vector $\mathbf{r}^t$ is $\sigma^2$-sub-Gaussian conditioned on $o^t$, and $\left\|\mathbf{R}^M(o^t) - \mathbf{R}^\star(o^t)\right\|_2 \le D$ almost surely for all $t \in [T]$. Then, Algorithm 2 with any learning rate $\eta_{\mathrm{p}}, \eta_{\mathrm{r}} > 0$ such that $2\eta_{\mathrm{p}} + 2\sigma^2\eta_{\mathrm{r}} < 1$ achieves the following with probability at least $1 - \delta$:*

$$\sum_{t=1}^T \mathbb{E}_{M \sim \mu^t}\left[\mathrm{Err}_M^t\right] \le C \log(|\mathcal{M}|/\delta),$$

*where $C = \max\left\{\frac{1}{\eta_{\mathrm{p}}}, \frac{1}{(1-2\eta_{\mathrm{p}})c'}\right\}$, $c' := (1 - e^{-c(1-2\sigma^2 c)D^2})/D^2$ and $c := \eta_{\mathrm{r}}/(1 - 2\eta_{\mathrm{p}})$ are constants depending on $(\eta_{\mathrm{p}}, \eta_{\mathrm{r}}, \sigma^2, D)$ only. Furthermore, it also achieves the following in-expectation*

---

[7]We use $\mathbb{P}^{M,\pi}(o)$ and $\mathbb{P}^M(o|\pi)$ interchangeably in the following.

[8]To extend to the continuous setting, only slight modifications are needed, see e.g. Foster et al. (2021, Section 3.2.3).

[9]In other words, $\mathbb{E}_t$ is the conditional expectation on $\mathcal{F}_{t-1} = \sigma(\mu^1, \pi^1, o^1, \mathbf{r}^1, \cdots, \pi^{t-1}, o^{t-1}, \mathbf{r}^{t-1}, \mu^t)$.

*guarantee:*

$$\mathbb{E}\left[\sum_{t=1}^{T}\mathbb{E}_{M\sim\mu^t}\big[\mathrm{Err}_M^t\big]\right]\leqslant C\log|\mathcal{M}|.$$

The proof of Theorem D.1 can be found in Appendix D.1.1.

Lemma 3 now follows as a direct corollary of Theorem D.1, which we restate and prove below.

**Corollary D.2** (Restatement of Lemma 3). *The Tempered Aggregation subroutine (3) in Algorithm 1 with $4\eta_{\mathrm{p}}+\eta_{\mathrm{r}}<2$ achieves the following bound with probability at least $1-\delta$:*

$$\mathbf{Est}_{\mathrm{RL}}:=\sum_{t=1}^{T}\mathbb{E}_{\pi^t\sim p^t}\mathbb{E}_{\widehat{M}^t\sim\mu^t}\Big[D_{\mathrm{RL}}^2(M^\star(\pi^t),\widehat{M}^t(\pi^t))\Big]\leqslant C\cdot\log(|\mathcal{M}|/\delta),$$

*where $C$ depends only on $(\eta_{\mathrm{p}},\eta_{\mathrm{r}})$. Specifically, we can choose $\eta_{\mathrm{p}}=\eta_{\mathrm{r}}=1/3$ and $C=10$. Furthermore, when $\eta_{\mathrm{p}}=\eta\in(0,\frac{1}{2}]$, $\eta_{\mathrm{r}}=0$, (3) achieves*

$$\mathbf{Est}_{\mathrm{H}}:=\sum_{t=1}^{T}\mathbb{E}_{\pi^t\sim p^t}\mathbb{E}_{\widehat{M}^t\sim\mu^t}\Big[D_{\mathrm{H}}^2\left(\mathbb{P}^{M^\star}(\pi^t),\mathsf{P}^{\widehat{M}^t}(\pi^t)\right)\Big]\leqslant\frac{1}{\eta}\cdot\log(|\mathcal{M}|/\delta)$$

*with probability at least $1-\delta$.*

*Proof.* Note that subroutine (3) in Algorithm 1 is exactly an instantiation of the Tempered Aggregation algorithm (Algorithm 2) with context $\pi^t$ sampled from distribution $p^t$ (which depends on $\mu^t$), observation $o^t$, and reward $\mathbf{r}^t$. Therefore, we can apply Theorem D.1, where we further note that $\mathbb{E}_{M\sim\mu^t}\big[\mathrm{Err}_M^t\big]$ corresponds exactly to

$$\mathbb{E}_{M\sim\mu^t}\big[\mathrm{Err}_M^t\big]=\mathbb{E}_{\widehat{M}^t\sim\mu^t}\mathbb{E}_{\pi^t\sim p^t}\left[D_{\mathrm{H}}^2(\mathsf{P}^{M^\star}(\pi^t),\mathsf{P}^{\widehat{M}^t}(\pi^t))+\mathbb{E}_{o\sim\mathsf{P}^{M^\star}(\pi^t)}\left\|\mathbf{R}^{M^\star}(o)-\mathbf{R}^{\widehat{M}^t}(o)\right\|_2^2\right]$$

$$=\mathbb{E}_{\widehat{M}^t\sim\mu^t}\mathbb{E}_{\pi^t\sim p^t}\Big[D_{\mathrm{RL}}^2(M^\star(\pi^t),\widehat{M}^t(\pi^t))\Big].$$

Notice that we can pick $\sigma^2=1/4$ and $D=\sqrt{2}$, as each individual reward $r_h\in[0,1]$ almost surely (so is $1/4$-sub-Gaussian by Hoeffding's Lemma), and

$$\left\|\mathbf{R}^M(o)-\mathbf{R}^{M'}(o)\right\|_2^2=\sum_{h=1}^{H}\left|R_h^M(o)-R_h^{M'}(o)\right|^2$$

$$\leqslant\sum_{h=1}^{H}\left|R_h^M(o)-R_h^{M'}(o)\right|\leqslant\sum_{h=1}^{H}\left|R_h^M(o)\right|+\left|R_h^{M'}(o)\right|=2.$$

for any two models $M,M'$ and any $o\in\mathcal{O}$. Therefore, Theorem D.1 yields that, as long as $4\eta_{\mathrm{p}}+\eta_{\mathrm{r}}<2$, we have with probability at least $1-\delta$ that

$$\mathbf{Est}_{\mathrm{RL}}:=\sum_{t=1}^{T}\mathbb{E}_{\pi^t\sim p^t}\mathbb{E}_{\widehat{M}^t\sim\mu^t}\Big[D_{\mathrm{RL}}^2(M^\star(\pi^t),\widehat{M}^t(\pi^t))\Big]\leqslant C\cdot\log(|\mathcal{M}|/\delta),$$

where $C=\max\left\{\frac{1}{\eta_{\mathrm{p}}},\frac{1}{(1-2\eta_{\mathrm{p}})c'}\right\}$, $c'=(1-e^{-c(2-c)})/2$, and $c=\eta_{\mathrm{r}}/(1-2\eta_{\mathrm{p}})$. Choosing $\eta_{\mathrm{p}}=\eta_{\mathrm{r}}=1/3$, we have $c=1$, $c'=(1-e^{-1})/2$, and $C=\max\{3,3/c'\}\leqslant 10$ by numerical calculations. This is the desired result. The case $\eta_{\mathrm{r}}=0$ follows similarly. $\square$

### D.1.1 PROOF OF THEOREM D.1

For all $t\in[T]$ define the random variable

$$\Delta^t:=-\log\mathbb{E}_{M\sim\mu^t}\left[\exp\left(\eta_{\mathrm{p}}\log\frac{\mathbb{P}^M(o^t|\pi^t)}{\mathbb{P}^\star(o^t|\pi^t)}+\eta_{\mathrm{r}}\delta_M^t\right)\right],$$

where

$$\delta_M^t := \left\| \mathbf{r}^t - \mathbf{R}^\star(o^t) \right\|_2^2 - \left\| \mathbf{r}^t - \mathbf{R}^M(o^t) \right\|_2^2. \tag{11}$$

Recall that $\mathbb{E}_t$ is taken with respect to all randomness after prediction $\mu^t$ is made. Then

$$
\begin{aligned}
\mathbb{E}_t\big[\exp\left(-\Delta^t\right)\big] &= \mathbb{E}_t\bigg[\mathbb{E}_{M\sim\mu^t}\bigg[\exp\bigg(\eta_{\mathrm{p}}\log\frac{\mathbb{P}^M(o^t|\pi^t)}{\mathbb{P}^\star(o^t|\pi^t)} + \eta_{\mathrm{r}}\delta_M^t\bigg)\bigg]\bigg] \\
&= \sum_{M\in\mathcal{M}}\mu^t(M)\mathbb{E}_t\bigg[\exp\bigg(\eta_{\mathrm{p}}\log\frac{\mathbb{P}^M(o^t|\pi^t)}{\mathbb{P}^\star(o^t|\pi^t)} + \eta_{\mathrm{r}}\delta_M^t\bigg)\bigg] \\
&\leqslant \sum_{M\in\mathcal{M}}\mu^t(M)\mathbb{E}_t\bigg[2\eta_{\mathrm{p}}\exp\bigg(\frac{1}{2}\log\frac{\mathbb{P}^M(o^t|\pi^t)}{\mathbb{P}^\star(o^t|\pi^t)}\bigg) + (1-2\eta_{\mathrm{p}})\exp\bigg(\frac{\eta_{\mathrm{r}}}{1-2\eta_{\mathrm{p}}}\delta_M^t\bigg)\bigg] \quad (12) \\
&= 2\eta_{\mathrm{p}}\sum_{M\in\mathcal{M}}\mu^t(M)\mathbb{E}_t\bigg[\mathbb{E}_{o\sim\mathbb{P}^\star(\cdot|\pi^t)}\bigg[\sqrt{\frac{\mathbb{P}^M(o|\pi^t)}{\mathbb{P}^\star(o|\pi^t)}}\bigg]\bigg] \\
&\quad + (1-2\eta_{\mathrm{p}})\sum_{M\in\mathcal{M}}\mu^t(M)\mathbb{E}_t\bigg[\exp\bigg(\frac{\eta_{\mathrm{r}}}{1-2\eta_{\mathrm{p}}}\delta_M^t\bigg)\bigg].
\end{aligned}
$$

For the first term, by definition

$$\mathbb{E}_{o\sim\mathbb{P}^\star(\cdot|\pi^t)}\bigg[\sqrt{\frac{\mathbb{P}^M(o|\pi^t)}{\mathbb{P}^\star(o|\pi^t)}}\bigg] = 1 - \frac{1}{2}D_{\mathrm{H}}^2(\mathbb{P}^\star(\cdot|\pi^t), \mathbb{P}^M(\cdot|\pi^t)). \tag{13}$$

To bound the second term, we abbreviate $c := \frac{\eta_{\mathrm{r}}}{1-2\eta_{\mathrm{p}}}$, and invoke the following lemma. The proof can be found in Appendix D.1.2.

**Lemma D.3.** *Suppose that $\mathbf{r} \in \mathbb{R}^d$ is a $\sigma^2$-sub-Gaussian random vector, $\bar{\mathbf{r}} = \mathbb{E}[\mathbf{r}]$ is the mean of $\mathbf{r}$, and $\hat{\mathbf{r}} \in \mathbb{R}^d$ is any fixed vector. Then the random variable*

$$\delta := \left\|\mathbf{r} - \bar{\mathbf{r}}\right\|_2^2 - \left\|\mathbf{r} - \hat{\mathbf{r}}\right\|_2^2,$$

*satisfies $\mathbb{E}[\exp(\lambda\delta)] \leqslant \exp\left(-\lambda(1-2\sigma^2\lambda)\left\|\bar{\mathbf{r}} - \hat{\mathbf{r}}\right\|_2^2\right)$ for any $\lambda \in \mathbb{R}$.*

Therefore,

$$
\begin{aligned}
\mathbb{E}_t\big[\exp\left(c\delta_M^t\right)\big] &\leqslant \mathbb{E}_t\bigg[\exp\left(-c(1-2\sigma^2c)\left\|\mathbf{R}^M(o^t) - \mathbf{R}^\star(o^t)\right\|_2^2\right)\bigg] \\
&\leqslant 1 - c'\mathbb{E}_t\bigg[\left\|\mathbf{R}^M(o^t) - \mathbf{R}^\star(o^t)\right\|_2^2\bigg], \tag{14}
\end{aligned}
$$

where the second inequality is due to the fact that for all $x \in [0, D^2]$, it holds that $e^{-c(1-2\sigma^2c)x} \leqslant 1 - c'x$, which is ensured by our choice of $c \in [0, 2/\sigma^2)$ and $c' := (1 - e^{-D^2c(1-2\sigma^2c)})/D^2 > 0$. Therefore, by flipping (12) and adding one on both sides, and plugging in (13) and (14), we get

$$
\begin{aligned}
1 - \mathbb{E}_t\big[\exp\left(-\Delta^t\right)\big] &\geqslant \eta_{\mathrm{p}}\mathbb{E}_{M\sim\mu^t}\mathbb{E}_{\pi^t\sim\cdot|\mathcal{F}_{t-1}}\big[D_{\mathrm{H}}^2(\mathbb{P}^M(\cdot|\pi^t), \mathbb{P}^\star(\cdot|\pi^t))\big] \\
&\quad + (1-2\eta_{\mathrm{p}})c'\mathbb{E}_{M\sim\mu^t}\mathbb{E}_t\bigg[\left\|\mathbf{R}^M(o^t) - \mathbf{R}^\star(o^t)\right\|_2^2\bigg].
\end{aligned}
$$

Thus, by martingale concentration (Lemma B.2), we have with probability at least $1 - \delta$ that

$$
\begin{aligned}
\sum_{t=1}^T \Delta^t + \log(1/\delta) &\geqslant \sum_{t=1}^T -\log\mathbb{E}_t\big[\exp\left(-\Delta^t\right)\big] \geqslant \sum_{t=1}^T 1 - \mathbb{E}_t\big[\exp\left(-\Delta^t\right)\big] \\
&\geqslant \eta_{\mathrm{p}}\sum_{t=1}^T\mathbb{E}_{M\sim\mu^t}\mathbb{E}_t\big[D_{\mathrm{H}}^2(\mathbb{P}^M(\cdot|\pi^t), \mathbb{P}^\star(\cdot|\pi^t))\big] \quad (15) \\
&\quad + (1-2\eta_{\mathrm{p}})c'\sum_{t=1}^T\mathbb{E}_{M\sim\mu^t}\mathbb{E}_t\bigg[\left\|\mathbf{R}^M(o^t) - \mathbf{R}^\star(o^t)\right\|_2^2\bigg] \\
&\geqslant \min\left\{\eta_{\mathrm{p}}, (1-2\eta_{\mathrm{p}})c'\right\} \cdot \mathbb{E}_{M\sim\mu^t}\big[\mathrm{Err}_M^t\big].
\end{aligned}
$$

It remains to upper bound $\sum_{t=1}^{T} \Delta^t$. Note that the update rule of Algorithm 2 can be written in the following Follow-The-Regularized-Leader form:

$$\mu^t(M) = \frac{\mu^1(M) \exp\left(\sum_{s \leqslant t-1} \eta_{\mathrm{p}} \log \mathbb{P}^M(o^s|\pi^s) + \eta_{\mathrm{r}} \delta_M^s\right)}{\sum_{M' \in \mathcal{M}} \mu^1(M') \exp\left(\sum_{s \leqslant t-1} \eta_{\mathrm{p}} \log \mathbb{P}^{M'}(o^s|\pi^s) + \eta_{\mathrm{r}} \delta_{M'}^s\right)},$$

where we have used that $\delta_M^t = -\left\|\mathbf{r}^t - \mathbf{R}^M(o^t)\right\|_2^2 + \left\|\mathbf{r}^t - \mathbf{R}^\star(o^t)\right\|_2^2$ in which $\left\|\mathbf{r}^t - \mathbf{R}^\star(o^t)\right\|_2^2$ is a constant that does not depend on $M$ for all $t \in [T]$. Therefore we have

$$\exp(-\Delta^t) = \mathbb{E}_{M \sim \mu^t}\left[\exp\left(\eta_{\mathrm{p}} \log \frac{\mathbb{P}^M(o^t|\pi^t)}{\mathbb{P}^\star(o^t|\pi^t)} + \eta_{\mathrm{r}} \delta_M^t\right)\right]$$

$$= \sum_{M \in \mathcal{M}} \mu^t(M) \exp\left(\eta_{\mathrm{p}} \log \frac{\mathbb{P}^M(o^t|\pi^t)}{\mathbb{P}^\star(o^t|\pi^t)} + \eta_{\mathrm{r}} \delta_M^t\right)$$

$$= \sum_{M \in \mathcal{M}} \frac{\mu^1(M) \exp\left(\sum_{s \leqslant t-1} \eta_{\mathrm{p}} \log \mathbb{P}^M(o^s|\pi^s) + \eta_{\mathrm{r}} \delta_M^s\right)}{\sum_{M' \in \mathcal{M}} \mu^1(M') \exp\left(\sum_{s \leqslant t-1} \eta_{\mathrm{p}} \log \mathbb{P}^{M'}(o^s|\pi^s) + \eta_{\mathrm{r}} \delta_{M'}^s\right)} \exp\left(\eta_{\mathrm{p}} \log \frac{\mathbb{P}^M(o^t|\pi^t)}{\mathbb{P}^\star(o^t|\pi^t)} + \eta_{\mathrm{r}} \delta_M^t\right)$$

$$= \frac{\sum_{M \in \mathcal{M}} \mu^1(M) \exp\left(\sum_{s \leqslant t} \eta_{\mathrm{p}} \log \frac{\mathbb{P}^M(o^s|\pi^s)}{\mathbb{P}^\star(o^s|\pi^s)} + \eta_{\mathrm{r}} \delta_M^s\right)}{\sum_{M \in \mathcal{M}} \mu^1(M) \exp\left(\sum_{s \leqslant t-1} \eta_{\mathrm{p}} \log \frac{\mathbb{P}^M(o^s|\pi^s)}{\mathbb{P}^\star(o^s|\pi^s)} + \eta_{\mathrm{r}} \delta_M^s\right)}, \tag{16}$$

where the last equality used again the fact that $-\eta_{\mathrm{p}} \log \mathbb{P}^\star(o^s|\pi^s)$ is a constant that does not depend on $M$ for all $s \in [t]$.

Taking $-\log$ on both sides above and summing over $t \in [T]$, we have by telescoping that

$$\sum_{t=1}^{T} \Delta^t = -\log \sum_{M \in \mathcal{M}} \mu^1(M) \exp\left(\sum_{t=1}^{T} \eta_{\mathrm{p}} \log \frac{\mathbb{P}^M(o^t|\pi^t)}{\mathbb{P}^\star(o^t|\pi^t)} + \eta_{\mathrm{r}} \delta_M^t\right). \tag{17}$$

By realizability $M^\star \in \mathcal{M}$, we have

$$\sum_{t=1}^{T} \Delta^t \leqslant -\log \mu^1(M^\star) = \log |\mathcal{M}|.$$

Plugging this bound into (15) gives the desired high-probability statement. The in-expectation statement follows similarly by further noticing that in (15), taking the expectation $\mathbb{E}\left[\sum_{t=1}^{T} \Delta^t\right]$ gives the same right-hand side, but without the additional $\log(1/\delta)$ term on the left-hand side. $\qquad\square$

### D.1.2   PROOF OF LEMMA D.3

By definition,

$$\delta = 2\left\langle \mathbf{r} - \bar{\mathbf{r}}, \hat{\mathbf{r}} - \bar{\mathbf{r}}\right\rangle - \left\|\hat{\mathbf{r}} - \bar{\mathbf{r}}\right\|_2^2,$$

and therefore,

$$\mathbb{E}[\exp(\lambda \delta)] = \exp\left(-\lambda \left\|\hat{\mathbf{r}} - \bar{\mathbf{r}}\right\|_2^2\right) \mathbb{E}[\exp(2\lambda \langle \mathbf{r} - \bar{\mathbf{r}}, \hat{\mathbf{r}} - \bar{\mathbf{r}}\rangle)]$$

$$\leqslant \exp\left(2\sigma^2 \lambda^2 \left\|\hat{\mathbf{r}} - \bar{\mathbf{r}}\right\|_2^2 - \lambda \left\|\hat{\mathbf{r}} - \bar{\mathbf{r}}\right\|_2^2\right)$$

$$= \exp\left(-\lambda(1 - 2\sigma^2 \lambda) \left\|\bar{\mathbf{r}} - \hat{\mathbf{r}}\right\|_2^2\right),$$

where the inequality is due to the definition of $\sigma^2$-sub-Gaussian random vector: For $\mathbf{v} = 2\lambda(\hat{\mathbf{r}} - \bar{\mathbf{r}}) \in \mathbb{R}^d$,

$$\mathbb{E}[\exp(\langle v, \mathbf{r}\rangle)] \leqslant \exp\left(\frac{\sigma^2 \left\|\mathbf{v}\right\|_2^2}{2}\right).$$

$\qquad\square$

---

**Algorithm 3** E2D Meta-Algorithm with Randomized Model Estimators

---

**Input:** Parameter $\gamma > 0$; Online estimation subroutine $\mathbf{Alg_{Est}}$; Prior distribution $\mu^1 \in \Delta(\mathcal{M})$.

1: **for** $t = 1, \ldots, T$ **do**
2:     Set $p^t \leftarrow \arg\min_{p \in \Delta(\Pi)} \widehat{V}_\gamma^{\mu^t}(p)$, where $\widehat{V}_\gamma^{\mu^t}$ is defined in (2).
3:     Sample $\pi^t \sim p^t$. Execute $\pi^t$ and observe $(o^t, \mathbf{r}^t)$.
4:     Update randomized model estimator by online estimation subroutine:

$$\mu^{t+1} \leftarrow \mathbf{Alg}_{\mathbf{Est}}^t\Big(\{(\pi^s, o^s, \mathbf{r}^s)\}_{s \in [t]}\Big).$$

---

## D.2 GENERAL E2D & PROOF OF THEOREM 2

We first prove a guarantee for the following E2D meta-algorithm that allows any (randomized) online estimation subroutine, which includes Algorithm 1 as a special case by instantiating $\mathbf{Alg_{Est}}$ as the Tempered Aggregation subroutine (for finite model classes) and thus proving Theorem 2.

The following theorem is an instantiation of Foster et al. (2021, Theorem 4.3) by choosing the divergence function to be $D_{\mathrm{RL}}$. For completeness, we provide a proof in Appendix D.4. Let

$$\mathbf{Est}_{\mathrm{RL}} := \sum_{t=1}^{T} \mathbb{E}_{\pi^t \sim p^t} \mathbb{E}_{\widehat{M}^t \sim \mu^t}\Big[ D_{\mathrm{RL}}^2(\widehat{M}^t(\pi^t), M^\star(\pi^t)) \Big] \tag{18}$$

denote the online estimation error of $\{\mu^t\}_{t=1}^{T}$ in $D_{\mathrm{RL}}^2$ divergence (achieved by $\mathbf{Alg_{Est}}$).

**Theorem D.4** (E2D Meta-Algorithm (Foster et al., 2021)). *Algorithm 3 achieves*

$$\mathbf{Reg_{DM}} \leqslant T \cdot \overline{\mathrm{dec}}_\gamma(\mathcal{M}) + \gamma \cdot \mathbf{Est}_{\mathrm{RL}}.$$

We are now ready to prove the main theorem (finite $\mathcal{M}$).

**Proof of Theorem 2**      Note that Algorithm 1 is an instantiation of Algorithm 3 with $\mathbf{Alg_{Est}}$ chosen as Tempered Aggregation. By Lemma 3, choosing $\eta_{\mathrm{p}} = \eta_{\mathrm{r}} = 1/3$, the Tempered Aggregation subroutine achieves

$$\mathbf{Est}_{\mathrm{RL}} \leqslant 10 \log(|\mathcal{M}|/\delta)$$

with probability at least $1 - \delta$. On this event, by Theorem D.4 we have that

$$\mathbf{Reg_{DM}} \leqslant T \cdot \overline{\mathrm{dec}}_\gamma(\mathcal{M}) + \gamma \cdot \mathbf{Est}_{\mathrm{RL}} \leqslant T \cdot \overline{\mathrm{dec}}_\gamma(\mathcal{M}) + 10\gamma \log(|\mathcal{M}|/\delta).$$

This is the desired result. $\qquad\qquad\qquad\qquad\qquad\qquad\qquad\qquad\qquad\qquad\qquad\qquad\qquad$ $\square$

## D.3 E2D-TA WITH COVERING

In many scenarios, we have to work with an infinite model class $\mathcal{M}$ instead of a finite one. In the following, we define a covering number suitable for divergence $D_{\mathrm{RL}}$, and provide the analysis of the Tempered Aggregation subroutine (as well as the corresponding E2D-TA algorithm) with such coverings.

We consider the following definition of optimistic covering.

**Definition D.5** (Optimistic covering). *Given $\rho \in [0, 1]$, an optimistic $\rho$-cover of $\mathcal{M}$ is a tuple $(\widetilde{\mathbb{P}}, \mathcal{M}_0)$, where $\mathcal{M}_0$ is a finite subset of $\mathcal{M}$, and each $M_0 \in \mathcal{M}_0$ is assigned with an* optimistic likelihood function $\widetilde{\mathbb{P}}^{M_0}$, *such that the following holds:*

*(1) For $M_0 \in \mathcal{M}_0$, for each $\pi$, $\widetilde{\mathbb{P}}^{M_0,\pi}(\cdot)$ specifies a un-normalized distribution over $\mathcal{O}$, and it holds that $\left\| \mathbb{P}^{M_0,\pi}(\cdot) - \widetilde{\mathbb{P}}^{M_0,\pi}(\cdot) \right\|_1 \leqslant \rho^2$.*

*(2) For any $M \in \mathcal{M}$, there exists a $M_0 \in \mathcal{M}_0$ that* covers *$M$: for all $\pi \in \Pi$, $o \in \mathcal{O}$, it holds $\widetilde{\mathbb{P}}^{M_0,\pi}(o) \geq \mathbb{P}^{M,\pi}(o)$[10], and $\left\| \mathbf{R}^M(o) - \mathbf{R}^{M_0}(o) \right\|_1 \leq \rho$.*

*The optimistic covering number $\mathcal{N}(\mathcal{M}, \rho)$ is defined as the minimal cardinality of $\mathcal{M}_0$ such that there exists $\widetilde{\mathbb{P}}$ such that $(\widetilde{\mathbb{P}}, \mathcal{M}_0)$ is an optimistic $\rho$-cover of $\mathcal{M}$.*

With the definition of an optimistic covering at hand, the Tempered Aggregation algorithm can be directly generalized to infinite model classes by performing the updates on an optimistic cover (Algorithm 4).

**Proposition D.6** (Tempered Aggregation with covering for RL)**.** *For any model class $\mathcal{M}$ and an associated optimistic $\rho$-cover $(\widetilde{\mathbb{P}}, \mathcal{M}_0)$, the Tempered Aggregation subroutine*

$$\mu^{t+1}(M) \propto_M \mu^t(M) \cdot \exp \left( \eta_{\mathrm{p}} \log \widetilde{\mathbb{P}}^{M,\pi^t}(o^t) - \eta_{\mathrm{r}} \left\| \mathbf{r}^t - \mathbf{R}^M(o^t) \right\|_2^2 \right) \qquad (19)$$

*with $\mu^1 = \mathrm{Unif}(\mathcal{M}_0)$ and $\eta_{\mathrm{p}} = \eta_{\mathrm{r}} = 1/3$ achieves the following bound with probability at least $1 - \delta$:*

$$\mathbf{Est}_{\mathrm{RL}} \leq 10 \cdot \left[ \log |\mathcal{M}_0| + 2T\rho + 2\log(2/\delta) \right].$$

**E2D-TA with covering**    Define

$$\mathrm{est}(\mathcal{M}, K) := \inf_{\rho \geq 0} \left( \log \mathcal{N}(\mathcal{M}, \rho) + K\rho \right) \qquad (20)$$

which measures the estimation complexity of $\mathcal{M}$ for $K$-step interaction.

Proposition D.6 implies that, based on the model class $\mathcal{M}$, we can suitably design the optimistic likelihood function $\widetilde{\mathbb{P}}$ and the prior $\mu^1$, so that Algorithm 3 with $\mathbf{Alg_{Est}}$ chosen as (19) achieves $\mathbf{Est}_{\mathrm{RL}} = \widetilde{\mathcal{O}}\left( \mathrm{est}(\mathcal{M}, T) \right)$. Therefore, by Theorem D.4 we directly have the following guarantee.

**Theorem D.7** (E2D-TA with covering)**.** *Algorithm 3 with $\mathbf{Alg_{Est}}$ chosen as* TEMPERED AGGRE-GATION WITH COVERING *(19) and optimally chosen $\gamma$ achieves*

$$\mathbf{Reg_{DM}} \leq C \inf_{\gamma > 0} \left( T \cdot \overline{\mathrm{dec}}_\gamma(\mathcal{M}) + \gamma\, \mathrm{est}(\mathcal{M}, T) + \gamma \log(1/\delta) \right)$$

*with probability at least $1 - \delta$, where $C$ is a universal constant.*

### D.3.1    DISCUSSIONS ABOUT OPTIMISTIC COVERING

We make a few remarks regarding our definition of the optimistic covering. Examples of optimistic covers on concrete model classes can be found in e.g. Example K.13, Proposition K.15; see also (Liu et al., 2022a, Appendix B).

**A more relaxed definition**    We first remark that Definition D.5(2) can actually be relaxed to

*(2') For any $M \in \mathcal{M}$, there exists a $M_0 \in \mathcal{M}_0$, such that $\max_{o \in \mathcal{O}} \left\| \mathbf{R}^M(o) - \mathbf{R}^{M_0}(o) \right\|_1 \leq \rho$, and*

$$\mathbb{E}_{o \sim \mathbb{P}^M(\cdot|\pi)} \left[ \frac{\mathbb{P}^M(o|\pi)}{\widetilde{\mathbb{P}}^{M_0}(o|\pi)} \right] \leq 1 + \rho, \quad \forall \pi \in \Pi. \qquad (\dagger)$$

For the simplicity of presentation, we state all the results in terms of Definition D.5. But the proof of Theorem D.8 can be directly adapted to ($\dagger$); see Remark D.9.

**Relation to Foster et al. (2021, Definition 3.2)**    We comment on the relationship between our optimistic covering and the covering introduced in Foster et al. (2021, Definition 3.2) (which is also used in their algorithms to handle infinite model classes). First, the covering in Foster et al. (2021) needs to cover *the distribution of reward*, while ours only need to cover the mean reward function. More importantly, Foster et al. (2021, Lemma A.16) explicitly introduces a factor $\log B$, where

---

[10]An important observation is that, along with (1), this requirement implies $D_{\mathrm{TV}}\left( \mathbb{P}^{M,\pi}(\cdot), \mathbb{P}^{M_0,\pi}(\cdot) \right) \leq \rho^2$ (for proof, see e.g. (33)). Therefore, a $\rho$-optimistic covering must be a $\rho^2$-covering in TV distance.

---

**Algorithm 4** TEMPERED AGGREGATION WITH COVERING

---

**Input:** Learning rate $\eta_{\mathrm{p}} \in (0, \frac{1}{2}), \eta_{\mathrm{r}} > 0$, number of steps $T$, $\rho$-optimistic cover $(\widetilde{\mathbb{P}}, \mathcal{M}_0)$.

1: Initialize $\mu^1 \leftarrow \mathrm{Unif}(\mathcal{M}_0)$.
2: **for** $t = 1, \ldots, T$ **do**
3:     Receive $(\pi^t, o^t, \mathbf{r}^t)$.
4:     Update randomized model estimator:

$$\mu^{t+1}(M) \propto_M \mu^t(M) \cdot \exp\left(\eta_{\mathrm{p}} \log \widetilde{\mathbb{P}}^M(o^t|\pi^t) - \eta_{\mathrm{r}} \left\|\mathbf{r}^t - \mathbf{R}^M(o^t)\right\|_2^2\right).$$

---

$B \geqslant \sup_{o\in\mathcal{O},\pi\in\Pi,M\in\mathcal{M}} \frac{\mathbb{P}^M(o|\pi)}{\nu(o|\pi)}$ with $\nu$ being certain base distribution. Actually, with such a $B$, we can show that

$$\mathcal{N}'(\mathcal{M}, \rho) \leqslant \mathcal{N}_{\mathrm{TV}}(\mathcal{M}, \rho^2/4B),$$

where $\mathcal{N}_{\mathrm{TV}}$ is the covering number in the TV sense, and $\mathcal{N}'$ is the optimistic covering number with respect to (†).

**Relation to other notions of covering numbers** Ignoring the reward component, our optimistic covering number is essentially equivalent to the *bracketing number*. We further remark that optimistic covering can be slightly weaker than the covering in $\chi^2$-distance sense: given a $\rho^2$-covering $\mathcal{M}_0$ in the latter sense, we can take $\widetilde{\mathbb{P}} = (1 + \rho^2)\mathbb{P}$ to obtain a $\rho$-optimistic covering defined by (†).

### D.4 PROOF OF THEOREM D.4

We have by definition of $\mathbf{Reg}_{\mathbf{DM}}$ that

$$
\begin{aligned}
\mathbf{Reg}_{\mathbf{DM}} &= \sum_{t=1}^T \mathbb{E}_{\pi^t\sim p^t}\left[f^{M^\star}(\pi_{M^\star}) - f^{M^\star}(\pi^t)\right] \\
&= \sum_{t=1}^T \mathbb{E}_{\pi^t\sim p^t}\left[f^{M^\star}(\pi_{M^\star}) - f^{M^\star}(\pi^t) - \gamma\mathbb{E}_{\widehat{M}^t\sim\mu^t}\left[D_{\mathrm{RL}}^2(M^\star(\pi^t), \widehat{M}^t(\pi^t))\right]\right] \\
&\quad + \gamma \cdot \sum_{t=1}^T \mathbb{E}_{\pi^t\sim p^t}\mathbb{E}_{\widehat{M}^t\sim\mu^t}\left[D_{\mathrm{RL}}^2(M^\star(\pi^t), \widehat{M}^t(\pi^t))\right] \\
&\overset{(i)}{\leqslant} \sum_{t=1}^T \sup_{M\in\mathcal{M}} \mathbb{E}_{\pi^t\sim p^t}\mathbb{E}_{\overline{M}\sim\mu^t}\left[f^M(\pi_M) - f^M(\pi^t) - \gamma D_{\mathrm{RL}}^2(M(\pi^t), \overline{M}(\pi^t))\right] + \gamma \cdot \mathbf{Est}_{\mathrm{RL}} \\
&\overset{(ii)}{=} \sum_{t=1}^T \underbrace{\widehat{V}_\gamma^{\mu^t}(p^t)}_{=\inf_{p\in\Delta(\Pi)}\widehat{V}_\gamma^{\mu^t}(p)} + \gamma \cdot \mathbf{Est}_{\mathrm{RL}} \\
&\overset{(iii)}{=} \sum_{t=1}^T \mathrm{dec}_\gamma(\mathcal{M}, \mu^t) + \gamma \cdot \mathbf{Est}_{\mathrm{RL}} \leqslant T \cdot \overline{\mathrm{dec}}_\gamma(\mathcal{M}) + \gamma \cdot \mathbf{Est}_{\mathrm{RL}}.
\end{aligned}
$$

Above, (i) follows by the realizability assumption $M^\star \in \mathcal{M}$; (ii) follows by definition of the risk $\widehat{V}_\gamma^{\mu^t}$ (cf. (2)) as well as the fact that $p^t$ minimizes $\widehat{V}_\gamma^{\mu^t}(\cdot)$ in Algorithm 3; (iii) follows by definition of $\mathrm{dec}_\gamma(\mathcal{M}, \mu^t)$. This completes the proof. $\qquad\square$

### D.5 PROOF OF PROPOSITION D.6

We first restates the TEMPERED AGGREGATION WITH COVERING subroutine (19) in the general setup of online model estimation in Algorithm 4.

**Theorem D.8** (Tempered Aggregation over covering). *For any $\mathcal{M}$ that is not necessarily finite, but otherwise under the same setting as Theorem D.1, Algorithm 4 with $2\eta_{\mathrm{p}} + 2\sigma^2\eta_{\mathrm{r}} < 1$ achieves with*

*probability at least $1 - \delta$ that*

$$\sum_{t=1}^{T} \mathbb{E}_{M \sim \mu^t} \left[ \mathrm{Err}_M^t \right] \leqslant C[\log |\mathcal{M}_0| + 2 \log(2/\delta) + 2T\rho(\eta_{\mathrm{r}} + \eta_{\mathrm{p}})],$$

*where $C$ is defined same as in Theorem D.1.*

Plugging Theorem D.8 into the RL setting, picking $(\eta_{\mathrm{p}}, \eta_{\mathrm{r}})$ and performing numerical calculations, we directly have the Proposition D.6. The proof follows the same arguments as Corollary D.2 and hence omitted. Similarly, when $\eta_{\mathrm{p}} = \eta \in (0, \frac{1}{2}), \eta_{\mathrm{r}} = 0$, the proof of Theorem D.8 implies that (19) with $\mu^1 = \mathrm{Unif}(\mathcal{M}_0)$ achieves the following bound with probability at least $1 - \delta$:

$$\sum_{t=1}^{T} \mathbb{E}_{\pi^t \sim p^t} \mathbb{E}_{\widehat{M}^t \sim \mu^t} \left[ D_{\mathrm{H}}^2 \left( \mathsf{P}^{M^\star}(\pi^t), \mathsf{P}^{\widehat{M}^t}(\pi^t) \right) \right] \leqslant \frac{1}{\eta} \cdot [\log |\mathcal{M}_0| + 2\eta T \rho + 2 \log(2/\delta)]. \quad (21)$$

**Proof of Theorem D.8**     The proof is similar to that of Theorem D.1. Consider the random variable

$$\Delta^t := -\log \mathbb{E}_{M \sim \mu^t} \left[ \exp \left( \eta_{\mathrm{p}} \log \frac{\widetilde{\mathbb{P}}^M(o^t|\pi^t)}{\mathbb{P}^\star(o^t|\pi^t)} + \eta_{\mathrm{r}} \delta_M^t \right) \right],$$

for all $t \in [T]$, where $\delta$ is defined in (11). Then by (12) and (14), we have

$$\mathbb{E}_t \left[ \exp \left( -\Delta^t \right) \right] \leqslant 2\eta_{\mathrm{p}} \mathbb{E}_{M \sim \mu^t} \mathbb{E}_t \left[ \sqrt{\frac{\widetilde{\mathbb{P}}^M(o^t|\pi^t)}{\mathbb{P}^\star(o^t|\pi^t)}} \right]$$

$$+ (1 - 2\eta_{\mathrm{p}}) \left( 1 - c' \mathbb{E}_{M \sim \mu^t} \mathbb{E}_t \left[ \left\| \mathbf{R}^M(o^t) - \mathbf{R}^\star(o^t) \right\|_2^2 \right] \right),$$

where $c'$ is the same as in Theorem D.1. To bound the first term, we notice that for all $\pi \in \Pi$, and $o \sim \mathbb{P}^\star(\cdot|\pi)$, we have

$$\mathbb{E}_{o \sim \mathbb{P}^\star(\cdot|\pi)} \left[ \sqrt{\frac{\widetilde{\mathbb{P}}^M(o|\pi)}{\mathbb{P}^\star(o|\pi)}} \right] = \mathbb{E}_{o \sim \mathbb{P}^\star(\cdot|\pi)} \left[ \sqrt{\frac{\mathbb{P}^M(o|\pi)}{\mathbb{P}^\star(o|\pi)}} \right] + \mathbb{E}_{o \sim \mathbb{P}^\star(\cdot|\pi)} \left[ \frac{\sqrt{\widetilde{\mathbb{P}}^M(o|\pi)} - \sqrt{\mathbb{P}^M(o|\pi)}}{\sqrt{\mathbb{P}^\star(o|\pi)}} \right]$$

$$\leqslant 1 - \frac{1}{2} D_{\mathrm{H}}^2(\mathbb{P}^M(\cdot|\pi), \mathbb{P}^\star(\cdot|\pi)) + \mathbb{E}_{o \sim \mathbb{P}^\star(\cdot|\pi)} \left[ \frac{\left( \sqrt{\widetilde{\mathbb{P}}^M(o|\pi)} - \sqrt{\mathbb{P}^M(o|\pi)} \right)^2}{\mathbb{P}^\star(o|\pi)} \right]^{\frac{1}{2}}$$

$$\leqslant 1 - \frac{1}{2} D_{\mathrm{H}}^2(\mathbb{P}^M(\cdot|\pi), \mathbb{P}^\star(\cdot|\pi)) + \mathbb{E}_{o \sim \mathbb{P}^\star(\cdot|\pi)} \left[ \frac{\left| \widetilde{\mathbb{P}}^M(o|\pi) - \mathbb{P}^M(o|\pi) \right|}{\mathbb{P}^\star(o|\pi)} \right]^{\frac{1}{2}}$$

$$= 1 - \frac{1}{2} D_{\mathrm{H}}^2(\mathbb{P}^M(\cdot|\pi), \mathbb{P}^\star(\cdot|\pi)) + \left\| \widetilde{\mathbb{P}}^M(\cdot|\pi) - \mathbb{P}^M(\cdot|\pi) \right\|_1^{\frac{1}{2}}$$

$$\leqslant 1 - \frac{1}{2} D_{\mathrm{H}}^2(\mathbb{P}^M(\cdot|\pi), \mathbb{P}^\star(\cdot|\pi)) + \rho, \quad (22)$$

where the last inequality is due to the fact that $\left\| \mathbb{P}^M(\cdot|\pi) - \widetilde{\mathbb{P}}^M(\cdot|\pi) \right\|_1 \leqslant \rho^2$. (22) directly implies that

$$\mathbb{E}_t \left[ \sqrt{\frac{\widetilde{\mathbb{P}}^M(o^t|\pi^t)}{\mathbb{P}^\star(o^t|\pi^t)}} \right] \leqslant 1 - \frac{1}{2} \mathbb{E}_t \left[ D_{\mathrm{H}}^2(\mathbb{P}^M(\cdot|\pi^t), \mathbb{P}^\star(\cdot|\pi^t)) \right] + \rho. \quad (23)$$

Therefore, by Lemma B.2, with probability at least $1 - \delta/2$, it holds that

$$\sum_{t=1}^{T} \Delta^t + \log(2/\delta) \geqslant \sum_{t=1}^{T} -\log \mathbb{E}_t \left[ \exp \left( -\Delta^t \right) \right] \geqslant \sum_{t=1}^{T} 1 - \mathbb{E}_t \left[ \exp \left( -\Delta^t \right) \right]$$

$$\geqslant \eta_{\mathrm{p}}\left[\sum_{t=1}^{T} \mathbb{E}_{M\sim\mu^t}\mathbb{E}_t\big[D_{\mathrm{H}}^2(\mathbb{P}^M(\cdot|\pi^t),\mathbb{P}^\star(\cdot|\pi^t))\big]-2T\rho\right]$$
$$+(1-2\eta_{\mathrm{p}})c'\sum_{t=1}^{T}\mathbb{E}_{M\sim\mu^t}\mathbb{E}_t\Big[\big\|\mathbf{R}^M(o^t)-\mathbf{R}^\star(o^t)\big\|_2^2\Big].$$

In the following, we complete the proof by showing that with probability at least $1-\delta/2$,

$$\sum_{t=1}^{T}\Delta^t\leqslant\log|\mathcal{M}_0|+2T\eta_{\mathrm{r}}\rho+\log(2/\delta).$$

By a telescoping argument same as (17), we have

$$\sum_{t=1}^{T}\Delta^t=-\log\sum_{M\in\mathcal{M}_0}\mu^1(M)\exp\left(\sum_{t=1}^{T}\eta_{\mathrm{p}}\log\frac{\widetilde{\mathbb{P}}^M(o^t|\pi^t)}{\mathbb{P}^\star(o^t|\pi^t)}+\eta_{\mathrm{r}}\delta_M^t\right). \tag{24}$$

By the definition of $\mathcal{M}_0$ and the realizability $M^\star\in\mathcal{M}$, there exists a $M\in\mathcal{M}_0$ such that $M^\star$ is covered by $M$ (i.e. $\big\|\mathbf{R}^{M_0}(o)-\mathbf{R}^\star(o)\big\|_\infty\leqslant\rho$ and $\widetilde{\mathbb{P}}^M(\cdot|\pi)\geqslant\mathbb{P}^\star(\cdot|\pi)$ for all $\pi$). Then

$$\mathbb{E}\left[\exp\left(\sum_{t=1}^{T}\Delta^t\right)\right]\leqslant|\mathcal{M}_0|\,\mathbb{E}\left[\exp\left(-\sum_{t=1}^{T}\eta_{\mathrm{p}}\log\frac{\widetilde{\mathbb{P}}^M(o^t|\pi^t)}{\mathbb{P}^\star(o^t|\pi^t)}-\eta_{\mathrm{r}}\delta_M^t\right)\right]. \tag{25}$$

Now

$$\begin{aligned}
\mathbb{E}\left[\exp\left(-\sum_{t=1}^{T}\eta_{\mathrm{p}}\log\frac{\widetilde{\mathbb{P}}^M(o^t|\pi^t)}{\mathbb{P}^\star(o^t|\pi^t)}-\eta_{\mathrm{r}}\delta_M^t\right)\right]&=\mathbb{E}\left[\prod_{t=1}^{T}\left(\frac{\mathbb{P}^\star(o^t|\pi^t)}{\widetilde{\mathbb{P}}^M(o^t|\pi^t)}\right)^{\eta_{\mathrm{p}}}\cdot\exp(-\eta_{\mathrm{r}}\delta_M^t)\right]\\
&\leqslant\mathbb{E}\left[\prod_{t=1}^{T}\exp(-\eta_{\mathrm{r}}\delta_M^t)\right]=\mathbb{E}\left[\prod_{t=1}^{T-1}\exp(-\eta_{\mathrm{r}}\delta_M^t)\cdot\mathbb{E}\left[\exp(-\eta_{\mathrm{r}}\delta_M^T)\big|\,o^T\right]\right]\\
&\leqslant\exp(2\rho\eta_{\mathrm{r}})\mathbb{E}\left[\prod_{t=1}^{T-1}\exp(-\eta_{\mathrm{r}}\delta_M^t)\right]\\
&\leqslant\cdots\leqslant\exp(2T\rho\eta_{\mathrm{r}}),
\end{aligned} \tag{26}$$

where the first inequality is due to $\widetilde{\mathbb{P}}^M\geqslant\mathbb{P}^\star$, the second inequality is because for all $t\in[T]$,

$$\mathbb{E}\left[\exp(-\eta_{\mathrm{r}}\delta_M^t)\big|\,o^t\right]\leqslant\exp\left(\eta_{\mathrm{r}}(1+2\sigma^2\eta_{\mathrm{r}})\big\|\mathbf{R}^M(o^t)-\mathbf{R}^\star(o^t)\big\|_2^2\right)\leqslant\exp(2\rho\eta_{\mathrm{r}}),$$

which is due to Lemma D.3 and $\big\|\mathbf{R}^M(o^t)-\mathbf{R}^\star(o^t)\big\|_2^2\leqslant\big\|\mathbf{R}^M(o^t)-\mathbf{R}^\star(o^t)\big\|_1\big\|\mathbf{R}^M(o^t)-\mathbf{R}^\star(o^t)\big\|_\infty\leqslant\rho$. Applying Chernoff's bound completes the proof. $\qquad\square$

**Remark D.9.** From the proof above, it is clear that Theorem D.8 also holds for for the alternative definition of covering number in (†): Under that definition, we can proceed in (26) by using the fact $\mathbb{E}_{o\sim\mathbb{P}^\star(\cdot|\pi)}\left[\frac{\mathbb{P}^\star(o|\pi)}{\widetilde{\mathbb{P}}^M(o|\pi)}\right]\leqslant 1+\rho$ and the fact $\mathbb{E}\left[\exp(-\eta_{\mathrm{r}}\delta_M^t)\big|\,o^t\right]\leqslant\exp(2\rho\eta_{\mathrm{r}})$ alternately.

## E   CONNECTIONS TO OPTIMISTIC ALGORITHMS

Motivated by the close connection between E2D-TA and posteriors/likelihoods, in this section, we re-analyze two existing optimistic algorithms: Model-based Optimistic Posterior Sampling (MOPS), and Optimistic Maximum Likelihood Estimation (OMLE), in a parallel fashion to E2D-TA. We show that these two algorithms—in addition to algorithmic similarity to E2D-TA—work under general structural conditions related to the DEC, for which we establish formal relationships.

### E.1 Model-based Optimistic Posterior Sampling (MOPS)

We consider the following version of the MOPS algorithm of Agarwal & Zhang (2022a)[11]. Similar as E2D-TA, MOPS also maintains a posterior $\mu^t \in \Delta(\mathcal{M})$ over models, initialized at a suitable prior $\mu^1$. The policy in the $t$-th episode is directly obtained by posterior sampling: $\pi^t = \pi_{M^t}$ where $M^t \sim \mu^t$. After executing $\pi^t$ and observing $(o^t, r^t)$, the algorithm updates the posterior as

$$\mu^{t+1}(M) \propto_M \mu^t(M) \cdot \exp\left(\gamma^{-1} f^M(\pi_M) + \eta_{\mathrm{p}} \log \mathbb{P}^{M,\pi^t}(o^t) - \eta_{\mathrm{r}} \left\|\mathbf{r}^t - \mathbf{R}^M(o^t)\right\|_2^2\right). \quad (27)$$

This update is similar as Tempered Aggregation (3), and differs in the additional *optimism* term $\gamma^{-1} f^M(\pi_M)$ which favors models with higher optimal values. (Full algorithm in Algorithm 5.)

We now state the structural condition and theoretical guarantee for the MOPS algorithm.

**Definition E.1** (Posterior sampling coefficient). *The Posterior Sampling Coefficient (PSC) of model class $\mathcal{M}$ with respect to reference model $\overline{M} \in \mathcal{M}$ and parameter $\gamma > 0$ is defined as*

$$\mathrm{psc}_\gamma(\mathcal{M}, \overline{M}) := \sup_{\mu \in \Delta(\mathcal{M})} \mathbb{E}_{M \sim \mu} \mathbb{E}_{M' \sim \mu} \left[ f^M(\pi_M) - f^{\overline{M}}(\pi_M) - \gamma D_{\mathrm{RL}}^2(\overline{M}(\pi_{M'}), M(\pi_{M'})) \right].$$

**Theorem E.2** (Regret bound for MOPS). *Choosing $\eta_{\mathrm{p}} = 1/6$, $\eta_{\mathrm{r}} = 0.6$ and the uniform prior $\mu^1 = \mathrm{Unif}(\mathcal{M})$, Algorithm 5 achieves the following with probability at least $1 - \delta$:*

$$\mathbf{Reg_{DM}} \leqslant T\left[\mathrm{psc}_{\gamma/6}(\mathcal{M}, M^\star) + 2/\gamma\right] + 4\gamma \cdot \log(|\mathcal{M}|/\delta).$$

Theorem E.2 (proof in Appendix F.1) is similar as Agarwal & Zhang (2022a, Theorem 1) and is slightly more general in the assumed structural condition, as the PSC is bounded whenever the "Hellinger decoupling coefficient" used in their theorem is bounded (Proposition F.5).

**Relationship between DEC and PSC** The definition of the PSC resembles that of the DEC. We show that the DEC can indeed be upper bounded by the PSC modulo a (lower-order) additive constant; in other words, low PSC implies a low DEC. The proof can be found in Appendix F.4.

**Proposition E.3** (Bounding DEC by PSC). *Suppose $\Pi$ is finite, then we have for any $\gamma > 0$ that*

$$\overline{\mathrm{dec}}_\gamma(\mathcal{M}) \leqslant \sup_{\overline{M} \in \mathcal{M}} \mathrm{psc}_{\gamma/6}(\mathcal{M}, \overline{M}) + 2(H+1)/\gamma.$$

### E.2 Optimistic Maximum Likelihood Estimation (OMLE)

Standard versions of the OMLE algorithm (e.g. Liu et al. (2022a)) use the log-likelihood of all observed data as the risk function. Here we consider the following risk function involving the log-likelihood of the observations and the negative $L_2^2$ loss of the rewards, to be parallel with E2D-TA and MOPS:

$$\mathcal{L}_t(M) := \sum_{s=1}^{t-1} \left[\log \mathbb{P}^{M,\pi^s}(o^s) - \left\|\mathbf{r}^s - \mathbf{R}^M(o^s)\right\|_2^2\right]. \quad (28)$$

In the $t$-th iteration, the OMLE algorithm plays the greedy policy of the most optimistic model within a $\beta$-superlevel set of the above risk (Full algorithm in Algorithm 6):

$$(M^t, \pi^t) := \underset{(M,\pi) \in \mathcal{M} \times \Pi}{\arg\max} f^M(\pi) \quad \text{such that} \quad \mathcal{L}_t(M) \geqslant \max_{M'} \mathcal{L}_t(M') - \beta. \quad (29)$$

We now state the structural condition and theoretical guarantee for the OMLE algorithm.

**Definition E.4** (Maximum likelihood estimation coefficient). *The maximum likelihood estimation coefficient (MLEC) of model class $\mathcal{M}$ with respect to reference model $\overline{M} \in \mathcal{M}$, parameter $\gamma > 0$, and length $K \in \mathbb{Z}_{\geqslant 1}$ is defined as*

$$\mathrm{mlec}_{\gamma,K}(\mathcal{M}, \overline{M}) := \sup_{\{M^k\} \in \mathcal{M}} \frac{1}{K} \sum_{k=1}^{K} \left[ f^{M^k}(\pi_{M^k}) - f^{\overline{M}}(\pi_{M^k}) \right]$$

$$- \frac{\gamma}{K}\left[ \left(\max_{k \in [K]} \sum_{t \leqslant k-1} D_{\mathrm{RL}}^2(\overline{M}(\pi_{M^t}), M^k(\pi_{M^t})) \right) \vee 1 \right].$$

---

[11]Our version is equivalent to Agarwal & Zhang (2022a, Algorithm 1) except that we look at the full observation and reward vector (of all layers), whereas they only look at a random layer $h^t \sim \mathrm{Unif}([H])$.

---

**Algorithm 5** MOPS (Agarwal & Zhang, 2022a)

---

1: **Input:** Parameters $\eta, \gamma > 0$; prior distribution $\mu^1 \in \Delta(\mathcal{M})$; optimistic likelihood function $\widetilde{\mathbb{P}}$.
2: **for** $t = 1, \ldots, T$ **do**
3:     Sample $M^t \sim \mu^t(\cdot)$ and set $\pi^t = \pi_{M^t}$.
4:     Execute $\pi^t$ and observe $(o^t, r^t)$.
5:     Update posterior of models by Optimistic Posterior Sampling (OPS):

$$\mu^{t+1}(M) \propto_M \mu^t(M) \cdot \exp\left( \gamma^{-1} f^M(\pi_M) + \eta_{\mathrm{p}} \log \widetilde{\mathbb{P}}^{M,\pi^t}(o^t) - \eta_{\mathrm{r}} \left\| \mathbf{r}^t - \mathbf{R}^M(o^t) \right\|_2^2 \right). \quad (30)$$

---

**Theorem E.5** (Regret bound for OMLE). *Choosing $\beta = 3\log(|\mathcal{M}|/\delta) \geqslant 1$, with probability at least $1 - \delta$, Algorithm 6 achieves*

$$\mathbf{Reg_{DM}} \leqslant \inf_{\gamma > 0} \{ T \cdot \mathrm{mlec}_{\gamma, T}(\mathcal{M}, M^\star) + 12\gamma \cdot \log(|\mathcal{M}|/\delta) \}.$$

Existing sample-efficiency guarantees for OMLE-type algorithms are only established on specific RL problems through case-by-case analyses (Mete et al., 2021; Uehara et al., 2021; Liu et al., 2022a;b). By contrast, Theorem E.5 shows that OMLE works on any problem with bounded MLEC, thereby offering a more unified understanding. The proof of Theorem E.5 is deferred to Appendix G.2.

We remark that the MLEC is also closely related to the PSC, in that bounded MLEC (under a slightly modified definition) implies bounded PSC (Proposition G.4 & Appendix G.3).

## F    PROOFS FOR SECTION E.1

### F.1    ALGORITHM MOPS

Here we present a more general version of the MOPS algorithm where we allow $\mathcal{M}$ to be a possibly infinite model class, and require a prior $\mu^1 \in \Delta(\mathcal{M})$ and an optimistic likelihood function $\widetilde{\mathbb{P}}$ (cf. Definition D.5) as inputs. The algorithm stated in Appendix E.1 is a special case of Algorithm 5 with $|\mathcal{M}| < \infty$, $\widetilde{\mathbb{P}} = \mathbb{P}$, and $\mu^1 = \mathrm{Unif}(\mathcal{M})$.

We state the theoretical guarantee for Algorithm 5 as follows.

**Theorem F.1** (MOPS). *Given a $\rho$-optimistic cover $(\widetilde{\mathbb{P}}, \mathcal{M}_0)$, Algorithm 5 with $\eta_{\mathrm{p}} = 1/6$, $\eta_{\mathrm{r}} = 0.6$ and $\mu^1 = \mathrm{Unif}(\mathcal{M}_0)$ achieves the following with probability at least $1 - \delta$:*

$$\mathbf{Reg_{DM}} \leqslant T \left[ \mathrm{psc}_{\gamma/6}(\mathcal{M}, M^\star) + \frac{2}{\gamma} \right] + \gamma[\log|\mathcal{M}_0| + 3T\rho + 2\log(2/\delta)].$$

*Choosing the optimal $\gamma > 0$, with probability at least $1 - \delta$, suitable implementation of Algorithm 5 achieves*

$$\mathbf{Reg_{DM}} \leqslant 12 \inf_{\gamma > 0} \left\{ T \, \mathrm{psc}_\gamma(\mathcal{M}, M^\star) + \frac{T}{\gamma} + \gamma[\mathrm{est}(\mathcal{M}, T) + \log(1/\delta)] \right\}.$$

When $\mathcal{M}$ is finite, clearly $(\mathbb{P}, \mathcal{M})$ itself is a 0-optimistic covering, and hence Theorem F.1 implies Theorem E.2 directly.

It is worth noting that Agarwal & Zhang (2022a) state the guarantee of MOPS in terms of a general prior, with the regret depending on a certain "prior around true model" like quantity. The proof of Theorem F.2 can be directly adapted to work in their setting; however, we remark that, obtaining an explicit upper bound on their "prior around true model" in a concrete problem likely requires constructing an explicit covering, similar as in Theorem F.1.

**Proof of Theorem F.1**    By definition,

$$\mathbf{Reg_{DM}} = \sum_{t=1}^T \mathbb{E}_{M \sim \mu^t} \left[ f^{M^\star}(\pi_{M^\star}) - f^{M^\star}(\pi_M) \right]$$

$$= \sum_{t=1}^{T} \underbrace{\mathbb{E}_{M \sim \mu^t} \left[ f^{M^\star}(\pi_{M^\star}) - f^M(\pi_M) + \frac{\gamma}{6} \mathbb{E}_{\pi \sim p^t} \left[ D_{\mathrm{RL}}^2 \left( M^\star(\pi), M(\pi) \right) \right] \right]}_{\text{Bounded by Corollary F.3}}$$

$$+ \sum_{t=1}^{T} \underbrace{\mathbb{E}_{M \sim \mu^t} \left[ f^M(\pi_M) - f^{M^\star}(\pi_M) - \frac{\gamma}{6} \mathbb{E}_{\pi \sim p^t} \left[ D_{\mathrm{RL}}^2 \left( M^\star(\pi), M(\pi) \right) \right] \right]}_{\text{Bounded by psc}}$$

$$\leqslant \gamma[\log|\mathcal{M}_0| + 3T\rho + 2\log(2/\delta)] + \frac{2T}{\gamma} + T \operatorname{psc}_{\gamma/6}(\mathcal{M}, M^\star).$$

$\square$

## F.2 OPTIMISTIC POSTERIOR SAMPLING

In this section, we analyze the following Optimistic Posterior Sampling algorithm under a more general setting. The problem setting and notation are the same as the online model estimation problem introduced in Appendix D.1. Additionally, we assume that each $M \in \mathcal{M}$ is assigned with a scalar $V_M \in [0, 1]$; in our application, $V_M$ is going to be optimal value of model $M$.

**Theorem F.2** (Analysis of posterior in OPS). *Fix a $\rho > 0$ and a $\rho$-optimistic covering $(\widetilde{\mathbb{P}}, \mathcal{M}_0)$ of $\mathcal{M}$. Under the assumption of Theorem D.1, the following update rule*

$$\mu^{t+1}(M) \propto_M \mu^t(M) \cdot \exp \left( \gamma^{-1} V_M + \eta_{\mathrm{p}} \log \widetilde{\mathbb{P}}^M(o^t | \pi^t) - \eta_{\mathrm{r}} \left\| \mathbf{r}^t - \mathbf{R}^M(o^t) \right\|_2^2 \right). \quad (31)$$

*with $2\eta_{\mathrm{p}} + 4\sigma^2 \eta_{\mathrm{r}} < 1$ and $\mu^1 = \mathrm{Unif}(\mathcal{M}_0)$ achieves with probability at least $1 - \delta$ that*

$$\sum_{t=1}^{T} \mathbb{E}_{M \sim \mu^t} \left[ V_\star - V_M + c_0 \gamma \operatorname{Err}_M^t \right] \leqslant \frac{T}{8\gamma(1 - 2\eta_{\mathrm{p}} - 4\sigma^2 \eta_{\mathrm{r}})} + \gamma \log|\mathcal{M}_0|$$
$$+ \gamma \left[ T\rho(2\gamma^{-1} + 2\eta_{\mathrm{p}} + \eta_{\mathrm{r}}) + 2\log(2/\delta) \right],$$

*where $V_\star = V_{M^\star}$ and $c_0 = \min\{\eta_{\mathrm{p}}, 4\sigma^2 \eta_{\mathrm{r}} (1 - e^{-D^2/8\sigma^2})/D^2\}$, as long as there exists $M \in \mathcal{M}_0$ such that $M^\star$ is covered by $M$ (cf. Definition D.5) and $V_M \geqslant V_\star - 2\rho$.*

The proof of Theorem F.2 can be found in Appendix F.2.1.

As a direct corollary of Theorem F.2, the posterior $\mu^t$ maintained in the MOPS algorithm (Algorithm 5) achieves the following guarantee.

**Corollary F.3.** *Given a $\rho$-optimistic covering $(\widetilde{\mathbb{P}}, \mathcal{M}_0)$, subroutine (30) within Algorithm 5 with $\eta_{\mathrm{p}} = 1/6, \eta_{\mathrm{r}} = 0.6, \gamma \geqslant 1$ and uniform prior $\mu^1 = \mathrm{Unif}(\mathcal{M}_0)$ achieves with probability at least $1 - \delta$ that*

$$\sum_{t=1}^{T} \mathbb{E}_{M \sim \mu^t} \left[ f^{M^\star}(\pi_{M^\star}) - f^M(\pi_M) + \frac{\gamma}{6} \mathbb{E}_{\pi \sim p^t} \left[ D_{\mathrm{RL}}^2 \left( M^\star(\pi), M(\pi) \right) \right] \right]$$
$$\leqslant \frac{2T}{\gamma} + \gamma[\log|\mathcal{M}_0| + 3T\rho + 2\log(2/\delta)].$$

*Proof.* Note that subroutine (30) in Algorithm 5 is exactly an instantiation of (31) with context $\pi^t$ sampled from distribution $p^t$ (which depends on $\mu^t$), observation $o^t$, reward $\mathbf{r}^t$, and $V_M = f^M(\pi_M)$. Furthermore, $\mathbb{E}_{M \sim \mu^t} \left[ \operatorname{Err}_M^t \right]$ corresponds to $\mathbb{E}_{\widehat{M}^t \sim \mu^t} \mathbb{E}_{\pi^t \sim p^t} \left[ D_{\mathrm{RL}}^2 (M^\star(\pi^t), \widehat{M}^t(\pi^t)) \right]$ (cf. Corollary D.2).

Therefore, in order to apply Theorem F.2, we have to verify: as long as $M \in \mathcal{M}_0$ covers the ground truth model $M^\star$ (i.e. $\left\| \mathbf{R}^{M_0}(o) - \mathbf{R}^\star(o) \right\|_1 \leqslant \rho$ and $\widetilde{\mathbb{P}}^M(\cdot|\pi) \geqslant \mathbb{P}^\star(\cdot|\pi)$ for all $\pi$), it holds that $V_M \geqslant V_\star - 2\rho$. We note that $V_\star \geqslant f^{M^\star}(\pi_M)$, thus

$$V_\star - V_M \leqslant \sup_\pi \left| f^M(\pi) - f^{M^\star}(\pi) \right| \leqslant \sup_\pi D_{\mathrm{TV}} \left( \mathbb{P}^\star(\cdot|\pi), \mathbb{P}^M(\cdot|\pi) \right) + \rho \quad (32)$$

by definition. An important observation is that, for $\pi \in \Pi$,

$$D_{\mathrm{TV}}\left(\mathbb{P}^\star(\cdot|\pi), \mathbb{P}^M(\cdot|\pi)\right) = \sum_{o\in\mathcal{O}}\left[\mathbb{P}^\star(o|\pi) - \mathbb{P}^M(o|\pi)\right]_+ \leqslant \sum_{o\in\mathcal{O}}\widetilde{\mathbb{P}}^M(o|\pi) - \mathbb{P}^M(o|\pi) \leqslant \rho^2. \quad (33)$$

Therefore, $V_\star - V_M \leqslant \rho + \rho^2 \leqslant 2\rho$. Now we can apply Theorem F.2 and plug in $\sigma^2 = 1/4$, $D^2 = 2$ as in Corollary D.2. Choosing $\eta_{\mathrm{p}} = 1/6$, $\eta_{\mathrm{r}} = 0.6$, and $\gamma \geqslant 1$, we have $4\eta_{\mathrm{r}} + \gamma^{-1} + \eta_{\mathrm{p}} \leqslant 3$, $8(1 - 2\eta_{\mathrm{p}} - \eta_{\mathrm{r}}) \geqslant 1/2$, and $c_0 = 1/6$. This completes the proof. $\qquad\square$

### F.2.1 PROOF OF THEOREM F.2

For all $t \in [T]$ define the random variable

$$\Delta^t := -\log\mathbb{E}_{M\sim\mu^t}\left[\exp\left(\gamma^{-1}(V_M - V_\star) + \eta_{\mathrm{p}}\log\frac{\widetilde{\mathbb{P}}^M(o^t|\pi^t)}{\mathbb{P}^\star(o^t|\pi^t)} + \eta_{\mathrm{r}}\delta_M^t\right)\right],$$

where $\delta_M^t$ is defined as in (11).

Similar as the proof of Theorem D.1, we begin by noticing that

$$\log\mathbb{E}_t\left[\exp\left(-\Delta^t\right)\right] = \log\mathbb{E}_{M\sim\mu^t}\mathbb{E}_t\left[\exp\left(\gamma^{-1}(V_M - V_\star) + \eta_{\mathrm{p}}\log\frac{\widetilde{\mathbb{P}}^M(o^t|\pi^t)}{\mathbb{P}^\star(o^t|\pi^t)} + \eta_{\mathrm{r}}\delta_M^t\right)\right]$$

$$\leqslant (1 - 2\eta_{\mathrm{p}} - 4\sigma^2\eta_{\mathrm{r}})\log\mathbb{E}_{M\sim\mu^t}\left[\exp\left(\frac{V_M - V_\star}{\gamma(1 - 2\eta_{\mathrm{p}} - 4\sigma^2\eta_{\mathrm{r}})}\right)\right]$$

$$+ 2\eta_{\mathrm{p}}\log\mathbb{E}_{M\sim\mu^t}\mathbb{E}_t\left[\exp\left(\frac{1}{2}\log\frac{\widetilde{\mathbb{P}}^M(o^t|\pi^t)}{\mathbb{P}^\star(o^t|\pi^t)}\right)\right]$$

$$+ 4\sigma^2\eta_{\mathrm{r}}\log\mathbb{E}_{M\sim\mu^t}\mathbb{E}_t\left[\exp\left(\frac{1}{4\sigma^2}\delta_M^t\right)\right], \quad (34)$$

which is due to Jensen's inequality. For the first term, we abbreviate $\eta_0 = 1 - 2\eta_{\mathrm{p}} - 4\sigma^2\eta_{\mathrm{r}}$ and consider $a_M := (V_\star - V_M)/\gamma\eta_0$. Then by the boundedness of $a_M$ and Hoeffding's Lemma,

$$\mathbb{E}_{M\sim\mu^t}[\exp(-a_M)] \leqslant \exp\left(\frac{\mathbb{E}_{M\sim\mu^t}[V_M] - V_\star}{\gamma\eta_0}\right)\cdot\exp\left(\frac{1}{8\gamma^2\eta_0^2}\right). \quad (35)$$

The second term can be bounded as in (22):

$$\log\mathbb{E}_{M\sim\mu^t}\mathbb{E}_t\left[\exp\left(\frac{1}{2}\log\frac{\widetilde{\mathbb{P}}^M(o^t|\pi^t)}{\mathbb{P}^\star(o^t|\pi^t)}\right)\right] \leqslant \log\mathbb{E}_{M\sim\mu^t}\left[1 - \frac{1}{2}\mathbb{E}_t\left[D_{\mathrm{H}}^2(\mathbb{P}^M(\cdot|\pi^t), \mathbb{P}^\star(\cdot|\pi^t))\right] + \rho\right]$$

$$\leqslant -\frac{1}{2}\mathbb{E}_{M\sim\mu^t}\mathbb{E}_t\left[D_{\mathrm{H}}^2(\mathbb{P}^M(\cdot|\pi^t), \mathbb{P}^\star(\cdot|\pi^t))\right] + \rho, \quad (36)$$

and the third term can be bounded by Lemma D.3 (similar to (14)):

$$\log\mathbb{E}_{M\sim\mu^t}\mathbb{E}_t\left[\exp\left(\frac{1}{4\sigma^2}\delta_M^t\right)\right] \leqslant \log\mathbb{E}_{M\sim\mu^t}\mathbb{E}_t\left[\exp\left(\frac{1}{8\sigma^2}\left\|\mathbf{R}^M(o^t) - \mathbf{R}^\star(o^t)\right\|_2^2\right)\right]$$

$$\leqslant \log\mathbb{E}_{M\sim\mu^t}\mathbb{E}_t\left[1 - (1 - e^{-D^2/8\sigma^2})/D^2\left\|\mathbf{R}^M(o^t) - \mathbf{R}^\star(o^t)\right\|_2^2\right] \quad (37)$$

$$\leqslant -(1 - e^{-D^2/8\sigma^2})/D^2\mathbb{E}_{M\sim\mu^t}\mathbb{E}_t\left[\left\|\mathbf{R}^M(o^t) - \mathbf{R}^\star(o^t)\right\|_2^2\right].$$

Plugging (35), (36), and (37) into (34) gives

$$-\log\mathbb{E}_t\left[\exp\left(-\Delta^t\right)\right] \geqslant \frac{V_\star - \mathbb{E}_{M\sim\mu^t}[V_M]}{\gamma} - \frac{1}{8\gamma^2\eta_0}$$

$$+ 2\eta_{\mathrm{p}}\left[\frac{1}{2}\mathbb{E}_{M\sim\mu^t}\mathbb{E}_t\left[D_{\mathrm{H}}^2(\mathbb{P}^M(\cdot|\pi^t), \mathbb{P}^\star(\cdot|\pi^t))\right] - \rho\right]$$

$$+ 4\sigma^2\eta_{\mathrm{r}}\left(1 - e^{-D^2/8\sigma^2}\right)/D^2\cdot\mathbb{E}_{M\sim\mu^t}\mathbb{E}_t\left[\left\|\mathbf{R}^M(o^t) - \mathbf{R}^\star(o^t)\right\|_2^2\right] \quad (38)$$

$$\geqslant \mathbb{E}_{M\sim\mu^t}\left[\gamma^{-1}(V_\star - V_M) + c_0\,\mathrm{Err}_M^t\right] - \frac{1}{8\gamma^2\eta_0} - 2\eta_{\mathrm{p}}\rho.$$

On the other hand, by Lemma B.2, we have with probability at least $1 - \delta/2$ that

$$\sum_{t=1}^{T} \Delta^t + \log(2/\delta) \geqslant \sum_{t=1}^{T} - \log \mathbb{E}_t \big[ \exp\left( -\Delta^t \right) \big]. \tag{39}$$

It remains to bound $\sum_{t=1}^{T} \Delta^t$. By the update rule (31) and a telescoping argument similar to (16), we have

$$\sum_{t=1}^{T} \Delta^t = - \log \mathbb{E}_{M \sim \mu^1} \exp \left( \sum_{t=1}^{T} \gamma^{-1} (V_M - V_\star) + \eta_{\mathrm{p}} \log \frac{\widetilde{\mathbb{P}}^M(o^t|\pi^t)}{\mathbb{P}^\star(o^t|\pi^t)} + \eta_{\mathrm{r}} \delta_M^t \right).$$

The following argument is almost the same as the argument we make to bound (24). Fix a $M \in \mathcal{M}_0$ that covers $M^\star$ and $V_M - V_\star \geqslant -2\rho$. We bound the following moment generating function

$$\mathbb{E} \left[ \exp \left( \sum_{t=1}^{T} \Delta^t \right) \right] = \mathbb{E} \left[ \frac{1}{\mathbb{E}_{M \sim \mu^1} \exp \left( \sum_{t=1}^{T} \gamma^{-1}(V_M - V_\star) + \eta_{\mathrm{p}} \log \frac{\widetilde{\mathbb{P}}_M(o^t|\pi^t)}{\mathbb{P}^\star(o^t|\pi^t)} + \eta_{\mathrm{r}} \delta_M^t \right)} \right]$$

$$\leqslant |\mathcal{M}_0| \, \mathbb{E} \left[ \exp \left( - \sum_{t=1}^{T} \gamma^{-1}(V_M - V_\star) + \eta_{\mathrm{p}} \log \frac{\widetilde{\mathbb{P}}_M(o^t|\pi^t)}{\mathbb{P}^\star(o^t|\pi^t)} + \eta_{\mathrm{r}} \delta_M^t \right) \right]$$

$$\leqslant \exp \left( 2T\gamma^{-1}\rho \right) |\mathcal{M}_0| \, \mathbb{E} \left[ \prod_{t=1}^{T} \exp \left( -\eta_{\mathrm{r}} \delta_M^t \right) \right]$$

$$\leqslant \exp \left( 2T\gamma^{-1}\rho + T\rho\eta_{\mathrm{r}} \right) |\mathcal{M}_0|,$$

where the first inequality is because $\mu^1(M) = 1/|\mathcal{M}_0|$, the second inequality is due to $\widetilde{\mathbb{P}}^M \geqslant \mathbb{P}^\star$, and the last inequality follows from the same argument as (26): by Lemma D.3 we have $\mathbb{E} \big[ \exp(-\eta_{\mathrm{r}} \delta_M^t) \, | \, o^t \big] \leqslant \exp(\eta_{\mathrm{r}}\rho)$, and applying this inequality recursively yields the desired result.

Therefore, with at least with probability $1 - \delta/2$,

$$\sum_{t=1}^{T} \Delta^t \leqslant \log |\mathcal{M}_0| + T\rho \big( 2\gamma^{-1} + 2\eta_{\mathrm{r}} \big) + \log(2/\delta). \tag{40}$$

Summing (38) over $t \in [T]$, then taking union of (40) and (39) establish the theorem. $\qquad \square$

### F.3 Bounding PSC by Hellinger decoupling coefficient

The Hellinger decoupling coefficient is introduced by Agarwal & Zhang (2022a) as a structural condition for the sample efficiency of the MOPS algorithm.

**Definition F.4** (Hellinger decoupling coefficient)**.** *Given* $\alpha \in (0,1)$, $\varepsilon \geqslant 0$, *the coefficient* $\mathrm{dcp}^{h,\alpha}(\mathcal{M}, \overline{M}, \varepsilon)$ *is the smallest positive number* $c^h \geqslant 0$ *such that for all* $\mu \in \Delta(\mathcal{M})$,

$$\mathbb{E}_{M \sim \mu} \mathbb{E}^{\overline{M}, \pi_M} \Big[ Q_h^{M, \pi_M}(s_h, a_h) - r_h - V_{h+1}^{M, \pi_M}(s_{h+1}) \Big]$$

$$\leqslant \left( c^h \, \mathbb{E}_{M, M' \sim \mu} \mathbb{E}^{\overline{M}, \pi_{M'}} \Big[ D_{\mathrm{H}}^2 \Big( \mathbb{P}_h^M(\cdot|s_h, a_h), \mathbb{P}_h^{\overline{M}}(\cdot|s_h, a_h) \Big) + \Big| R_h^M(s_h, a_h) - R_h^{\overline{M}}(s_h, a_h) \Big|^2 \Big] \right)^{\alpha} + \varepsilon.$$

*The Hellinger decoupling coefficient* $\mathrm{dcp}$ *is defined as*

$$\mathrm{dcp}^{\alpha}(\mathcal{M}, \overline{M}, \varepsilon) := \left( \frac{1}{H} \sum_{h=1}^{H} \mathrm{dcp}^{h,\alpha}(\mathcal{M}, \overline{M}, \varepsilon)^{\alpha/(1-\alpha)} \right)^{(1-\alpha)/\alpha}.$$

We remark that the main difference between the PSC and the Hellinger decoupling coefficient is that, the Hellinger decoupling coefficient is defined in terms of Bellman errors and Hellinger distances within each layer $h \in [H]$ separately, whereas the PSC is defined in terms of the overall value function and Hellinger distances of the entire observable $(o, \mathbf{r})$.

The following result shows that the PSC can be upper bounded by the Hellinger decoupling coefficient, and thus is a slightly more general definition[12].

**Proposition F.5** (Bounding PSC by Hellinger decoupling coefficient). *For any $\alpha \in (0, 1)$, we have*

$$\mathrm{psc}_\gamma(\mathcal{M}, \overline{M}) \leqslant H \inf_{\varepsilon \geqslant 0} \left( (1-\alpha) \left( \frac{2\alpha H \, \mathrm{dcp}^\alpha(\mathcal{M}, \overline{M}, \varepsilon)}{\gamma} \right)^{\alpha/(1-\alpha)} + \varepsilon \right).$$

*Proof.* Fix $\mathcal{M}, \overline{M} \in \mathcal{M}$, and $\alpha \in (0, 1)$. Consider

$$\Delta_{h, s_h, a_h}(M, \overline{M}) := D_{\mathrm{H}}^2 \left( \mathbb{P}_h^M(\cdot | s_h, a_h), \mathbb{P}_h^{\overline{M}}(\cdot | s_h, a_h) \right) + \left| R_h^M(s_h, a_h) - R_h^{\overline{M}}(s_h, a_h) \right|^2.$$

By the definition of $D_{\mathrm{RL}}$ and Lemma B.4, we have for any $h \in [H]$ that

$$\mathbb{E}^{\overline{M}, \pi_{M'}} \left[ \Delta_{h, s_h, a_h}(M, \overline{M}) \right] \leqslant 2D_{\mathrm{RL}}^2 \left( \overline{M}(\pi_{M'}), M(\pi_{M'}) \right).$$

Fix $\varepsilon \geqslant 0$ and and write $c^h = \mathrm{dcp}^{h, \alpha}(\mathcal{M}, \overline{M}, \varepsilon)$. For any $\mu \in \Delta(\mathcal{M})$, we have

$$\mathbb{E}_{M \sim \mu} \left[ f^M(\pi_M) - f^{\overline{M}}(\pi_M) \right]$$

$$= \sum_{h=1}^H \mathbb{E}_{M \sim \mu} \mathbb{E}^{\overline{M}, \pi_M} \left[ Q_h^{M, \pi_M}(s_h, a_h) - r_h - V_{h+1}^{M, \pi_M}(s_{h+1}) \right]$$

$$\leqslant \sum_{h=1}^H \left( c^h \, \mathbb{E}_{M, M' \sim \mu} \mathbb{E}^{\overline{M}, \pi_{M'}} \left[ \Delta_{h, s_h, a_h}(M, \overline{M}) \right] \right)^\alpha + H\varepsilon$$

$$\leqslant \sum_{h=1}^H \left( c^h \right)^\alpha \left( 2 \mathbb{E}_{M, M' \sim \mu} \left[ D_{\mathrm{RL}}^2 \left( \overline{M}(\pi_{M'}), M(\pi_{M'}) \right) \right] \right)^\alpha + H\varepsilon$$

$$\leqslant \gamma \mathbb{E}_{M, M' \sim \mu} \left[ D_{\mathrm{RL}}^2 \left( \overline{M}(\pi_{M'}), M(\pi_{M'}) \right) \right] + (1-\alpha) \left( \frac{2\alpha H}{\gamma} \right)^{\alpha/(1-\alpha)} \sum_{h=1}^H \left( c^h \right)^{\alpha/(1-\alpha)} + H\varepsilon$$

$$= \gamma \mathbb{E}_{M, M' \sim \mu} \left[ D_{\mathrm{RL}}^2 \left( \overline{M}(\pi_{M'}), M(\pi_{M'}) \right) \right] + (1-\alpha) H^{1/(1-\alpha)} \left( \frac{2\alpha \, \mathrm{dcp}^\alpha(\mathcal{M}, \overline{M}, \varepsilon)}{\gamma} \right)^{\alpha/(1-\alpha)} + H\varepsilon,$$

where the last inequality is due to the fact that for all $x, y \geqslant 0$, $\alpha \in (0, 1)$,

$$x^\alpha y^\alpha \leqslant \alpha \cdot \frac{\gamma x}{\alpha H} + (1-\alpha) \left( \frac{\alpha H y}{\gamma} \right)^{\alpha/(1-\alpha)} = \frac{\gamma x}{H} + (1-\alpha) \left( \frac{\alpha H}{\gamma} \right)^{\alpha/(1-\alpha)} \cdot y^{\alpha/(1-\alpha)},$$

by weighted AM-GM inequality. $\square$

## F.4 PROOF OF PROPOSITION E.3

Our overall argument is to bound the DEC by strong duality and the probability matching argument (similar as Foster et al. (2021, Section 4.2)), after which we show that the resulting quantity is related nicely to the PSC.

In the following, we denote $\mathrm{psc}_\gamma(\mathcal{M}) := \sup_{\overline{M} \in \mathcal{M}} \mathrm{psc}_\gamma(\mathcal{M}, \overline{M})$. By definition, it suffices to bound $\mathrm{dec}_\gamma(\mathcal{M}, \overline{\mu})$ for any fixed $\overline{\mu} \in \Delta(\mathcal{M})$. We have

$$\mathrm{dec}(\mathcal{M}, \overline{\mu}) = \inf_{p \in \Delta(\Pi)} \sup_{M \in \mathcal{M}} \mathbb{E}_{\overline{M} \sim \overline{\mu}} \mathbb{E}_{\pi \sim p} \left[ f^M(\pi_M) - f^M(\pi) - \gamma D_{\mathrm{RL}}^2(M(\pi), \overline{M}(\pi)) \right]$$

---

[12]We remark that Agarwal & Zhang (2022a) defines the Hellinger decoupling coefficient in terms of a general function $\pi_{\mathrm{gen}}(h, \mu)$ that maps a $\mu \in \Delta(\mathcal{M})$ to a $p \in \Delta(\Pi)$, and we only present the version that $\pi_{\mathrm{gen}}(h, \mu)$ follows a sample from $\mu$, i.e. $\pi_{\mathrm{gen}}(h, \mu) \overset{d}{=} \pi_M, M \sim \mu$. For the general case that $\pi_{\mathrm{gen}}(h, \mu) \overset{d}{=} \pi_{M, h}^{\mathrm{est}}, M \sim \mu$, the corresponding Hellinger decoupling coefficient can be bounded by $\mathrm{psc}^{\mathrm{est}}$ (Definition K.23), by a same argument.

---

**Algorithm 6** OMLE

---

1: **Input:** Parameter $\beta > 0$.
2: Initialize confidence set $\mathcal{M}^1 = \mathcal{M}$.
3: **for** $t = 1, \ldots, T$ **do**
4:     Compute $(M^t, \pi^t) = \arg\max_{M \in \mathcal{M}^t, \pi \in \Pi} f^M(\pi)$.
5:     Execute $\pi^t$ and observe $\tau^t = (o^t, \mathbf{r}^t)$.
6:     Update confidence set with (28):

$$\mathcal{M}^{t+1} := \left\{ M \in \mathcal{M} : \mathcal{L}_{t+1}(M) \geqslant \max_{M' \in \mathcal{M}} \mathcal{L}_{t+1}(M') - \beta \right\}.$$

---

$$= \inf_{p \in \Delta(\Pi)} \sup_{\mu \in \Delta(\mathcal{M})} \mathbb{E}_{M \sim \mu, \overline{M} \sim \overline{\mu}} \mathbb{E}_{\pi \sim p} \big[ f^M(\pi_M) - f^M(\pi) - \gamma D_{\mathrm{RL}}^2(M(\pi), \overline{M}(\pi)) \big]$$

$$= \sup_{\mu \in \Delta(\mathcal{M})} \inf_{p \in \Delta(\Pi)} \mathbb{E}_{M \sim \mu, \overline{M} \sim \overline{\mu}} \mathbb{E}_{\pi \sim p} \big[ f^M(\pi_M) - f^M(\pi) - \gamma D_{\mathrm{RL}}^2(M(\pi), \overline{M}(\pi)) \big],$$

where the last equality follows by strong duality (Theorem B.1).

Now, fix any $\mu \in \Delta(\mathcal{M})$, we pick $p \in \Delta(\Pi)$ by probability matching: $\pi \sim p$ is equal in distribution to $\pi = \pi_{M'}$ where $M' \sim \mu$ is an independent copy of $M$. For this choice of $p$, the quantity inside the sup-inf above is

$$\mathbb{E}_{M \sim \mu, M' \sim \mu} \mathbb{E}_{\overline{M} \sim \overline{\mu}} \big[ f^M(\pi_M) - f^M(\pi_{M'}) - \gamma D_{\mathrm{RL}}^2(M(\pi_{M'}), \overline{M}(\pi_{M'})) \big]$$

$$= \mathbb{E}_{M \sim \mu, M' \sim \mu} \mathbb{E}_{\overline{M} \sim \overline{\mu}} \left[ f^M(\pi_M) - f^{\overline{M}}(\pi_{M'}) - \frac{5\gamma}{6} D_{\mathrm{RL}}^2(M(\pi_{M'}), \overline{M}(\pi_{M'})) \right]$$

$$\quad + \mathbb{E}_{M \sim \mu} \mathbb{E}_{\overline{M} \sim \overline{\mu}} \left[ f^{\overline{M}}(\pi_{M'}) - f^M(\pi_{M'}) - \frac{\gamma}{6} D_{\mathrm{RL}}^2(M(\pi_{M'}), \overline{M}(\pi_{M'})) \right]$$

$$\overset{(i)}{=} \mathbb{E}_{M \sim \mu, M' \sim \mu} \mathbb{E}_{\overline{M} \sim \overline{\mu}} \left[ f^M(\pi_M) - f^{\overline{M}}(\pi_M) - \frac{5\gamma}{6} D_{\mathrm{RL}}^2(M(\pi_{M'}), \overline{M}(\pi_{M'})) \right]$$

$$\quad + \mathbb{E}_{M \sim \mu} \mathbb{E}_{\overline{M} \sim \overline{\mu}} \left[ f^{\overline{M}}(\pi_{M'}) - f^M(\pi_{M'}) - \frac{\gamma}{6} D_{\mathrm{RL}}^2(M(\pi_{M'}), \overline{M}(\pi_{M'})) \right]$$

$$\overset{(ii)}{\leqslant} \underbrace{\mathbb{E}_{M \sim \mu, M' \sim \mu} \mathbb{E}_{\overline{M} \sim \overline{\mu}} \left[ f^M(\pi_M) - f^{\overline{M}}(\pi_M) - \frac{\gamma}{6} D_{\mathrm{RL}}^2(\overline{M}(\pi_{M'}), M(\pi_{M'})) \right]}_{\leqslant \mathbb{E}_{\overline{M} \sim \overline{\mu}}[\mathrm{psc}_{\gamma/6}(\mathcal{M}, \overline{M})] \leqslant \mathrm{psc}_{\gamma/6}(\mathcal{M})}$$

$$\quad + \mathbb{E}_{M \sim \mu, M' \sim \mu} \mathbb{E}_{\overline{M} \sim \overline{\mu}} \left[ \sqrt{H+1} D_{\mathrm{RL}}(M(\pi_{M'}), \overline{M}(\pi_{M'})) - \frac{\gamma}{6} D_{\mathrm{RL}}^2(M(\pi_{M'}), \overline{M}(\pi_{M'})) \right]$$

$$\overset{(iii)}{\leqslant} \mathrm{psc}_{\gamma/6}(\mathcal{M}) + \frac{2(H+1)}{\gamma}.$$

Above, (i) uses the fact that $f^{\overline{M}}(\pi_{M'})$ is equal in distribution to $f^{\overline{M}}(\pi_M)$ (since $M \sim \mu$ and $M' \sim \mu$); (ii) uses Lemma B.7 and the fact that $5 D_{\mathrm{RL}}^2(M(\pi_{M'}), \overline{M}(\pi_{M'})) \geqslant D_{\mathrm{RL}}^2(\overline{M}(\pi_{M'}), M(\pi_{M'}))$ that is due to Lemma B.6; (iii) uses the inequality $\sqrt{H+1} x \leqslant \frac{\gamma}{6} x^2 + \frac{3(H+1)}{2\gamma}$ for any $x \in \mathbb{R}$. Finally, by the arbitrariness of $\overline{\mu} \in \Delta(\Pi)$, we have shown that $\overline{\mathrm{dec}}_\gamma(\mathcal{M}) \leqslant \mathrm{psc}_{\gamma/6}(\mathcal{M}) + 2(H+1)/\gamma$. This is the desired result. $\qquad \square$

## G  PROOFS FOR SECTION E.2

### G.1  ALGORITHM OMLE

In this section, we present the Algorithm OMLE (Algorithm 6), and then state the basic guarantees of its confidence sets, as follows.

**Theorem G.1** (Guarantee of MLE)**.** *By choosing* $\beta \geqslant 3\,\mathrm{est}(\mathcal{M}, 2T) + 3\log(1/\delta)$, *Algorithm* 6 *achieves the following with probability at least* $1 - \delta$: *for all* $t \in [T]$, $M^\star \in \mathcal{M}^t$, *and it holds that*

$$\sum_{s<t} D_{\mathrm{RL}}^2(M^\star(\pi^s), M(\pi^s)) \leqslant 2\beta + 6\,\mathrm{est}(\mathcal{M}, 2T) + 6\log(1/\delta) \leqslant 4\beta, \qquad \forall M \in \mathcal{M}^t.$$

**Proof of Theorem G.1**     The proof of Theorem G.1 is mainly based on the following lemma.

**Lemma G.2.** *Fix a* $\rho > 0$. *With probability at least* $1 - \delta$, *it holds that for all* $t \in [T]$ *and* $M \in \mathcal{M}$,

$$\sum_{s<t} D_{\mathrm{RL}}^2\left(M^\star(\pi^s), M(\pi^s)\right) \leqslant 2(\mathcal{L}_t(M^\star) - \mathcal{L}_t(M)) + 6\log\frac{\mathcal{N}(\mathcal{M}, \rho)}{\delta} + 8T\rho.$$

Now, we can take $\rho$ that attains $\mathrm{est}(\mathcal{M}, 2T)$ and apply Lemma G.2. Conditional on the success of Lemma G.2, it holds that for all $t \in [T]$ and $M \in \mathcal{M}$,

$$\mathcal{L}_t(M) - \mathcal{L}_t(M^\star) \leqslant 3\,\mathrm{est}(\mathcal{M}, 2T) + 3\log(1/\delta).$$

Therefore, our choice of $\beta$ is enough to ensure that $M^\star \in \mathcal{M}^t$. Then, for $M \in \mathcal{M}^t$, we have

$$\mathcal{L}_t(M) \geqslant \max_{M'\in\mathcal{M}} \mathcal{L}_t(M') - \beta \geqslant \mathcal{L}_t(M^\star) - \beta.$$

Applying Lemma G.2 again completes the proof. $\qquad\qquad\qquad\qquad\qquad\qquad\qquad\qquad\square$

The proof of Lemma G.2 is mostly a direct adaption of the proof of Theorem D.1 and Theorem D.8.

**Proof of Lemma G.2**     For simplicity, we denote $\mathbb{P}^\star(o|\pi) := \mathbb{P}^{M^\star,\pi}(o)$ and $\mathbf{R}^\star(o) := \mathbf{R}^M(o)$.

We pick a $\rho$-optimistic covering $(\widetilde{\mathbb{P}}, \mathcal{M}_0)$ of $\mathcal{M}$ such that $|\mathcal{M}_0| = \mathcal{N}(\mathcal{M}, \rho)$.

Recall that the MLE functional is defined as

$$\mathcal{L}_t(M) := \sum_{s=1}^{t-1} \log \mathbb{P}^{M,\pi^s}(o^s) - \left\|\mathbf{r}^s - \mathbf{R}^M(o^s)\right\|_2^2.$$

For $M \in \mathcal{M}_0$, we consider

$$\ell_M^t := \log\frac{\widetilde{\mathbb{P}}^M(o^t|\pi^t)}{\mathbb{P}^\star(o^t|\pi^t)} + \delta_M^t, \qquad \delta_M^t := \left\|\mathbf{r}^t - \mathbf{R}^\star(o^t)\right\|_2^2 - \left\|\mathbf{r}^t - \mathbf{R}^M(o^t)\right\|_2^2,$$

where the definition of $\delta$ agrees with (11). We first show that with probability at least $1 - \delta$, for all $M \in \mathcal{M}_0$ and all $t \in [T]$,

$$\sum_{s<t} 1 - \mathbb{E}_s\left[\sqrt{\frac{\widetilde{\mathbb{P}}^M(o^s|\pi^s)}{\mathbb{P}^\star(o^s|\pi^s)}}\right] + \frac{1}{2}\mathbb{E}_s\left[\left\|\mathbf{R}^\star(o^s) - \mathbf{R}^M(o^s)\right\|_2^2\right] \leqslant -\sum_{s<t} \ell_M^s + 3\log\frac{|\mathcal{M}_0|}{\delta}. \quad (41)$$

This is because by Lemma B.2, it holds that with probability at least $1 - \delta$, for all $t \in [T]$ and $M \in \mathcal{M}_0$,

$$\sum_{s<t} -\frac{1}{3}\ell_M^s + \log(|\mathcal{M}_0|/\delta) \geqslant \sum_{s<t} -\log \mathbb{E}_s\left[\exp\left(\frac{1}{3}\ell_M^s\right)\right].$$

Further,

$$-\log\mathbb{E}_s\left[\exp\left(\frac{1}{3}\ell_M^s\right)\right] \geqslant -\frac{2}{3}\log\mathbb{E}_s\left[\exp\left(\frac{1}{2}\log\frac{\widetilde{\mathbb{P}}^M(o^s|\pi^s)}{\mathbb{P}^\star(o^s|\pi^s)}\right)\right] - \frac{1}{3}\log\mathbb{E}_s[\exp(\delta_M^s)]$$

$$\geqslant \frac{1}{3}\left(1 - \mathbb{E}_s\left[\sqrt{\frac{\widetilde{\mathbb{P}}^M(o^s|\pi^s)}{\mathbb{P}^\star(o^s|\pi^s)}}\right]\right) + \frac{1}{6}\mathbb{E}_s\left[\left\|\mathbf{R}^\star(o^s) - \mathbf{R}^M(o^s)\right\|_2^2\right],$$

where the second inequality is due to the fact that $-\log x \geqslant 1 - x$ and Lemma D.3 (with $\sigma^2 = 1/4$). Hence (41) is proven.

Now condition on the success of (41) for all $M_0 \in \mathcal{M}_0$. Fix a $M \in \mathcal{M}$, there is a $M_0 \in \mathcal{M}_0$ such that $M$ is covered by $M_0$ (i.e. $\left\|\mathbf{R}^{M_0}(o) - \mathbf{R}^M(o)\right\|_1 \leqslant \rho$ and $\widetilde{\mathbb{P}}^{M_0}(\cdot|\pi) \geqslant \mathbb{P}^M(\cdot|\pi)$ for all $\pi$). Notice that $\sum_{o \in \mathcal{O}} \widetilde{\mathbb{P}}^{M_0}(o|\pi) \leqslant 1 + \rho^2$, and therefore $\left\|\widetilde{\mathbb{P}}^{M_0}(\cdot|\pi) - \mathbb{P}^M(\cdot|\pi)\right\|_1 \leqslant \rho^2$. Then the first term in (41) (plug in $M_0$) can be lower bounded as

$$1 - \mathbb{E}_s\left[\sqrt{\frac{\widetilde{\mathbb{P}}^{M_0}(o^s|\pi^s)}{\mathbb{P}^\star(o^s|\pi^s)}}\right] \geqslant \frac{1}{2}\mathbb{E}_s\left[D_\mathrm{H}^2\left(\mathbb{P}^M(\cdot|\pi^s), \mathbb{P}^\star(\cdot|\pi^s)\right)\right] - \rho,$$

by (22). For the second term, by the fact that $\mathbf{R} \in [0,1]^H$ and $\left\|\mathbf{R}^{M_0}(o) - \mathbf{R}^M(o)\right\|_1 \leqslant \rho$, we have

$$\mathbb{E}_s\left[\left\|\mathbf{R}^\star(o^s) - \mathbf{R}^M(o^s)\right\|_2^2\right] \geqslant \mathbb{E}_s\left[\left\|\mathbf{R}^\star(o^s) - \mathbf{R}^M(o^s)\right\|_2^2\right] - 2\rho.$$

Similarly, $\delta_{M_0}^s \geqslant \delta_M^s - 2\rho$, and hence $-\sum_{s<t} \ell_{M_0}^s \leqslant \mathcal{L}^t(M^\star) - \mathcal{L}^t(M) + 2T\rho$, which completes the proof. $\qquad\square$

### G.2 Proof of Theorem E.5

In the following, we show the following general result.

**Theorem G.3** (Full version of Theorem E.5). *Choosing $\beta \geqslant 3\,\mathrm{est}(\mathcal{M}, 2T) + 3\log(1/\delta) \geqslant 1$, with probability at least $1 - \delta$, Algorithm 6 achieves*

$$\mathbf{Reg_{DM}} \leqslant \inf_{\gamma>0} \{T \cdot \mathrm{mlec}_{\gamma,T}(\mathcal{M}, M^\star) + 4\gamma\beta\}.$$

Especially, when $\mathcal{M}$ is finite, we can take $\beta = 3\log(|\mathcal{M}|/\delta)$ (because $\mathrm{est}(\mathcal{M}, 2T) \leqslant \log|\mathcal{M}|$), and Theorem G.3 implies Theorem E.5 directly.

**Proof of Theorem G.3** Condition on the success of Theorem G.1. Then, for $t \in [T]$, it holds that $M^\star \in \mathcal{M}^t$. Therefore, by the choice of $(M^t, \pi^t)$, it holds that $f^{M^t}(\pi^t) \geqslant f^{M^\star}(\pi_{M^\star})$. Then,

$$\mathbf{Reg_{DM}} = \sum_{t=1}^T \left[f^{M^\star}(\pi_{M^\star}) - f^{M^\star}(\pi^t)\right] \leqslant \sum_{t=1}^T \left[f^{M^t}(\pi^t) - f^{M^\star}(\pi^t)\right]$$

$$= T \cdot \underbrace{\left\{\frac{1}{T}\sum_{t=1}^T \left[f^{M^t}(\pi^t) - f^{M^\star}(\pi^t)\right] - \frac{\gamma}{T}\left[\left(\max_{t\in[T]}\sum_{s\leqslant t-1} D_\mathrm{RL}^2(M^\star(\pi_{M^s}), M^t(\pi_{M^s}))\right) \vee 1\right]\right\}}_{\text{bounded by } \mathrm{mlec}_{\gamma,T}(\mathcal{M}, M^\star)}$$

$$+ \gamma \cdot \underbrace{\left(\max_{t\in[T]}\sum_{s\leqslant t-1} D_\mathrm{RL}^2(M^\star(\pi_{M^s}), M^t(\pi_{M^s}))\right) \vee 1}_{\text{bounded by } 4\beta}$$

$$\leqslant T\,\mathrm{mlec}_{\gamma,T}(\mathcal{M}, M^\star) + 4\gamma\beta.$$

Taking $\inf_{\gamma>0}$ completes the proof. $\qquad\square$

### G.3 Relationship between PSC and MLEC

The MLEC resembles the PSC in that both control a certain *decoupling* error between a family of models and their optimal policies. The main difference is that the MLEC concerns any *sequence* of models whereas the PSC concerns any *distribution* of models. Intuitively, the sequential nature of the MLEC makes it a stronger requirement than the PSC.

Formally, we show that low MLEC with a slightly modified definition of MLEC (where the $\max_{k \in [K]}$ is replaced by the average; cf. (42)) indeed implies low PSC. Note that the modified MLEC defined in (42) is larger than the MLEC defined in Definition E.4; however, in most concrete applications they can be bounded by the same upper bounds. We present Definition E.4 as the main definition of MLEC in order to capture generic classes of problems such as RL with low Bellman-Eluder dimensions or (more generally) low-Eluder-dimension Bellman representations (Proposition K.7).

**Proposition G.4** (Bounding PSC by modified MLEC). *Consider the following modified MLEC:*

$$\widetilde{\mathrm{mlec}}_{\gamma,K}(\mathcal{M}, \overline{M}) := \sup_{\{M^k\} \in \mathcal{M}} \frac{1}{K} \sum_{k=1}^{K} \left[ f^{M^k}(\pi_{M^k}) - f^{\overline{M}}(\pi_{M^k}) \right]$$
$$- \frac{\gamma}{K^2} \left( \sum_{k=1}^{K} \sum_{t \leq k-1} D_{\mathrm{RL}}^2(\overline{M}(\pi_{M^t}), M^k(\pi_{M^t})) \right). \tag{42}$$

*Then it holds that*

$$\mathrm{psc}_\gamma(\mathcal{M}, \overline{M}) \leq \inf_{K \geq 1} \left( \widetilde{\mathrm{mlec}}_{\gamma,K}(\mathcal{M}, \overline{M}) + (\gamma+1) \sqrt{\frac{2 \log(|\Pi| \wedge |\mathcal{M}| + 1)}{K}} \right).$$

**Proof of Proposition G.4**     Fix a $K \geq 1$ and $\mu \in \Delta(\mathcal{M})$. We first prove the bound in terms of $|\Pi|$. The proof uses the probabilistic method to establish the desired deterministic bound. We draw $K$ i.i.d samples $M^1, \cdots, M^K$ from $\mu$, and we write $\widehat{\mu} = \mathrm{Unif}(\{M^1, \cdots, M^K\})$. Then with probability at least $1 - (|\Pi| + 1)\delta$, the following holds simultaneously:

$$(\mathbb{E}_{M \sim \mu} - \mathbb{E}_{M \sim \widehat{\mu}})\left[ f^M(\pi_M) - f^{\overline{M}}(\pi_M) \right] \leq \sqrt{\frac{2 \log(1/\delta)}{K}},$$

$$(\mathbb{E}_{M \sim \widehat{\mu}} - \mathbb{E}_{M \sim \mu})\left[ D_{\mathrm{RL}}^2\left(\overline{M}(\pi), M(\pi)\right) \right] \leq \sqrt{\frac{2 \log(1/\delta)}{K}}, \qquad \forall \pi \in \Pi.$$

Therefore, with probability at least $1 - (|\Pi| + 1)\delta$ (over the randomness of $\widehat{\mu}$), we have

$$\mathbb{E}_{M \sim \mu} \mathbb{E}_{M' \sim \mu} \left[ f^M(\pi_M) - f^{\overline{M}}(\pi_M) - \gamma D_{\mathrm{RL}}^2(\overline{M}(\pi_{M'}), M(\pi_{M'})) \right]$$

$$\leq \mathbb{E}_{M \sim \widehat{\mu}} \mathbb{E}_{M' \sim \widehat{\mu}} \left[ f^M(\pi_M) - f^{\overline{M}}(\pi_M) - \gamma D_{\mathrm{RL}}^2(\overline{M}(\pi_{M'}), M(\pi_{M'})) \right] + (1+\gamma)\sqrt{\frac{2 \log(1/\delta)}{K}}$$

$$\leq \frac{1}{K} \sum_{k=1}^{K} \left[ f^{M^k}(\pi_{M^k}) - f^{\overline{M}}(\pi_{M^k}) \right] - \frac{\gamma}{K^2} \left( \sum_{k=1}^{K} \sum_{t=1}^{K} D_{\mathrm{RL}}^2(\overline{M}(\pi_{M^t}), M^k(\pi_{M^t})) \right)$$

$$+ (1+\gamma)\sqrt{\frac{2 \log(1/\delta)}{K}}$$

$$\leq \widetilde{\mathrm{mlec}}_{\gamma,K}(\mathcal{M}, \overline{M}) + (1+\gamma)\sqrt{\frac{2 \log(1/\delta)}{K}}.$$

In particular, for any $\delta < 1/(1 + |\Pi|)$, the above holds with positive probability, and thus there exists one $\widehat{\mu}$ such that the above holds. Taking $\delta \uparrow 1/(1 + |\Pi|)$ on the right-hand side and supremum over $\mu \in \Delta(\mathcal{M})$ on the left-hand side, we get

$$\mathrm{psc}_\gamma(\mathcal{M}) \leq \widetilde{\mathrm{mlec}}_{\gamma,K}(\mathcal{M}, \overline{M}) + (1+\gamma)\sqrt{\frac{2 \log(|\Pi| + 1)}{K}}.$$

The bound in term of $|\mathcal{M}|$ follows analogously, by noticing that with probability at least $1 - (|\mathcal{M}| + 1)\delta$, the following holds simultaneously:

$$(\mathbb{E}_{M \sim \mu} - \mathbb{E}_{M \sim \widehat{\mu}})\left[ f^M(\pi_M) - f^{\overline{M}}(\pi_M) \right] \leq \sqrt{\frac{2 \log(1/\delta)}{K}},$$

$$(\mathbb{E}_{M' \sim \widehat{\mu}} - \mathbb{E}_{M' \sim \mu})\left[ D_{\mathrm{RL}}^2\left(\overline{M}(\pi_{M'}), M(\pi_{M'})\right) \right] \leq \sqrt{\frac{2 \log(1/\delta)}{K}}, \qquad \forall M \in \mathcal{M},$$

and repeating the same argument as above.     $\square$

---

**Algorithm 7** Exploartive E2D with Tempered Aggregation (EXPLORATIVE E2D)

---

**Input:** Parameter $\gamma > 0$; Learning rate $\eta_{\mathrm{p}} \in (0, \frac{1}{2})$, $\eta_{\mathrm{r}} > 0$.

1: Initialize $\mu^1 \leftarrow \mathrm{Unif}(\mathcal{M})$.
2: **for** $t = 1, \ldots, T$ **do**
3:     Set $(p_{\exp}^t, p_{\mathrm{out}}^t) \leftarrow \arg\min_{(p_{\exp}, p_{\mathrm{out}}) \in \Delta(\Pi)^2} \widehat{V}_{\mathrm{pac},\gamma}^{\mu^t}(p_{\exp}, p_{\mathrm{out}})$, where $\widehat{V}_{\mathrm{pac}}^{\mu^t}$ is defined in (6).

4:     Sample $\pi^t \sim p_{\exp}^t$. Execute $\pi^t$ and observe $(o^t, \mathbf{r}^t)$.
5:     Update randomized model estimator by Tempered Aggregation:

$$\mu^{t+1}(M) \propto_M \mu^t(M) \cdot \exp\left(\eta_{\mathrm{p}} \log \mathbb{P}^{M,\pi^t}(o^t) - \eta_{\mathrm{r}} \left\|\mathbf{r}^t - \mathbf{R}^M(o^t)\right\|_2^2\right). \tag{43}$$

**Output:** Policy $\widehat{p}_{\mathrm{out}} := \frac{1}{T} \sum_{t=1}^T p_{\mathrm{out}}^t$.

---

# H PROOFS FOR SECTION 4.1

We describe the full EXPLORATIVE E2D algorithm in Algorithm 7.

## H.1 PROOF OF THEOREM 5

We first show that Algorithm 7 achieves the following:

$$f^{M^\star}(\pi_{M^\star}) - \mathbb{E}_{\pi \sim p_{\mathrm{out}}}\left[f^{M^\star}(\pi)\right] \le \overline{\mathrm{edec}}_\gamma(\mathcal{M}) + \gamma \frac{\mathbf{Est}_{\mathrm{RL}}}{T}, \tag{44}$$

where $\mathbf{Est}_{\mathrm{RL}}$ denotes the following online estimation error (cf. (18)):

$$\mathbf{Est}_{\mathrm{RL}} := \sum_{t=1}^T \mathbb{E}_{\pi^t \sim p_{\exp}^t} \mathbb{E}_{\widehat{M}^t \sim \mu^t}\left[D_{\mathrm{RL}}^2(\widehat{M}^t(\pi^t), M^\star(\pi^t))\right].$$

We have

$$\sum_{t=1}^T \mathbb{E}_{\pi^t \sim p_{\mathrm{out}}^t}\left[f^{M^\star}(\pi_{M^\star}) - f^{M^\star}(\pi^t)\right]$$

$$= \sum_{t=1}^T \mathbb{E}_{\pi \sim p_{\mathrm{out}}^t}\left[f_M(\pi_M) - f_M(\pi)\right] - \gamma \mathbb{E}_{\pi \sim p_{\exp}^t} \mathbb{E}_{\widehat{M}^t \sim \mu^t}\left[D_{\mathrm{RL}}^2(M(\pi), \widehat{M}^t(\pi))\right]$$

$$\qquad + \gamma \cdot \sum_{t=1}^T \mathbb{E}_{\pi^t \sim p_{\exp}^t} \mathbb{E}_{\widehat{M}^t \sim \mu^t}\left[D_{\mathrm{RL}}^2(M^\star(\pi^t), \widehat{M}^t(\pi^t))\right]$$

$$\overset{(i)}{\le} \sum_{t=1}^T \sup_{M \in \mathcal{M}} \mathbb{E}_{\pi \sim p_{\mathrm{out}}^t}\left[f_M(\pi_M) - f_M(\pi)\right] - \gamma \mathbb{E}_{\pi \sim p_{\exp}^t} \mathbb{E}_{\widehat{M}^t \sim \mu^t}\left[D_{\mathrm{RL}}^2(M(\pi), \widehat{M}^t(\pi))\right]$$

$$\qquad + \gamma \cdot \sum_{t=1}^T \mathbb{E}_{\pi^t \sim p_{\exp}^t} \mathbb{E}_{\widehat{M}^t \sim \mu^t}\left[D_{\mathrm{RL}}^2(M^\star(\pi^t), \widehat{M}^t(\pi^t))\right]$$

$$\overset{(ii)}{=} \sum_{t=1}^T \underbrace{\widehat{V}_{\mathrm{pac},\gamma}^{\mu^t}(p_{\exp}^t, p_{\mathrm{out}}^t)}_{=\inf_{(p_{\exp}, p_{\mathrm{out}}) \in \Delta(\Pi)^2} \widehat{V}_{\mathrm{pac},\gamma}^{\mu^t}} + \gamma \cdot \mathbf{Est}_{\mathrm{RL}}$$

$$\overset{(iii)}{=} \sum_{t=1}^T \mathrm{edec}_\gamma(\mathcal{M}, \mu^t) + \gamma \cdot \mathbf{Est}_{\mathrm{RL}} \le T \cdot \overline{\mathrm{edec}}_\gamma(\mathcal{M}) + \gamma \cdot \mathbf{Est}_{\mathrm{RL}}.$$

Above, (i) follows by the realizability assumption $M^\star \in \mathcal{M}$; (ii) follows by definition of the risk $\widehat{V}_{\mathrm{pac},\gamma}^{\mu^t}$ (cf. (6)) as well as the fact that $(p_{\exp}^t, p_{\mathrm{out}}^t)$ minimizes $\widehat{V}_{\mathrm{pac},\gamma}^{\mu^t}(\cdot, \cdot)$ in Algorithm 7; (iii) follows by definition of $\mathrm{edec}_\gamma(\mathcal{M}, \mu^t)$. Dividing both sides by $T$ proves (44).

Combining (44) with the online estimation guarantee of Tempered Aggregation (Corollary D.2) completes the proof. □

Theorem 5 can be extended to the more general case with infinite model classes. Combining (44) with Proposition D.6, we can establish the following general guarantee of EXPLORATIVE E2D WITH COVERING.

**Theorem H.1** (EXPLORATIVE E2D WITH COVERING). *Suppose model class $\mathcal{M}$ admits a $\rho$-optimistic cover $(\widetilde{\mathbb{P}}, \mathcal{M}_0)$. Then, Algorithm 7 with the Tempered Aggregation subroutine (3) replaced by by (19) with $\eta_{\mathrm{p}} = \eta_{\mathrm{r}} = 1/3$ and $\mu^1 = \mathrm{Unif}(\mathcal{M}_0)$ achieves the following with probability at least $1 - \delta$:*

$$\mathbf{SubOpt} \leqslant \overline{\mathrm{edec}}_\gamma(\mathcal{M}) + \frac{10\gamma}{T}[\log|\mathcal{M}_0| + T\rho + 2\log(2/\delta)].$$

*By tuning $\gamma > 0$, with probability at least $1 - \delta$, Algorithm 7 achieves*

$$\mathbf{SubOpt} \leqslant C \inf_{\gamma > 0} \left\{ \overline{\mathrm{edec}}_\gamma(\mathcal{M}) + \frac{\gamma}{T}[\mathrm{est}(\mathcal{M}, T) + \log(1/\delta)] \right\},$$

*where $C$ is a universal constant.*

## H.2 PROOF OF PROPOSITION 6

We present the full version of Proposition 6 here, and then provide its proof, which is a generalization of Foster et al. (2021, Theorem 3.2).

**Proposition H.2** (PAC lower bound). *Consider a model class $\mathcal{M}$ and $T \geqslant 1$ a fixed integer. Define $V(\mathcal{M}) := 3\sup_{M,\overline{M}} \sup_{\pi,o} \frac{\mathsf{P}^M(o|\pi)}{\mathsf{P}^{\overline{M}}(o|\pi)}$, $C(T) := 2^{13}\log(2T \wedge V(\mathcal{M}))$ and $\underline{\varepsilon}_\gamma := \frac{\gamma}{C(T)T}$. Then we can assign each model $M \in \mathcal{M}$ with a reward distribution (with $\mathbf{r} \in [-2, 2]^H$ almost surely), such that for any algorithm $\mathfrak{A}$, there exists a model $M \in \mathcal{M}$ for which*

$$\mathbb{E}^{M,\mathfrak{A}}[\mathbf{SubOpt}] \geqslant \frac{1}{6} \max_{\gamma > 0} \sup_{\overline{M} \in \mathcal{M}} \mathrm{edec}_\gamma(\mathcal{M}^\infty_{\underline{\varepsilon}_\gamma}(\overline{M}), \overline{M}),$$

*where we define $g^M(\pi) := f^M(\pi_M) - f^M(\pi)$ and the localization*

$$\mathcal{M}^\infty_\varepsilon(\overline{M}) = \left\{ M \in \mathcal{M} : \left| g^M(\pi) - g^{\overline{M}}(\pi) \right| \leqslant \varepsilon, \ \forall \pi \in \Pi \right\}. \tag{45}$$

*Proof.* First, we need to specify the reward distribution for each $M \in \mathcal{M}$, given its mean reward function $\mathbf{R}^M : \mathcal{O} \to [0, 1]^H$: conditional on the observation $o$, we let $\mathbf{r} = (r^h)$ be a random vector, with each entry $r^h$ independently sampled from

$$\mathbb{P}^M\left(r^h = -\frac{1}{2}\Big|o\right) = \frac{3}{4} - \frac{R_h^M(o)}{2}, \qquad \mathbb{P}^M\left(r^h = \frac{3}{2}\Big|o\right) = \frac{1}{4} + \frac{R_h^M(o)}{2}.$$

Then a simple calculation gives $D_{\mathrm{H}}^2\left(\mathsf{R}_h^M(o), \mathsf{R}_h^{\overline{M}}(o)\right) \leqslant \frac{1}{2}\left|R_h^M(o) - R_h^{\overline{M}}(o)\right|^2$. Therefore, by the fact that $(r^h)$ are mutually independent conditional on $o$, we have

$$1 - \frac{1}{2}D_{\mathrm{H}}^2\left(\mathsf{R}^M(o), \mathsf{R}^{\overline{M}}(o)\right) = \prod_h \left(1 - \frac{1}{2}D_{\mathrm{H}}^2\left(\mathsf{R}_h^M(o), \mathsf{R}_h^{\overline{M}}(o)\right)\right) \geqslant \prod_h \left(1 - \frac{1}{4}\left|R_h^M(o) - R_h^{\overline{M}}(o)\right|^2\right)$$

$$\geqslant \exp\left(-\log(4/3)\left\|\mathbf{R}^M(o) - \mathbf{R}^{\overline{M}}(o)\right\|_2^2\right) \geqslant 1 - \log(4/3)\left\|\mathbf{R}^M(o) - \mathbf{R}^{\overline{M}}(o)\right\|_2^2,$$

where the second inequality is because $1 - x/4 \geqslant \exp(-\log(4/3)x)$ for all $x \in [0, 1]$. Therefore, by Lemma B.4,

$$D_{\mathrm{H}}^2\left(M(\pi), \overline{M}(\pi)\right) \leqslant 3D_{\mathrm{H}}^2\left(\mathsf{P}^M(\pi), \mathsf{P}^{\overline{M}}(\pi)\right) + 2\mathbb{E}_{o \sim \mathsf{P}^M(\pi)}\left[D_{\mathrm{H}}^2\left(\mathsf{R}^M(o), \mathsf{R}^{\overline{M}}(o)\right)\right]$$

$$\leqslant 3D_{\mathrm{H}}^2\left(\mathsf{P}^M(\pi), \mathsf{P}^{\overline{M}}(\pi)\right) + 3\mathbb{E}_{o \sim \mathsf{P}^M(\pi)}\left[\left\|\mathbf{R}^M(o) - \mathbf{R}^{\overline{M}}(o)\right\|_2^2\right] \leqslant 3D_{\mathrm{RL}}^2\left(M(\pi), \overline{M}(\pi)\right).$$

Next, suppose that algorithm $\mathfrak{A}$ is given by rules $p = \left\{ p_{\exp}^{(t)}(\cdot \mid \cdot) \right\}_{t=1}^{T} \bigcup \{ p_{\text{out}}(\cdot \mid \cdot) \}$, where $p_{\exp}^{(t)}(\cdot \mid \mathcal{H}^{(t-1)}) \in \Delta(\Pi)$ is the rule of interaction at $t$-th step (given $\mathcal{H}^{(t-1)}$ the history before $t$-th step), and $p_{\text{out}}(\cdot \mid \mathcal{H}^{(T)}) \in \Delta(\Pi)$ is the rule of the output policy. For $M \in \mathcal{M}$, we define

$$p_{M,\exp} = \mathbb{E}^{M,\mathfrak{A}} \left[ \frac{1}{T} \sum_{t=1}^{T} p_{\exp}^{(t)} \left( \cdot \mid \mathcal{H}^{(t-1)} \right) \right] \in \Delta(\Pi), \qquad p_{M,\text{out}} = \mathbb{E}^{M,\mathfrak{A}} \left[ p_{\text{out}}(\cdot \mid \mathcal{H}^{T}) \right] \in \Delta(\Pi),$$

where $\mathbb{P}^{M,\mathfrak{A}}$ is the probability distribution induced by $\mathfrak{A}$ when interacting with $M$. Notice that $\mathbb{E}^{M,\mathfrak{A}}[\mathbf{SubOpt}] = \mathbb{E}_{\pi \sim p_{M,\text{out}}} [g^{M}(\pi)]$.

Let us abbreviate $\text{edec} := \sup_{\overline{M} \in \mathcal{M}} \text{edec}_{\gamma} \left( \mathcal{M}_{\varepsilon}^{\infty}(\overline{M}), \overline{M} \right)$, and let $\overline{M} \in \mathcal{M}$ attains the supremum. Then

$$\sup_{M \in \mathcal{M}_{\varepsilon}^{\infty}(\overline{M})} \mathbb{E}_{\pi \sim p_{\overline{M},\text{out}}} \left[ f^{M}(\pi_{M}) - f^{M}(\pi) \right] - \gamma \mathbb{E}_{\pi \sim p_{\overline{M},\exp}} \left[ D_{\text{RL}}^{2} \left( M(\pi), \overline{M}(\pi) \right) \right] \geqslant \text{edec}.$$

Let $M \in \mathcal{M}_{\varepsilon}^{\infty}(\overline{M})$ attain the supremum above. Then we have

$$\mathbb{E}_{\pi \sim p_{\overline{M},\text{out}}} \left[ g^{M}(\pi) \right] \geqslant \gamma \cdot \mathbb{E}_{\pi \sim p_{\overline{M},\exp}} \left[ D_{\text{RL}}^{2} \left( M(\pi), \overline{M}(\pi) \right) \right] + \text{edec} \geqslant \frac{\gamma}{3} \cdot \mathbb{E}_{\pi \sim p_{\overline{M},\exp}} \left[ D_{\text{H}}^{2} \left( M(\pi), \overline{M}(\pi) \right) \right] + \text{edec}.$$

Recall from the definition of $\mathcal{M}_{\varepsilon}^{\infty}(\overline{M})$ that $\left| g^{M}(\pi) - g^{\overline{M}}(\pi) \right| \leqslant \varepsilon$ for all $\pi$. Hence, by Lemma B.3, it holds that

$$\left| \mathbb{E}_{\pi \sim p_{\overline{M},\text{out}}} \left[ g^{M}(\pi) - g^{\overline{M}}(\pi) \right] - \mathbb{E}_{\pi \sim p_{\overline{M},\text{out}}} \left[ g^{M}(\pi) - g^{\overline{M}}(\pi) \right] \right|$$

$$\leqslant \sqrt{8\varepsilon \cdot \left( \mathbb{E}_{\pi \sim p_{\overline{M},\text{out}}} \left[ g^{M}(\pi) + g^{\overline{M}}(\pi) \right] + \mathbb{E}_{\pi \sim p_{\overline{M},\text{out}}} \left[ g^{M}(\pi) + g^{\overline{M}}(\pi) \right] \right) \cdot D_{\text{H}}^{2} \left( \mathbb{P}^{M,\mathfrak{A}}, \mathbb{P}^{\overline{M},\mathfrak{A}} \right)}$$

$$\leqslant 4\varepsilon D_{\text{H}}^{2} \left( \mathbb{P}^{M,\mathfrak{A}}, \mathbb{P}^{\overline{M},\mathfrak{A}} \right) + \frac{1}{2} \left( \mathbb{E}_{\pi \sim p_{\overline{M},\text{out}}} \left[ g^{M}(\pi) + g^{\overline{M}}(\pi) \right] + \mathbb{E}_{\pi \sim p_{\overline{M},\text{out}}} \left[ g^{M}(\pi) + g^{\overline{M}}(\pi) \right] \right),$$

which implies

$$\mathbb{E}_{\pi \sim p_{\overline{M},\text{out}}} \left[ g^{M}(\pi) \right] + \mathbb{E}_{\pi \sim p_{\overline{M},\text{out}}} \left[ g^{\overline{M}}(\pi) \right] \geqslant \frac{1}{3} \mathbb{E}_{\pi \sim p_{\overline{M},\text{out}}} \left[ g^{M}(\pi) \right] + \frac{1}{3} \mathbb{E}_{\pi \sim p_{\overline{M},\text{out}}} \left[ g^{\overline{M}}(\pi) \right]$$
$$- \frac{8}{3} \varepsilon D_{\text{H}}^{2} \left( \mathbb{P}^{M,\mathfrak{A}}, \mathbb{P}^{\overline{M},\mathfrak{A}} \right).$$

Furthermore, by the subadditivity of the squared Hellinger distance (Foster et al., 2021, Lemma A.13), we have

$$D_{\text{H}}^{2} \left( \mathbb{P}^{M,\mathfrak{A}}, \mathbb{P}^{\overline{M},\mathfrak{A}} \right) \leqslant C_{T} \sum_{t=1}^{T} \mathbb{E}^{\overline{M},\mathfrak{A}} \left[ D_{\text{H}}^{2} \left( M(\pi^{(t)}), \overline{M}(\pi^{(t)}) \right) \right]$$

$$= C_{T} T \cdot \mathbb{E}_{\pi \sim p_{\overline{M},\exp}} \left[ D_{\text{H}}^{2} \left( M(\pi), \overline{M}(\pi) \right) \right],$$

where $C_{T} := 2^{8} \left( \log(2T) \wedge \log(V(\mathcal{M})) \right)$ as in the proof of Foster et al. (2021, Theorem 3.2). As long as $24 C_{T} T \varepsilon \leqslant \gamma$, it holds that

$$\mathbb{E}_{\pi \sim p_{\overline{M},\text{out}}} \left[ g^{M}(\pi) \right] + \mathbb{E}_{\pi \sim p_{\overline{M},\text{out}}} \left[ g^{\overline{M}}(\pi) \right] \geqslant \frac{1}{3} \text{edec}.$$

This completes the proof. $\qquad \square$

### H.3 PROOF OF PROPOSITION 7

Fix a $\overline{\mu} \in \Delta(\mathcal{M})$, and we take

$$(\overline{p}_{\exp}, \overline{p}_{\text{out}}) := \underset{(p_{\exp}, p_{\text{out}}) \in \Delta(\mathcal{M})^{2}}{\arg \min} \sup_{M \in \mathcal{M}} \left\{ \mathbb{E}_{\pi \sim p_{\text{out}}} \left[ f^{M}(\pi_{M}) - f^{M}(\pi) \right] \right.$$

$$\left. - \gamma \mathbb{E}_{\pi \sim p_{\exp}} \mathbb{E}_{\overline{M} \sim \overline{\mu}} \left[ D_{\text{RL}}^{2}(M(\pi), \overline{M}(\pi)) \right] \right\}.$$

Then consider $\overline{p} = \alpha \overline{p}_{\text{exp}} + (1-\alpha)\overline{p}_{\text{out}}$. By definition,

$$
\begin{aligned}
&\text{dec}(\mathcal{M}, \overline{\mu}) \\
&\leqslant \sup_{M \in \mathcal{M}} \mathbb{E}_{\pi \sim \overline{p}} \left[ f^M(\pi_M) - f^M(\pi) \right] - \gamma \mathbb{E}_{\pi \sim \overline{p}} \mathbb{E}_{\overline{M} \sim \overline{\mu}} \left[ D_{\text{RL}}^2(M(\pi), \overline{M}(\pi)) \right] \\
&= \sup_{M \in \mathcal{M}} \left\{ \alpha \mathbb{E}_{\pi \sim \overline{p}_{\text{exp}}} \left[ f^M(\pi_M) - f^M(\pi) \right] - \gamma \alpha \mathbb{E}_{\pi \sim \overline{p}_{\text{exp}}} \mathbb{E}_{\overline{M} \sim \overline{\mu}} \left[ D_{\text{RL}}^2(M(\pi), \overline{M}(\pi)) \right] \right. \\
&\qquad\qquad \left. (1-\alpha) \mathbb{E}_{\pi \sim \overline{p}_{\text{out}}} \left[ f^M(\pi_M) - f^M(\pi) \right] - \gamma(1-\alpha) \mathbb{E}_{\pi \sim \overline{p}_{\text{out}}} \mathbb{E}_{\overline{M} \sim \overline{\mu}} \left[ D_{\text{RL}}^2(M(\pi), \overline{M}(\pi)) \right] \right\} \\
&\leqslant \sup_{M \in \mathcal{M}} \left\{ \alpha + (1-\alpha) \mathbb{E}_{\pi \sim \overline{p}_{\text{out}}} \left[ f^M(\pi_M) - f^M(\pi) \right] - \alpha \gamma \mathbb{E}_{\pi \sim \overline{p}_{\text{exp}}} \mathbb{E}_{\overline{M} \sim \overline{\mu}} \left[ D_{\text{RL}}^2(M(\pi), \overline{M}(\pi)) \right] \right\} \\
&= \alpha + (1-\alpha) \, \text{edec}_{\alpha\gamma/(1-\alpha)}(\mathcal{M}, \overline{\mu}).
\end{aligned}
$$

$\square$

### H.3.1 Additional discussions on bounding DEC by EDEC

Here we argue that, for classes with low EDEC, obtaining a PAC sample complexity through the implied DEC bound is in general worse than the bound obtained by the EDEC bound directly.

Consider any model class $\mathcal{M}$ with $\overline{\text{edec}}_\gamma(\mathcal{M}) \lesssim d/\gamma$, where $d$ is some dimension-like complexity measure. Using the EXPLORATIVE E2D algorithm, by Theorem 5, the suboptimality of the output policy scales as

$$
\begin{aligned}
f^{M^\star}(\pi_{M^\star}) - \mathbb{E}_{\pi \sim \widehat{p}_{\text{out}}} \left[ f^{M^\star}(\pi) \right] &\leqslant \overline{\text{edec}}_\gamma(\mathcal{M}) + 10 \frac{\gamma \log(|\mathcal{M}|/\delta)}{T} \\
&\lesssim \frac{d}{\gamma} + \frac{\gamma}{T} \cdot \log(|\mathcal{M}|/\delta) \lesssim \sqrt{\frac{d \log(|\mathcal{M}|/\delta)}{T}},
\end{aligned}
$$

where the last inequality follows by choosing the optimal $\gamma > 0$. This implies a PAC sample complexity $d \log(|\mathcal{M}|/\delta)/\varepsilon^2$ for finding an $\varepsilon$ near-optimal policy.

By contrast, suppose we use an algorithm designed for low DEC problems (such as E2D-TA). To first bound the DEC by the EDEC, by Proposition 7, we have

$$
\overline{\text{dec}}_\gamma(\mathcal{M}) \leqslant \inf_{\alpha > 0} \left\{ \alpha + (1-\alpha) \overline{\text{edec}}_{\gamma\alpha/(1-\alpha)}(\mathcal{M}) \right\} \leqslant \inf_{\alpha > 0} \left\{ \alpha + (1-\alpha)^2 \frac{d}{\gamma\alpha} \right\} \lesssim \sqrt{\frac{d}{\gamma}}.
$$

Then, using the E2D-TA algorithm, by Theorem 2 and the online-to-batch conversion, the suboptimality of the average policy scales as

$$
\begin{aligned}
\frac{\mathbf{Reg_{DM}}}{T} &= \frac{1}{T} \sum_{t=1}^{T} f^{M^\star}(\pi_{M^\star}) - \mathbb{E}_{\pi^t \sim p^t} \left[ f^{M^\star}(\pi^t) \right] \\
&\leqslant \overline{\text{dec}}_\gamma(\mathcal{M}) + 10 \frac{\gamma \log(|\mathcal{M}|/\delta)}{T} \\
&\lesssim \sqrt{\frac{d}{\gamma}} + \frac{\gamma}{T} \cdot \log(|\mathcal{M}|/\delta) \lesssim \left( \frac{d \log(|\mathcal{M}|/\delta)}{T} \right)^{1/3},
\end{aligned}
$$

where the last inequality follows by choosing the optimal $\gamma > 0$. This implies a PAC sample complexity $d \log(|\mathcal{M}|/\delta)/\varepsilon^3$ for finding an $\varepsilon$ near-optimal policy, which is an $1/\varepsilon$ factor worse than that obtained from the EDEC directly. Note that this $1/\varepsilon^3$ rate is the same as obtained from the standard explore-then-commit conversion from PAC algorithms with sample complexity $1/\varepsilon^2$ to no-regret algorithms.

We remark that the same calculations above also hold in general for problems with $\overline{\text{edec}}_\gamma(\mathcal{M}) \lesssim 1/\gamma^\beta$ (when only highlighting dependence on $\gamma$) for some $\beta > 0$. In that case, the EDEC yields PAC sample complexity $(1/\varepsilon)^{\frac{\beta+1}{\beta}}$, whereas the implied DEC bound only yields a slightly worse $(1/\varepsilon)^{\frac{\beta+2}{\beta}}$ sample complexity.

## H.4 EXAMPLE: EDEC LOWER BOUND FOR TABULAR MDPs

In this section, we follow Domingues et al. (2021); Foster et al. (2021) to construct a class of tabular MDPs whose (localized) EDEC has a desired lower bound, and hence establish a $\Omega\left(HSA/\varepsilon^2\right)$ lower bound of sample complexity for PAC learning in tabular MDPs, recovering the result of Domingues et al. (2021).

**Proposition H.3** (EDEC lower bound for tabular MDPs). *There exists $\mathcal{M}$ a class of MDPs with $S \geqslant 4$ states, $A \geqslant 2$ actions, horizon $H \geqslant 2\log_2(S)$ and the same reward function, such that*

$$\sup_{\overline{M}\in\mathcal{M}} \mathrm{edec}_\gamma(\mathcal{M}_\varepsilon^\infty(\overline{M}), \overline{M}) \geqslant c_1 \min\left\{1, \frac{HSA}{\gamma}\right\},$$

*for all $\gamma > 0$ such that $\varepsilon \geqslant c_2 HSA/\gamma$, where $c_1, c_2$ are two universal constants. As a corollary, applying the PAC lower bound in EDEC (Proposition H.2), we have that for any algorithm $\mathfrak{A}$ that interacts with the environment for $T$ episodes,*

$$\sup_{M\in\mathcal{M}} \mathbb{E}^{M,\mathfrak{A}}[\mathbf{SubOpt}] \geqslant c_0 \min\left\{1, \sqrt{\frac{HSA}{T}}\right\},$$

*where $c_0$ is a universal constant.*

Proposition H.3 implies a sample complexity lower bound of $\Omega\left(HSA/\varepsilon^2\right)$ for learning $\varepsilon$-optimal policy in tabular MDPs. This simple example illustrates the power of edec as a lower bound for PAC learning, analogously to the DEC for no-regret learning.

Moreover, notice that all models in $\mathcal{M}$ have the same reward function (denote it by $R$), hence for this class $\mathcal{M}$ it holds that

$$\mathrm{edec}_\gamma(\mathcal{M}_\varepsilon^\infty(\overline{M}), \overline{M}) \leqslant \mathrm{rfdec}_\gamma(\mathcal{M}_\varepsilon^{\infty,\mathrm{rf}}(\overline{\mathsf{P}}), \overline{\mathsf{P}}) \tag{46}$$

for all $\overline{M} = (\overline{\mathsf{P}}, R) \in \mathcal{M}$. Combining this fact with Proposition H.3 gives a lower bound of RFDEC of $\mathcal{M}$, and hence we can also obtain a sample complexity lower bound for reward-free learning in tabular MDPs.

*Proof.* Without loss of generality, we assume that $S = 2^{n+1} + 1$ and let $S' = 2^n$. We also write $A' = A - 1$, $H' = H - n \geqslant H/2$.

Fix a $\Delta \in (0, \frac{1}{3}]$, we consider $\mathcal{M}^\Delta$ the class of MDPs described as follows.

1. The state space $\mathcal{S} = \mathcal{S}_{\mathrm{tree}} \bigsqcup \{s_\oplus, s_\ominus\}$, where $\mathcal{S}_{\mathrm{tree}}$ is a binary tree of level $n + 1$ (hence $|\mathcal{S}_{\mathrm{tree}}| = 2^{n+1} - 1$), and $s_\oplus, s_\ominus$ are two auxiliary nodes. Let $s_0$ be the root of $\mathcal{S}_{\mathrm{tree}}$, and $S'$ be the set of leaves of $\mathcal{S}_{\mathrm{tree}}$ (hence $|S'| = 2^n$).

2. Each episode has horizon $H$.

3. The reward function is fixed and known: arriving at $s_\oplus$ emits a reward 1, and at all other states the reward is 0.

4. For $h^\star \in \mathcal{H}' := \{n+1, \cdots, H\}, s^\star \in S', a^\star \in [A']$, the transition dynamic of $M = M_{h^\star,s^\star,a^\star}$ is defined as follows:

   - The initial state is always $s_0$.
   - At a node $s \in \mathcal{S}_{\mathrm{tree}}$ such that $s$ is not leaf of $\mathcal{S}_{\mathrm{tree}}$, there are two available actions left and right, with left leads to the left children of $s$ and right leads to the right children of $s$.
   - At leaf $s \in S'$, there are $A$ actions: wait, $1, \cdots, A-1$. The dynamic of $M = M_{h^\star,s^\star,a^\star} \in \mathcal{M}$ at $s$ is given by: $\mathbb{P}_h^M(s|s, \mathsf{wait}) = 1$, and for $a \in [A'], h \in [H]$

$$\mathbb{P}_h^M(s_\oplus|s, a) = \frac{1}{2} + \Delta \cdot \mathbb{1}(h = h^\star, s = s^\star, a = a^\star),$$

$$\mathbb{P}_h^M(s_\ominus|s, a) = \frac{1}{2} - \Delta \cdot \mathbb{1}(h = h^\star, s = s^\star, a = a^\star),$$

- The state $s_\oplus$ always transits to $s_\ominus$, and $s_\ominus$ is the absorbing state (i.e. $\mathbb{P}(s_\ominus|s_\oplus, \cdot) = 1$, $\mathbb{P}(s_\ominus|s_\ominus, \cdot) = 1$).

Let $\overline{M}$ be the MDP model with the same transition dynamic and reward function as above, except that for all $h \in [H], s \in S', a \in [A']$ it holds $\mathbb{P}_h^M(s_\oplus|s,a) = \mathbb{P}_h^M(s_\ominus|s,a) = \frac{1}{2}$. Note that $\overline{M}$ does not depend on $\Delta$. We then define

$$\mathcal{M}^\Delta = \{\overline{M}\} \bigcup \{M_{h,s,a} : (h,s,a) \in \mathcal{H}' \times S' \times [A']\}.$$

Before lower bounding $\mathrm{edec}_\gamma(\mathcal{M}^\Delta, \overline{M})$, we make some preparations. Define

$$\nu_\pi(h,s,a) = \mathbb{P}^{\overline{M},\pi}(s_h = s, a_h = a), \qquad \forall(h,s,a) \in \mathcal{H}' \times S' \times [A'].$$

Note that due to the structure of $\overline{M}$, the events $A_{h,s,a} := \{s_h = s, a_h = a\}$ are disjoint for $(h,s,a) \in \mathcal{H}' \times S' \times [A']$; therefore,

$$\sum_{h\in\mathcal{H}',s\in S',a\in[A']} \nu_\pi(h,s,a) \leqslant 1.$$

Furthermore, for $M = M_{h,s,a} \in \mathcal{M}^\Delta$, we have

$$D_{\mathrm{H}}^2\left(M(\pi), \overline{M}(\pi)\right) = \mathbb{P}^{\overline{M},\pi}(s_h = s, a_h = a)D_{\mathrm{H}}^2\left(\mathbb{P}_h^M(\cdot|s,a), \mathbb{P}_h^{\overline{M}}(\cdot|s,a)\right)$$

because $M(\pi)$ and $\overline{M}(\pi)$ only differs at the conditional probability of $s_{h+1}|s_h = s, a_h = a$. Therefore, due to the fact that $D_{\mathrm{H}}^2\left(\mathbb{P}_h^M(\cdot|s,a), \mathbb{P}_h^{\overline{M}}(\cdot|s,a)\right) = D_{\mathrm{H}}^2\left(\mathrm{Bern}(\frac{1}{2}+\Delta), \mathrm{Bern}(\frac{1}{2})\right) \leqslant 3\Delta^2$, we have

$$D_{\mathrm{H}}^2\left(M(\pi), \overline{M}(\pi)\right) \leqslant 3\nu_\pi(h,s,a)\Delta^2.$$

Now, that for $p_{\mathrm{exp}}, p_{\mathrm{out}} \in \Delta(\pi)$ and $M \in \mathcal{M}'$, we have

$$\mathbb{E}_{\pi\sim p_{\mathrm{out}}}\left[f^M(\pi_M) - f^M(\pi)\right] - \gamma\mathbb{E}_{\pi\sim p_{\mathrm{exp}}}\left[D_{\mathrm{H}}^2\left(M(\pi), \overline{M}(\pi)\right)\right]$$

$$=\frac{1}{2} + \Delta - \mathbb{E}_{\pi\sim p_{\mathrm{out}}}\left[\frac{1}{2} + \Delta\mathbb{P}^{M,\pi}(h^\star, s^\star, a^\star)\right] - \gamma\mathbb{E}_{\pi\sim p_{\mathrm{exp}}}\left[D_{\mathrm{H}}^2\left(M(\pi), \overline{M}(\pi)\right)\right]$$

$$\geqslant\Delta(1 - \mathbb{E}_{\pi\sim p_{\mathrm{out}}}[\nu_\pi(h^\star, s^\star, a^\star)]) - 3\gamma\Delta^2\mathbb{E}_{\pi\sim p_{\mathrm{exp}}}[\nu_\pi(h^\star, s^\star, a^\star)].$$

Therefore, we define

$$\nu_{p_{\mathrm{exp}}}(h,s,a) = \mathbb{E}_{\pi\sim p_{\mathrm{exp}}}[\nu_\pi(h,s,a)], \qquad \nu_{p_{\mathrm{out}}}(h,s,a) = \mathbb{E}_{\pi\sim p_{\mathrm{out}}}[\nu_\pi(h,s,a)].$$

Then, for any fixed $p_{\mathrm{exp}}, p_{\mathrm{out}} \in \Delta(\Pi)$, by the fact that

$$\sum_{h\in\mathcal{H}',s\in S',a\in[A']} \{\nu_{p_{\mathrm{exp}}}(h,s,a) + 3\gamma\Delta\nu_{p_{\mathrm{out}}}(h,s,a)\} \leqslant 1 + 3\gamma\Delta,$$

we know that there exists $(h', s', a') \in \mathcal{H}' \times S' \times [A']$ such that

$$\nu_{p_{\mathrm{exp}}}(h', s', a') + 3\gamma\Delta\nu_{p_{\mathrm{out}}}(h', s', a') \leqslant \frac{1 + 3\gamma\Delta}{H'S'A'}.$$

Then we can consider $M' = M_{h',s',a'}$, and

$$\sup_{M\in\mathcal{M}'} \mathbb{E}_{\pi\sim p_{\mathrm{out}}}\left[f^M(\pi_M) - f^M(\pi)\right] - \gamma\mathbb{E}_{\pi\sim p_{\mathrm{exp}}}\left[D_{\mathrm{H}}^2\left(M(\pi), \overline{M}(\pi)\right)\right]$$

$$\geqslant\mathbb{E}_{\pi\sim p_{\mathrm{out}}}\left[f^{M'}(\pi_{M'}) - f^{M'}(\pi)\right] - \gamma\mathbb{E}_{\pi\sim p_{\mathrm{exp}}}\left[D_{\mathrm{H}}^2\left(M'(\pi), \overline{M}(\pi)\right)\right]$$

$$\geqslant\Delta\left(1 - \nu_{p_{\mathrm{exp}}}(h', s', a')\right) - 3\gamma\Delta^2\nu_{p_{\mathrm{out}}}(h', s', a')$$

$$\geqslant\Delta - \Delta \cdot \frac{1 + 3\gamma\Delta}{H'S'A'}.$$

By the arbitrariness of $p_{\mathrm{exp}}, p_{\mathrm{out}} \in \Delta(\Pi)$, we derive that

$$\mathrm{edec}_\gamma(\mathcal{M}^\Delta, \overline{M}) \geqslant \Delta - \Delta \cdot \frac{1 + 3\gamma\Delta}{H'S'A'}.$$

---

**Algorithm 8** REWARD-FREE E2D

---

1: **Input:** Parameters $\eta = 1/3$, $\gamma > 0$; prior distribution $\mu^1 \in \Delta(\mathcal{P})$.
2: *// Exploration phase*
3: **for** $t = 1, \dots, T$ **do**
4:   Set $p_{\exp}^t = \arg\min_{p_{\exp} \in \Delta(\Pi)} \sup_{R \in \mathcal{R}} \widehat{V}_{\mathrm{rf},\gamma}^{\mu^t}(p_{\exp}, R)$, where (cf. (7))

$$\widehat{V}_{\mathrm{rf},\gamma}^{\mu^t}(p_{\exp}, R) := \inf_{p_{\mathrm{out}}} \sup_{\mathsf{P} \in \mathcal{P}} \mathbb{E}_{\pi \sim p_{\mathrm{out}}} \big[ f^{\mathsf{P},R}(\pi_{\mathsf{P},R}) - f^{\mathsf{P},R}(\pi) \big] - \gamma \mathbb{E}_{\pi \sim p_{\exp}} \mathbb{E}_{\widehat{\mathsf{P}}^t \sim \mu^t} \Big[ D_{\mathrm{H}}^2(\mathsf{P}(\pi), \widehat{\mathsf{P}}^t(\pi)) \Big].$$

5:   Sample $\pi^t \sim p_{\exp}^t$. Execute $\pi^t$ and observe $o^t$.
6:   Compute $\mu^{t+1} \in \Delta(\mathcal{P})$ by Tempered Aggregation with observations only:

$$\mu^{t+1}(\mathsf{P}) \propto_{\mathsf{P}} \mu^t(\mathsf{P}) \cdot \exp\left( \eta \log \mathsf{P}^{\pi^t}(o^t) \right). \tag{47}$$

7: *// Planning phase*
8: **Input:** $R^\star \in \mathcal{R}$
9: **for** $t = 1, \dots, T$ **do**
10:   Compute

$$p_{\mathrm{out}}^t(R^\star) := \arg\min_{p_{\mathrm{out}} \in \Delta(\Pi)} \sup_{\mathsf{P} \in \mathcal{P}} \mathbb{E}_{\pi \sim p_{\mathrm{out}}} \Big[ f^{\mathsf{P},R^\star}(\pi_{\mathsf{P},R}) - f^{\mathsf{P},R^\star}(\pi) \Big] - \gamma \mathbb{E}_{\pi \sim p_{\exp}^t} \mathbb{E}_{\widehat{\mathsf{P}}^t \sim \mu^t} \Big[ D_{\mathrm{H}}^2(\mathsf{P}(\pi), \widehat{\mathsf{P}}^t(\pi)) \Big].$$

11: **Output:** $\widehat{p}_{\mathrm{out}}(R^\star) = \frac{1}{T} \sum_{t=1}^T p_{\mathrm{out}}^t(R^\star)$.

---

Also, by definition, it holds that $\mathcal{M}_{\varepsilon,\infty}^\Delta(\overline{M}) = \mathcal{M}^\Delta$,[13] $V(\mathcal{M}^\Delta) = \frac{3}{1-\Delta} \leqslant 6$. Therefore, given $T \geqslant 1$ and algorithm $\mathfrak{A}$, we set $C_0 = 2^{14}$, $\Delta = \min\left\{ \frac{1}{3}, \sqrt{\frac{H'S'A'}{12C_0 T}} \right\}$, $\gamma = C_0 T \Delta$, then applying Proposition H.2 to $\mathcal{M}^\Delta$ gives a $M \in \mathcal{M}^\Delta$ such that

$$\mathbb{E}^{M,\mathfrak{A}}[\mathbf{SubOpt}] \geqslant \frac{1}{6}\left( \Delta - \Delta \cdot \frac{1 + 3\gamma\Delta}{H'S'A'} \right) \geqslant \frac{\Delta}{24} \geqslant c_0 \min\left\{ 1, \sqrt{\frac{H'S'A'}{T}} \right\},$$

where the second inequality is due to the fact that $\frac{1+3\gamma\Delta}{H'S'A'} \leqslant \frac{3}{4}$ which follows from simple calculation, and $c_0$ is a universal constant.

On the other hand, we can similarly provide lower bound of $\mathrm{edec}(\mathcal{M}, \overline{M})$ for the model class $\mathcal{M} = \bigcup_{\Delta>0} \mathcal{M}^\Delta$: For any given $\gamma > 0$, we can take $\varepsilon = \Delta = \min\left\{ \frac{1}{3}, \frac{H'S'A'}{12\gamma} \right\}$, and then

$$\mathrm{edec}_\gamma(\mathcal{M}_\varepsilon^\infty(\overline{M}), \overline{M}) \geqslant \mathrm{edec}_\gamma(\mathcal{M}^\Delta, \overline{M}) \geqslant \Delta - \Delta \cdot \frac{1 + 3\gamma\Delta}{H'S'A'} \geqslant \frac{\Delta}{4} = \frac{1}{4} \min\left\{ \frac{1}{3}, \frac{H'S'A'}{12\gamma} \right\}.$$
$\square$

# I  PROOFS FOR SECTION 4.2

## I.1  ALGORITHM REWARD-FREE E2D

Algorithm 8 presents a slightly more general version of the REWARD-FREE E2D algorithm where we allow $|\mathcal{P}|$ to be possibly infinite. The algorithm described in Section 4.2 is a special case of Algorithm 8 with $|\mathcal{P}| < \infty$ and we pick the uniform prior $\mu^1 = \mathrm{Unif}(\mathcal{P})$.

More generally, just as Algorithm 7, Algorithm 8 can also apply to the case when $\mathcal{M}$ only admits a finite optimistic covering. Its guarantee in this setting is stated as follows.

**Theorem I.1** (REWARD-FREE E2D). *Given a $\rho$-optimistic cover $(\widetilde{\mathsf{P}}, \mathcal{P}_0)$ of $\mathcal{P}$, we can replace the subroutine (47) in Algorithm 8 with*

$$\mu^{t+1}(\mathsf{P}) \propto_{\mathsf{P}} \mu^t(\mathsf{P}) \cdot \exp\left( \eta \log \widetilde{\mathsf{P}}^{\pi^t}(o^t) \right). \tag{48}$$

---

[13] Here to avoid confusion, we move the $\infty$ in the superscript (cf. (45)) to the subscript.

*and let $\eta = 1/2$, $\mu^1 = \mathrm{Unif}(\mathcal{P}_0)$, then Algorithm 8 achieves the following with probability at least $1 - \delta$:*

$$\mathbf{SubOpt_{rf}} \leqslant \overline{\mathrm{rfdec}}_\gamma(\mathcal{M}) + \frac{2\gamma}{T}[\log|\mathcal{P}_0| + T\rho + 2\log(2/\delta)].$$

*By tuning $\gamma > 0$, with probability at least $1 - \delta$, Algorithm 8 achieves the following:*

$$\mathbf{SubOpt_{rf}} \leqslant 4\inf_{\gamma > 0}\left\{\overline{\mathrm{rfdec}}_\gamma(\mathcal{M}) + \frac{\gamma}{T}[\mathrm{est}(\mathcal{P}, T) + \log(1/\delta)]\right\}.$$

### I.2  PROOF OF THEOREM 9 AND THEOREM I.1

For $\mathsf{P} \in \mathcal{P}, R \in \mathcal{R}, p_{\exp}, p_{\mathrm{out}} \in \Delta(\Pi)$, we consider the function

$$V^t(p_{\exp}, R; p_{\mathrm{out}}, \mathsf{P}) := \mathbb{E}_{\pi \sim p_{\mathrm{out}}}\left[f^{\mathsf{P}, R}(\pi_{\mathsf{P}, R}) - f^{\mathsf{P}, R}(\pi)\right] - \gamma\mathbb{E}_{\overline{\mathsf{P}} \sim \mu^t}\mathbb{E}_{\pi \sim p_{\exp}}\left[D_{\mathrm{H}}^2(\mathsf{P}(\pi), \overline{\mathsf{P}}(\pi))\right].$$

Then the policy played by Algorithm 8 at step $t$ is exactly

$$p_{\exp}^t = \underset{p_{\exp} \in \Delta(\Pi)}{\arg\min}\sup_{R \in \mathcal{R}}\widehat{V}_{\mathrm{rf}, \gamma}^{\mu^t}(p_{\exp}, R) = \underset{p_{\exp} \in \Delta(\Pi)}{\arg\min}\sup_{R \in \mathcal{R}}\inf_{p_{\mathrm{out}} \in \Delta(\Pi)}\sup_{\mathsf{P} \in \mathcal{P}}V^t(p_{\exp}, R; p_{\mathrm{out}}, \mathsf{P}).$$

Therefore, for any $R^\star \in \mathcal{R}$, using the definition of the infs and sups in the risks, we have that

$$\overline{\mathrm{rfdec}}_\gamma(\mathcal{M}) \geqslant \mathrm{rfdec}_\gamma(\mathcal{M}, \mu^t) = \sup_{R \in \mathcal{R}}\widehat{V}^{\mu^t}(p_{\exp}^t, R) \geqslant \widehat{V}^{\mu^t}(p_{\exp}^t, R^\star)$$

$$\geqslant \inf_{p_{\mathrm{out}} \in \Delta(\Pi)}\sup_{\mathsf{P} \in \mathcal{P}}V^t(p_{\exp}^t, R^\star; p_{\mathrm{out}}, \mathsf{P})$$

$$= \sup_{\mathsf{P} \in \mathcal{P}}V^t(p_{\exp}^t, R^\star; p_{\mathrm{out}}^t(R^\star), \mathsf{P}) \geqslant V^t(p_{\exp}^t, R^\star; p_{\mathrm{out}}^t(R^\star), \mathsf{P}^\star).$$

Therefore, for any $R^\star \in \mathcal{R}$ and the associated $M^\star = (\mathsf{P}^\star, R^\star)$, we have

$$\mathbb{E}_{\pi \sim p_{\mathrm{out}}^t(R^\star)}\left[f^{M^\star}(\pi_{M^\star}) - f^{M^\star}(\pi)\right] \leqslant \overline{\mathrm{rfdec}}_\gamma(\mathcal{M}) + \gamma\mathbb{E}_{\overline{\mathsf{P}} \sim \mu^t}\mathbb{E}_{\pi \sim p_{\exp}^t}\left[D_{\mathrm{H}}^2(\mathsf{P}^\star(\pi), \overline{\mathsf{P}}(\pi))\right],$$

Taking average over $t \in [T]$ and taking supremum over $R^\star \in \mathcal{R}$ on the left-hand side, we get

$$\sup_{R^\star \in \mathcal{R}}\left\{f^{\mathsf{P}^\star, R^\star}(\pi_{\mathsf{P}^\star, R^\star}) - \mathbb{E}_{\pi \sim \widehat{p}_{\mathrm{out}}(R^\star)}\left[f^{\mathsf{P}^\star, R^\star}(\pi)\right]\right\} \leqslant \overline{\mathrm{rfdec}}_\gamma(\mathcal{M}) + \gamma \cdot \frac{\mathbf{Est_H}}{T}, \qquad (49)$$

where

$$\mathbf{Est_H} := \sum_{t=1}^T \mathbb{E}_{\overline{\mathsf{P}} \sim \mu^t}\mathbb{E}_{\pi \sim p_{\exp}^t}\left[D_{\mathrm{H}}^2(\mathsf{P}^\star(\pi), \overline{\mathsf{P}}(\pi))\right].$$

Note that (52) holds regardless of the subroutine ((47) or (48)) we use. Therefore, it remains to bound $\mathbf{Est_H}$ for (47) and (48).

When $|\mathcal{P}| < \infty$, Corollary D.2 implies that subroutine (47) (agrees with (3) with $\eta_{\mathrm{r}} = 0$ and $\eta_{\mathrm{p}} = \eta$) achieves with probability at least $1 - \delta$ that

$$\mathbf{Est_H} \leqslant \frac{1}{\eta}\log(|\mathcal{P}|/\delta).$$

This proves Theorem 9.

Similarly, when a $\rho$-optimistic covering $(\widetilde{\mathsf{P}}, \mathcal{P}_0)$ is given, Theorem D.8 implies that: subroutine (48) (agrees with (19) with $\eta_{\mathrm{r}} = 0$ and $\eta_{\mathrm{p}} = \eta$) achieves with probability at least $1 - \delta$ that

$$\mathbf{Est_H} \leqslant \frac{1}{\eta}[\log|\mathcal{P}_0| + T\rho + 2\log(2/\delta)].$$

This completes the proof of Theorem I.1. $\qquad\qquad\square$

### I.3   PROOF OF PROPOSITION 10

We state and prove the formal version of Proposition 10 as follows.

**Proposition I.2** (Reward-free lower bound). *Consider a model class $\mathcal{M} = \mathcal{P} \times \mathcal{R}$ and $T \geq 1$ a fixed integer. Define $V(\mathcal{M})$, $C(T)$ and $\underline{\varepsilon}_\gamma$ as in Proposition H.2. Then for any algorithm $\mathfrak{A}$ that returns a mapping $p_{\mathrm{out}} : \mathcal{R} \to \Delta(\Pi)$ after $T$ episodes, there exists a model $\mathsf{P} \in \mathcal{P}$ for which*

$$\mathbb{E}^{\mathsf{P},\mathfrak{A}}[\mathbf{SubOpt_{rf}}] \geq \sup_{R \in \mathcal{R}} \mathbb{E}^{\mathsf{P},\mathfrak{A}}\big[f^{\mathsf{P},R}(\pi_{\mathsf{P},R}) - \mathbb{E}_{\pi^{\mathrm{out}} \sim p_{\mathrm{out}}(R)}\big[f^{\mathsf{P},R}(\pi^{\mathrm{out}})\big]\big]$$

$$\geq \frac{1}{6} \cdot \max_{\gamma > 0} \sup_{\overline{\mathsf{P}} \in \mathcal{P}} \mathrm{rfdec}_\gamma(\mathcal{M}^{\infty,\mathrm{rf}}_{\underline{\varepsilon}_\gamma}(\overline{\mathsf{P}}), \overline{\mathsf{P}}),$$

*where the localization is defined as $\mathcal{M}^{\infty,\mathrm{rf}}_\varepsilon(\overline{\mathsf{P}}) := \mathcal{P}^{\infty,\mathrm{rf}}_\varepsilon(\overline{\mathsf{P}}) \times \mathcal{R}$, where*

$$\mathcal{P}^{\infty,\mathrm{rf}}_\varepsilon(\overline{\mathsf{P}}) = \Big\{\mathsf{P} \in \mathcal{P} : \big|g^{\mathsf{P},R}(\pi) - g^{\overline{\mathsf{P}},R}(\pi)\big| \leq \varepsilon, \ \forall \pi \in \Pi, R \in \mathcal{R}\Big\}, \tag{50}$$

*and $g^{\mathsf{P},R}(\pi) := f^{\mathsf{P},R}(\pi_{\mathsf{P},R}) - f^{\mathsf{P},R}(\pi)$ for any $(\mathsf{P}, R, \pi) \in \mathcal{P} \times \mathcal{R} \times \Pi$.*

*Proof.* Suppose that algorithm $\mathfrak{A}$ is given by rules $p = \big\{p^{(t)}_{\exp}(\cdot \mid \cdot)\big\}^T_{t=1} \bigcup \{p_{\mathrm{out}}(\cdot \mid \cdot)\}$, where $p^{(t)}_{\exp}\big(\cdot|\mathcal{H}^{(t-1)}\big) \in \Delta(\Pi)$ is the rule of interaction at $t$-th step with $\mathcal{H}^{(t-1)}$ the history before $t$-th step, and $p_{\mathrm{out}}\big(\cdot|\mathcal{H}^{(T)}, \cdot\big) : \mathcal{R} \to \Delta(\Pi)$ is the rule of outputting the policy given a reward. Let $\mathbb{P}^{\mathsf{P},\mathfrak{A}}$ refers to the distribution (of $\mathcal{H}^{(T)}$) induced by the exploration phase of $\mathfrak{A}$ under transition $\mathsf{P}$.

For any $\mathsf{P} \in \mathcal{M}$ and $R \in \mathcal{R}$, we define

$$p_{\exp}(\mathsf{P}) = \mathbb{E}^{\mathsf{P},\mathfrak{A}}\left[\frac{1}{T}\sum^T_{t=1} p^{(t)}_{\exp}\big(\cdot \mid \mathcal{H}^{(t-1)}\big)\right] \in \Delta(\Pi), \quad p_{\mathrm{out}}(\mathsf{P}, R) = \mathbb{E}^{\mathsf{P},\mathfrak{A}}\Big[p_{\mathrm{out}}(\cdot|\mathcal{H}^{(T)}, R)\Big] \in \Delta(\Pi).$$

Notice that $\mathbb{E}^{\mathsf{P},\mathfrak{A}}\big[f^{\mathsf{P},R}(\pi_{\mathsf{P},R}) - \mathbb{E}_{\pi^{\mathrm{out}} \sim p_{\mathrm{out}}(R)}\big[f^{\mathsf{P},R}(\pi^{\mathrm{out}})\big]\big] = \mathbb{E}_{\pi \sim p_{\mathrm{out}}(\mathsf{P},R)}\big[g^{\mathsf{P},R}(\pi)\big]$.

Let us abbreviate $\mathrm{rfdec} := \sup_{\overline{\mathsf{P}} \in \mathcal{P}} \mathrm{rfdec}_\gamma(\mathcal{M}^{\infty,\mathrm{rf}}_{\underline{\varepsilon}_\gamma}(\overline{\mathsf{P}}), \overline{\mathsf{P}})$, and let $\overline{\mathsf{P}} \in \mathcal{P}$ attain the supremum. Plug in $p_{\exp} = p_{\exp}(\overline{\mathsf{P}})$ and by definition of $\mathrm{rfdec}_\gamma$ (which considers $\inf_{p_{\exp}}$), we have

$$\sup_{R \in \mathcal{R}} \inf_{p_{\mathrm{out}} \in \Delta(\Pi)} \sup_{\mathsf{P} \in \mathcal{P}^{\infty,\mathrm{rf}}_{\underline{\varepsilon}_\gamma}(\overline{\mathsf{P}})} \mathbb{E}_{\pi \sim p_{\mathrm{out}}} \big[f^{\mathsf{P},R}(\pi_{\mathsf{P},R}) - f^{\mathsf{P},R}(\pi)\big] - \gamma \mathbb{E}_{\pi \sim p_{\exp}(\overline{\mathsf{P}})} \big[D^2_{\mathrm{H}}\big(\mathsf{P}(\pi), \overline{\mathsf{P}}(\pi)\big)\big] \geq \mathrm{rfdec}.$$

Let $R \in \mathcal{R}$ attain the supremum above, and plug in $p_{\mathrm{out}} = p_{\mathrm{out}}(\overline{\mathsf{P}}, R)$ and by definition of the above (which considers $\inf_{p_{\mathrm{out}}}$), we have

$$\sup_{\mathsf{P} \in \mathcal{P}^{\infty,\mathrm{rf}}_{\underline{\varepsilon}_\gamma}(\overline{\mathsf{P}})} \mathbb{E}_{\pi \sim p_{\mathrm{out}}(\overline{\mathsf{P}},R)} \big[f^{\mathsf{P},R}(\pi_{\mathsf{P},R}) - f^{\mathsf{P},R}(\pi)\big] - \gamma \mathbb{E}_{\pi \sim p_{\exp}(\overline{\mathsf{P}})} \big[D^2_{\mathrm{H}}\big(\mathsf{P}(\pi), \overline{\mathsf{P}}(\pi)\big)\big] \geq \mathrm{rfdec}.$$

Let $\mathsf{P} \in \mathcal{P}^{\infty,\mathrm{rf}}_{\underline{\varepsilon}_\gamma}(\overline{\mathsf{P}})$ attain the supremum above. Then we have

$$\mathbb{E}_{\pi \sim p_{\mathrm{out}}(\overline{\mathsf{P}},R)}\big[g^{\mathsf{P},R}(\pi)\big] \geq \gamma \cdot \mathbb{E}_{\pi \sim p_{\exp}(\overline{\mathsf{P}})} \big[D^2_{\mathrm{H}}\big(\mathsf{P}(\pi), \overline{\mathsf{P}}(\pi)\big)\big] + \mathrm{rfdec}.$$

Similar to the proof of Proposition H.2, it holds that

$$\mathbb{E}_{\pi \sim p_{\mathrm{out}}(\mathsf{P},R)} \big[g^{\mathsf{P},R}(\pi)\big] + \mathbb{E}_{\pi \sim p_{\mathrm{out}}(\overline{\mathsf{P}},R)} \big[g^{\overline{\mathsf{P}},R}(\pi)\big]$$

$$\geq \frac{1}{3}\mathbb{E}_{\pi \sim p_{\mathrm{out}}(\overline{\mathsf{P}},R)} \big[g^{\mathsf{P},R}(\pi)\big] + \frac{1}{3}\mathbb{E}_{\pi \sim p_{\mathrm{out}}(\overline{\mathsf{P}},R)} \big[g^{\overline{\mathsf{P}},R}(\pi)\big] - \frac{8}{3}\varepsilon D^2_{\mathrm{H}}\big(\mathbb{P}^{\mathsf{P},\mathfrak{A}}, \mathbb{P}^{\overline{\mathsf{P}},\mathfrak{A}}\big),$$

and

$$D^2_{\mathrm{H}}\big(\mathbb{P}^{\mathsf{P},\mathfrak{A}}, \mathbb{P}^{\overline{\mathsf{P}},\mathfrak{A}}\big) \leq C_T T \cdot \mathbb{E}_{\pi \sim p_{\exp}(\overline{\mathsf{P}})} \big[D^2_{\mathrm{H}}\big(\mathsf{P}(\pi), \overline{\mathsf{P}}(\pi)\big)\big],$$

where we recall that $\mathbb{P}^{\mathsf{P},\mathfrak{A}}$ refers to the distribution induced by the exploration phase of $\mathfrak{A}$. Requiring $8C_T T \varepsilon \leq \gamma$ gives

$$\mathbb{E}_{\pi \sim p_{\mathrm{out}}(\mathsf{P},R)} \big[g^{\mathsf{P},R}(\pi)\big] + \mathbb{E}_{\pi \sim p_{\mathrm{out}}(\overline{\mathsf{P}},R)} \big[g^{\overline{\mathsf{P}},R}(\pi)\big] \geq \frac{1}{3}\mathrm{rfdec},$$

which completes the proof. $\qquad\square$

### I.4 TRACTABLE UPPER BOUND OF RFDEC

In the following, we provide a more intuitive upper bound of rfdec useful for later proofs, which we call rrec:

$$\mathrm{rrec}_\gamma(\mathcal{M},\overline{\mu}) := \inf_{p\in\Delta(\Pi),\tilde{\mu}\in\Delta(\mathcal{P})} \sup_{\mathsf{P}\in\mathcal{P},R\in\mathcal{R},\pi'\in\Pi} \left| \mathbb{E}_{\overline{\mathsf{P}}\sim\tilde{\mu}}\left[ f^{\mathsf{P},R}(\pi') - f^{\overline{\mathsf{P}},R}(\pi') \right] \right| - \gamma\mathbb{E}_{\overline{\mathsf{P}}\sim\overline{\mu}}\mathbb{E}_{\pi\sim p}\left[ D_{\mathrm{H}}^2(\mathsf{P}(\pi),\overline{\mathsf{P}}(\pi)) \right].$$

**Proposition I.3.** *Suppose that $\Pi$ is finite. Then*

$$\mathrm{rfdec}_\gamma(\mathcal{M},\overline{\mu}) \leqslant 2\,\mathrm{rrec}_{\gamma/2}(\mathcal{M},\overline{\mu})$$

*Proof.* Fix any $\tilde{\mu}\in\Delta(\mathcal{P})$. By definition,

$$\mathrm{rfdec}_\gamma(\mathcal{M},\overline{\mu}) \overset{(i)}{=} \inf_{p_{\exp}\in\Delta(\Pi)} \sup_{R\in\mathcal{R}} \sup_{\mu\in\Delta(\mathcal{M})} \inf_{p_{\mathrm{out}}\in\Delta(\Pi)} \mathbb{E}_{\mathsf{P}\sim\mu}\mathbb{E}_{\pi\sim p_{\mathrm{out}}}\left[ f^{\mathsf{P},R}(\pi_{\mathsf{P},R}) - f^{\mathsf{P},R}(\pi) \right]$$
$$- \gamma\mathbb{E}_{\mathsf{P}\sim\mu}\mathbb{E}_{\overline{\mathsf{P}}\sim\overline{\mu}}\mathbb{E}_{\pi\sim p_{\exp}}\left[ D_{\mathrm{H}}^2(\mathsf{P}(\pi),\overline{\mathsf{P}}(\pi)) \right]$$

$$= \inf_{p_{\exp}\in\Delta(\Pi)} \sup_{R\in\mathcal{R}} \sup_{\mu\in\Delta(\mathcal{P})} \inf_{p_{\mathrm{out}}\in\Delta(\Pi)} \mathbb{E}_{\mathsf{P}\sim\mu}\left[ f^{\mathsf{P},R}(\pi_{\mathsf{P},R}) \right] - \mathbb{E}_{\pi\sim p_{\mathrm{out}}}\mathbb{E}_{\overline{\mathsf{P}}\sim\tilde{\mu}}\left[ f^{\overline{\mathsf{P}},R}(\pi) \right]$$
$$+ \mathbb{E}_{\mathsf{P}\sim\mu}\mathbb{E}_{\pi\sim p_{\mathrm{out}}}\mathbb{E}_{\overline{\mathsf{P}}\sim\tilde{\mu}}\left[ f^{\overline{\mathsf{P}},R}(\pi) - f^{\mathsf{P},R}(\pi) \right]$$
$$- \gamma\mathbb{E}_{\mathsf{P}\sim\mu}\mathbb{E}_{\overline{\mathsf{P}}\sim\overline{\mu}}\mathbb{E}_{\pi\sim p_{\exp}}\left[ D_{\mathrm{H}}^2(\mathsf{P}(\pi),\overline{\mathsf{P}}(\pi)) \right]$$

$$\overset{(ii)}{\leqslant} \inf_{p_{\exp}\in\Delta(\Pi)} \sup_{R\in\mathcal{R}} \sup_{\mu\in\Delta(\mathcal{P})} \mathbb{E}_{\mathsf{P}\sim\mu}\mathbb{E}_{\overline{\mathsf{P}}\sim\tilde{\mu}}\left[ f^{\mathsf{P},R}(\pi_{\mathsf{P},R}) - f^{\overline{\mathsf{P}},R}(\pi_{\mathsf{P},R}) \right]$$
$$+ \mathbb{E}_{\mathsf{P}\sim\mu}\mathbb{E}_{\mathsf{P}'\sim\mu}\mathbb{E}_{\overline{\mathsf{P}}\sim\tilde{\mu}}\left[ f^{\overline{\mathsf{P}},R}(\pi_{\mathsf{P}',R}) - f^{\mathsf{P},R}(\pi_{\mathsf{P}',R}) \right]$$
$$- \gamma\mathbb{E}_{\mathsf{P}\sim\mu}\mathbb{E}_{\overline{\mathsf{P}}\sim\overline{\mu}}\mathbb{E}_{\pi\sim p_{\exp}}\left[ D_{\mathrm{H}}^2(\mathsf{P}(\pi),\overline{\mathsf{P}}(\pi)) \right]$$

$$\overset{(iii)}{\leqslant} \inf_{p_{\exp}\in\Delta(\Pi)} \sup_{R\in\mathcal{R}} \sup_{\mu\in\Delta(\mathcal{P})} 2\mathbb{E}_{\mathsf{P}\sim\mu}\left[ \sup_{\pi'\in\Pi}\left| \mathbb{E}_{\overline{\mathsf{P}}\sim\tilde{\mu}}\left[ f^{\mathsf{P},R}(\pi') - f^{\overline{\mathsf{P}},R}(\pi') \right] \right| \right]$$
$$- \gamma\mathbb{E}_{\mathsf{P}\sim\mu}\mathbb{E}_{\overline{\mathsf{P}}\sim\overline{\mu}}\mathbb{E}_{\pi\sim p_{\exp}}\left[ D_{\mathrm{H}}^2(\mathsf{P}(\pi),\overline{\mathsf{P}}(\pi)) \right]$$

$$= 2\inf_{p_{\exp}\in\Delta(\Pi)} \sup_{R\in\mathcal{R},\mathsf{P}\in\mathcal{P}} \Bigg\{ \sup_{\pi'\in\Pi}\left| \mathbb{E}_{\overline{\mathsf{P}}\sim\tilde{\mu}}\left[ f^{\overline{\mathsf{P}},R}(\pi') - f^{\mathsf{P},R}(\pi') \right] \right|$$
$$- \frac{\gamma}{2}\mathbb{E}_{\overline{\mathsf{P}}\sim\overline{\mu}}\mathbb{E}_{\pi\sim p_{\exp}}\left[ D_{\mathrm{H}}^2(\mathsf{P}(\pi),\overline{\mathsf{P}}(\pi)) \right] \Bigg\},$$

where (i) is due to strong duality Theorem B.1, in (ii) we upper bound $\inf_{p_{\mathrm{out}}}$ by letting $p_{\mathrm{out}}\in\Delta(\Pi)$ be defined by $p_{\mathrm{out}}(\pi) = \mu(\{\mathsf{P} : \pi_{\mathsf{P},R} = \pi\})$, and in (iii) we upper bound

$$\mathbb{E}_{\overline{\mathsf{P}}\sim\tilde{\mu}}\left[ f^{\mathsf{P},R}(\pi_{\mathsf{P},R}) - f^{\overline{\mathsf{P}},R}(\pi_{\mathsf{P},R}) \right] \leqslant \sup_{\pi'\in\Pi}\left| \mathbb{E}_{\overline{\mathsf{P}}\sim\tilde{\mu}}\left[ f^{\overline{\mathsf{P}},R}(\pi') - f^{\mathsf{P},R}(\pi') \right] \right|,$$
$$\mathbb{E}_{\mathsf{P}'\sim\mu}\mathbb{E}_{\overline{\mathsf{P}}\sim\tilde{\mu}}\left[ f^{\overline{\mathsf{P}},R}(\pi_{\mathsf{P}',R}) - f^{\mathsf{P},R}(\pi_{\mathsf{P}',R}) \right] \leqslant \sup_{\pi'\in\Pi}\left| \mathbb{E}_{\overline{\mathsf{P}}\sim\tilde{\mu}}\left[ f^{\overline{\mathsf{P}},R}(\pi') - f^{\mathsf{P},R}(\pi') \right] \right|.$$

Taking $\inf_{\tilde{\mu}}$ over $\tilde{\mu}\in\Delta(\mathcal{P})$ gives the desired result. $\square$

## J MODEL-ESTIMATION COEFFICIENT

Define

$$\mathrm{mdec}(\mathcal{P},\overline{\mu}) := \inf_{p_{\exp}\in\Delta(\Pi),\mu_{\mathrm{out}}\in\Delta(\mathcal{P})} \sup_{\mathsf{P}\in\mathcal{P},\overline{\pi}\in\Pi} \mathbb{E}_{\hat{\mathsf{P}}\sim\mu_{\mathrm{out}}}\left[ D_{\mathrm{TV}}\left( \mathsf{P}(\overline{\pi}),\hat{\mathsf{P}}(\overline{\pi}) \right) \right] - \gamma\mathbb{E}_{\pi\sim p_{\exp}}\mathbb{E}_{\overline{\mathsf{P}}\sim\overline{\mu}}\left[ D_{\mathrm{H}}^2(\mathsf{P}(\pi),\overline{\mathsf{P}}(\pi)) \right].$$
$$(51)$$

### J.1 ALGORITHM MODEL-EST E2D

---

**Algorithm 9** Model-EST E2D

---

1: **Input:** Parameters $\eta = 1/3$, $\gamma > 0$; prior distribution $\mu^1 \in \Delta(\mathcal{P})$.
2: **for** $t = 1, \ldots, T$ **do**
3:    Set $(p_{\exp}^t, \mu_{\text{out}}^t) = \arg\min_{(p_{\exp}, \mu_{\text{out}}) \in \Delta(\Pi) \times \Delta(\mathcal{P})} \widehat{V}_{\text{me},\gamma}^{\mu^t}(p_{\exp}, \mu_{\text{out}})$, where

$$\widehat{V}_{\text{me},\gamma}^{\mu^t}(p_{\exp}, \mu_{\text{out}}) := \sup_{\mathsf{P} \in \mathcal{P}} \sup_{\bar{\pi} \in \Pi} \mathbb{E}_{\overline{\mathsf{P}} \sim \mu_{\text{out}}} \left[ D_{\text{TV}} \left( \mathsf{P}(\bar{\pi}), \overline{\mathsf{P}}(\bar{\pi}) \right) \right] - \gamma \mathbb{E}_{\pi \sim p_{\exp}} \mathbb{E}_{\widehat{\mathsf{P}}^t \sim \mu^t} \left[ D_{\text{H}}^2(\mathsf{P}(\pi), \widehat{\mathsf{P}}^t(\pi)) \right].$$

4:    Sample $\pi^t \sim p_{\exp}^t$. Execute $\pi^t$ and observe $o^t$.
5:    Compute $\mu^{t+1} \in \Delta(\mathcal{P})$ by Tempered Aggregation with observations only:

$$\mu^{t+1}(\mathsf{P}) \propto_{\mathsf{P}} \mu^t(\mathsf{P}) \cdot \exp\left( \eta \log \widetilde{\mathsf{P}}^{\pi^t}(o^t) \right).$$

6: Compute $\mu_{\text{out}} = \frac{1}{T} \sum_{t=1}^T \mu_{\text{out}}^t \in \Delta(\mathcal{P})$.
7: **Output:** $\widehat{\mathsf{P}} = \arg\min_{\mathsf{P} \in \mathcal{P}} \sup_{\bar{\pi} \in \Pi} \mathbb{E}_{\overline{\mathsf{P}} \sim \mu_{\text{out}}} \left[ D_{\text{TV}} \left( \mathsf{P}(\bar{\pi}), \overline{\mathsf{P}}(\bar{\pi}) \right) \right]$

---

**Theorem J.1.** *Given a $\rho$-optimistic cover $(\widetilde{\mathsf{P}}, \mathcal{P}_0)$ of $\mathcal{P}$, we choose $\eta = 1/2$, $\mu^1 = \text{Unif}(\mathcal{P}_0)$, then Algorithm 9 achieves the following with probability at least $1 - \delta$:*

$$D_{\text{TV}}^{\Pi}\left( \widehat{\mathsf{P}}, \mathsf{P}^\star \right) := \max_{\bar{\pi} \in \Pi} D_{\text{TV}}\left( \widehat{\mathsf{P}}(\bar{\pi}), \mathsf{P}^\star(\bar{\pi}) \right) \leqslant 2\overline{\text{mdec}}_\gamma(\mathcal{M}) + \frac{4\gamma}{T}[\log|\mathcal{P}_0| + T\rho + 2\log(2/\delta)].$$

For $\mathsf{P} \in \mathcal{P}$, $\bar{\pi} \in \Pi$, $p_{\exp} \in \Delta(\Pi)$, $\mu_{\text{out}} \in \Delta(\mathcal{P})$, we consider the function

$$V^t(p_{\exp}, \mu_{\text{out}}; \mathsf{P}, \bar{\pi}) := \mathbb{E}_{\overline{\mathsf{P}} \sim \mu_{\text{out}}} \left[ D_{\text{TV}}\left( \mathsf{P}(\bar{\pi}), \overline{\mathsf{P}}(\bar{\pi}) \right) \right] - \gamma \mathbb{E}_{\pi \sim p_{\exp}} \mathbb{E}_{\widehat{\mathsf{P}}^t \sim \mu^t} \left[ D_{\text{H}}^2(\mathsf{P}(\pi), \widehat{\mathsf{P}}^t(\pi)) \right].$$

Then, using the definition of the infs and sups in the risks, we have that

$$\overline{\text{mdec}}_\gamma(\mathcal{M}) \geqslant \text{mdec}_\gamma(\mathcal{M}, \mu^t) = \widehat{V}_{\text{me},\gamma}^{\mu^t}(p_{\exp}^t, \mu_{\text{out}}^t) \geqslant \sup_{\mathsf{P} \in \mathcal{P}, \bar{\pi} \in \Pi} V^t(p_{\exp}^t, \mu_{\text{out}}^t; \mathsf{P}, \bar{\pi})$$

$$\geqslant \sup_{\bar{\pi} \in \Pi} V^t(p_{\exp}^t, \mu_{\text{out}}^t; \mathsf{P}^\star, \bar{\pi})$$

$$= \sup_{\bar{\pi} \in \Pi} \mathbb{E}_{\overline{\mathsf{P}} \sim \mu_{\text{out}}^t} \left[ D_{\text{TV}}\left( \mathsf{P}^\star(\bar{\pi}), \overline{\mathsf{P}}(\bar{\pi}) \right) \right] - \gamma \mathbb{E}_{\pi \sim p_{\exp}^t} \mathbb{E}_{\widehat{\mathsf{P}}^t \sim \mu^t} \left[ D_{\text{H}}^2(\mathsf{P}^\star(\pi), \widehat{\mathsf{P}}^t(\pi)) \right]$$

Taking average over $t \in [T]$ gives

$$\sup_{\bar{\pi} \in \Pi} \mathbb{E}_{\overline{\mathsf{P}} \sim \mu_{\text{out}}} \left[ D_{\text{TV}}\left( \mathsf{P}^\star(\bar{\pi}), \overline{\mathsf{P}}(\bar{\pi}) \right) \right] \leqslant \frac{1}{T} \sum_{t=1}^T \sup_{\bar{\pi} \in \Pi} \mathbb{E}_{\overline{\mathsf{P}} \sim \mu_{\text{out}}^t} \left[ D_{\text{TV}}\left( \mathsf{P}^\star(\bar{\pi}), \overline{\mathsf{P}}(\bar{\pi}) \right) \right] \qquad (52)$$

$$\leqslant \overline{\text{mdec}}_\gamma(\mathcal{M}) + \gamma \cdot \frac{\mathbf{Est}_{\text{H}}}{T}, \qquad (53)$$

where

$$\mathbf{Est}_{\text{H}} := \sum_{t=1}^T \mathbb{E}_{\overline{\mathsf{P}} \sim \mu^t} \mathbb{E}_{\pi \sim p_{\exp}^t} \left[ D_{\text{H}}^2(\mathsf{P}^\star(\pi), \overline{\mathsf{P}}(\pi)) \right].$$

By definition, it holds that

$$\sup_{\bar{\pi} \in \Pi} \mathbb{E}_{\overline{\mathsf{P}} \sim \mu_{\text{out}}} \left[ D_{\text{TV}}\left( \widehat{\mathsf{P}}(\bar{\pi}), \overline{\mathsf{P}}(\bar{\pi}) \right) \right] \leqslant \sup_{\bar{\pi} \in \Pi} \mathbb{E}_{\overline{\mathsf{P}} \sim \mu_{\text{out}}} \left[ D_{\text{TV}}\left( \mathsf{P}^\star(\bar{\pi}), \overline{\mathsf{P}}(\bar{\pi}) \right) \right],$$

and therefore,

$$\max_{\bar{\pi} \in \Pi} D_{\text{TV}}\left( \widehat{\mathsf{P}}(\bar{\pi}), \mathsf{P}^\star(\bar{\pi}) \right) \leqslant \sup_{\bar{\pi} \in \Pi} \left\{ \mathbb{E}_{\overline{\mathsf{P}} \sim \mu_{\text{out}}} \left[ D_{\text{TV}}\left( \widehat{\mathsf{P}}(\bar{\pi}), \overline{\mathsf{P}}(\bar{\pi}) \right) \right] + \sup_{\bar{\pi} \in \Pi} \mathbb{E}_{\overline{\mathsf{P}} \sim \mu_{\text{out}}} \left[ D_{\text{TV}}\left( \mathsf{P}^\star(\bar{\pi}), \overline{\mathsf{P}}(\bar{\pi}) \right) \right] \right\}$$

$$\leqslant 2 \sup_{\bar{\pi} \in \Pi} \mathbb{E}_{\overline{\mathsf{P}} \sim \mu_{\text{out}}} \left[ D_{\text{TV}}\left( \mathsf{P}^\star(\bar{\pi}), \overline{\mathsf{P}}(\bar{\pi}) \right) \right]$$

$$\leqslant 2\overline{\mathrm{mdec}}_\gamma(\mathcal{M}) + 2\gamma \cdot \frac{\mathbf{Est}_\mathrm{H}}{T}.$$

Note that Theorem D.8 implies that: subroutine (48) (agrees with (19) with $\eta_\mathrm{r} = 0$ and $\eta_\mathrm{p} = \eta$) achieves with probability at least $1 - \delta$ that

$$\mathbf{Est}_\mathrm{H} \leqslant \frac{1}{\eta}[\log|\mathcal{P}_0| + T\rho + 2\log(2/\delta)].$$

This completes the proof of Theorem J.1. $\hfill\square$

## K    PROOFS FOR SECTION 5

This section provides the proofs for Section 5 along with some additional discussions, organized as follows. We begin by presenting some useful intermediate results in Appendix K.1 for proving Proposition 12; The proof of Proposition 12 then follows by combining several statements therein (see Appendix K.2). The proofs of Example 13-15 are provided in Appendix K.3. Appendix K.4 presents some discussions regarding the definition of Bellman representability (compared with (Foster et al., 2021)) along with some useful results regarding the complexities of general function classes. Finally, unless otherwise specified, the proofs of all new results in this section are presented in Appendix K.5.

### K.1    INTERMEDIATE RESULTS ON BELLMAN REPRESENTABILITY

**Complexity measures for general function classes**    We begin by introducing the concept of decoupling coefficient for a (general) function class, which acts as a convenient interface for both bounding the DECs and proving the examples.

**Definition K.1** (Decoupling coefficient). *Given a function class $\mathcal{F} \subset (\mathcal{X} \to \mathbb{R})$, the decoupling coefficient $\mathrm{dc}(\mathcal{F}, \gamma)$ is defined as*

$$\mathrm{dc}(\mathcal{F}, \gamma) := \sup_{\nu \in \Delta(\mathcal{F} \times \mathcal{X})} \mathbb{E}_{(f,x)\sim\nu}[|f(x)|] - \gamma \mathbb{E}_{f\sim\nu}\mathbb{E}_{x\sim\nu}\Big[|f(x)|^2\Big].$$

As examples, the decoupling coefficient can be bounded for linear function classes, and more generally any function class with low Eluder dimension (Russo & Van Roy, 2013) or star number (Foster et al., 2020).

**Example K.2.** Suppose that there exists $\phi : \mathcal{X} \to \mathbb{R}^d$ such that $\mathcal{F} \subset \{f_\theta : x \to \langle \theta, \phi(x) \rangle\}_{\theta \in \mathbb{R}^d}$, then $\mathrm{dc}(\mathcal{F}, \gamma) \leqslant d/(4\gamma)$.

**Definition K.3** (Eluder dimension). *The eluder dimension $\mathfrak{e}(\mathcal{F}, \Delta)$ is the maximum of the length of sequence $(f_1, \pi_1), \cdots, (f_n, \pi_n) \in \mathcal{F} \times \Pi$ such that there is a $\Delta' \geqslant \Delta$, and*

$$|f_i(\pi_i)| \geqslant \Delta', \qquad \sum_{j<i} |f_i(\pi_j)|^2 \leqslant (\Delta')^2, \qquad \forall i.$$

**Definition K.4** (Star number). *The star number $\mathfrak{s}(\mathcal{F}, \Delta)$ is the maximum of the length of sequence $(f_1, \pi_1), \cdots, (f_n, \pi_n) \in \mathcal{F} \times \Pi$ such that there is a $\Delta' \geqslant \Delta$, and*

$$|f_i(\pi_i)| \geqslant \Delta', \qquad \sum_{j \neq i} |f_i(\pi_j)|^2 \leqslant (\Delta')^2, \qquad \forall i.$$

**Example K.5.** When $\mathcal{F} \subset (\Pi \to [-1, 1])$ and $\gamma \geqslant e$, it holds that

$$\mathrm{dc}(\mathcal{F}, \gamma) \leqslant 24 \inf_{\Delta > 0} \left\{ \frac{\min\{\mathfrak{s}^2(\mathcal{F}, \Delta), \mathfrak{e}(\mathcal{F}, \Delta)\} \log^2(\gamma)}{\gamma} + \Delta \right\}.$$

More generally, the decoupling coefficient can be bounded by the *disagreement coefficient* introduced in (Foster et al., 2021, Definition 6.3). The proof of Example K.5 along with some further discussions can be found in Appendix K.4.

**Bounding DEC/EDEC by decoupling coefficient of** $\mathcal{M}$ We now state our main intermediate result for bounding the DEC/EDEC for any $\mathcal{M}$ admitting a low-complexity Bellman representation, in the sense of a bounded decoupling coefficient.

**Proposition K.6** (Bounding DEC/EDEC by decoupling coefficient of Bellman representation)**.** *Suppose that* $\mathcal{G} = (\mathcal{G}_h^{\overline{M}})_{\overline{M} \in \mathcal{M}, h \in [H]}$ *is a Bellman representation of* $\mathcal{M}$*. For any* $\overline{M} \in \mathcal{M}$*, we define*

$$\mathrm{comp}(\mathcal{G}^{\overline{M}}, \gamma) := H \max_h \mathrm{dc}(\mathcal{G}_h^{\overline{M}}, \gamma/12HL^2) + \frac{6H}{\gamma}, \tag{54}$$

*and let* $\mathrm{comp}(\mathcal{G}, \gamma) := \max_{\overline{M} \in \mathcal{M}} \mathrm{comp}(\mathcal{G}^{\overline{M}}, \gamma)$*. Then we have for any* $\gamma > 0$ *that*

*(1) If* $\pi_{M,h}^{\mathrm{est}} = \pi_M$ *(the on-policy case), we have* $\overline{\mathrm{dec}}_\gamma(\mathcal{M}) \leqslant \mathrm{comp}(\mathcal{G}, \gamma)$ *and* $\mathrm{psc}_\gamma(\mathcal{M}, M^\star) \leqslant \mathrm{comp}(\mathcal{G}^{M^\star}, \gamma)$ *for all* $M^\star \in \mathcal{M}$*.*
*(2) For* $\mathcal{G}$ *with general estimation policies, we have* $\overline{\mathrm{edec}}_\gamma(\mathcal{M}) \leqslant \mathrm{comp}(\mathcal{G}, \gamma)$*.*

Similarly, we show that the MLEC can also be bounded in terms of the Eluder dimension of $\mathcal{G}^{M^\star}$.

**Proposition K.7.** *Suppose that* $\mathcal{G} = (\mathcal{G}_h^{\overline{M}})_{\overline{M} \in \mathcal{M}, h \in [H]}$ *is a Bellman representation of* $\mathcal{M}$ *in the on-policy case* $(\pi_{M,h}^{\mathrm{est}} = \pi_M$ *for all* $(h, M))$*. Then we have for any* $\gamma > 0, K \in \mathbb{Z}_{\geqslant 1}$ *that*

$$\mathrm{mlec}_{\gamma,K}(\mathcal{M}, M^\star) \leqslant CH^2L^2 \inf_{\Delta > 0} \left\{ \frac{\max_{h \in [H]} \mathfrak{e}(\mathcal{G}_h^{M^\star}, \Delta)}{\gamma} + \Delta \right\},$$

*where* $C > 0$ *is an absolute constant.*

**Bounding RFDEC under strong Bellman representability** We define a *strong Bellman representation* as follows.

**Definition K.8** (Strong Bellman representation)**.** *Given a pair* $(\mathcal{M} = \mathcal{P} \times \mathcal{R}, \overline{\mathsf{P}} \in \mathcal{P})$*, its strong Bellman representation is a collection of function classes* $\mathcal{G}^{\overline{M}} = \{\mathcal{G}_h^{\overline{M}}\}_{h \in [H]}$ *with* $\mathcal{G}_h^{\overline{M}} = \{g^{M;\overline{\mathsf{P}}} : \Pi \to [-1, 1]\}_{M \in \mathcal{M}}$ *such that:*

*1. For* $M = (\mathsf{P}, R) \in \mathcal{M}, \pi \in \Pi$*, it holds that for* $\overline{M} = (\overline{\mathsf{P}}, R)$*,*

$$\left| \mathbb{E}^{\overline{M}, \pi} \left[ Q_h^{M, \pi}(s_h, a_h) - r_h - V_{h+1}^{M, \pi}(s_{h+1}) \right] \right| \leqslant \left| g_h^{M;\overline{\mathsf{P}}}(\pi) \right|.$$

*2. For* $M = (\mathsf{P}, R) \in \mathcal{M}, \pi \in \Pi$*, it holds that*

$$\left| g_h^{M;\overline{\mathsf{P}}}(\pi) \right| \leqslant LD_{\mathrm{H}}(\mathsf{P}(\pi_h^{\mathrm{est}}), \overline{\mathsf{P}}(\pi_h^{\mathrm{est}})),$$

*for some constant* $L \geqslant 1$*.*

Given a strong Bellman representation $\mathcal{G}$ of $\mathcal{M}$, the RFDEC of $\mathcal{M}$ can be bounded in terms of certain complexity measure of $\mathcal{G}$ as follows.

**Proposition K.9** (Bounding RFDEC by decoupling coefficient of strong Bellman representation)**.** *Suppose* $\mathcal{G}$ *is a strong Bellman representation of* $\mathcal{M}$*. Then for* $\overline{\mu} \in \Delta(\mathcal{P})$*,*

$$\overline{\mathrm{rfdec}}_\gamma(\mathcal{M}, \overline{\mu}) \leqslant 2H \cdot \max_{\overline{\mathsf{P}} \in \mathcal{P}} \max_{h \in [H]} \mathrm{dc}\left( \mathcal{G}_h^{\overline{\mathsf{P}}}, \gamma/4HL^2 \right) + \frac{H}{\gamma}.$$

The proof of Proposition K.9 can be found in Appendix K.5.4. For its applications, see Appendix K.3.4.

## K.2  PROOF OF PROPOSITION 12

The claims about bounded DEC/EDEC/RFDEC in (1)-(3) follows by combining the bounds in terms of the decoupling coefficients (Proposition K.6(1), Proposition K.6(2), and Proposition K.9) with the bound of decoupling coefficient by the Eluder dimension/star number (Example K.5). Then, the claims about the sample complexities of the E2D algorithms follow by applying Theorem 2, Theorem 5, and Theorem 9 and optimizing $\gamma > 0$, respectively. $\qquad \square$

Proposition 12 can be directly extended to the more general case with infinite model classes, by using the TEMPERED AGGREGATION WITH COVERING subroutine (19) (and (48) for reward-free learning) in the corresponding E2D algorithms (see Theorem D.7, Theorem H.1 and Theorem I.1). We summarize this in the following proposition.

**Proposition K.10** (Variant of Proposition 12 with covering). *Suppose $\mathcal{M}$ admits a Bellman representation $\mathcal{G}$ with low complexity:* $\min\{\mathfrak{e}(\mathcal{G}_h^{\overline{M}}, \Delta), \mathfrak{s}(\mathcal{G}_h^{\overline{M}}, \Delta)^2\} \leqslant \widetilde{\mathcal{O}}(d)$, *where $\widetilde{\mathcal{O}}(\cdot)$ contains possibly $\mathrm{polylog}(1/\Delta)$ factors. Further assume that $\mathcal{M}$ and $\mathcal{P}$ admits optimistic covers with bounded covering numbers:* $\log \mathcal{N}(\mathcal{M}, \rho) \leqslant \widetilde{\mathcal{O}}(\dim(\mathcal{M}))$ *and* $\log \mathcal{N}(\mathcal{P}, \rho) \leqslant \widetilde{\mathcal{O}}(\dim(\mathcal{P}))$ *for any $\rho > 0$, where $\dim(\mathcal{M}), \dim(\mathcal{P}) > 0$ and $\widetilde{\mathcal{O}}(\cdot)$ contains possibly $\mathrm{polylog}(1/\rho)$ factors. Then with subroutines changed correspondingly to the versions with covering, we have*

*(1)* **(No-regret learning)** *If $\pi_{M,h}^{\mathrm{est}} = \pi_M$ for all $M \in \mathcal{M}$ (the on-policy case), then Algorithm E2D-TA achieves* $\mathbf{Reg_{DM}} \leqslant \widetilde{\mathcal{O}}(HL\sqrt{d \cdot \dim(\mathcal{M})T})$.

*(2)* **(PAC learning)** *For any general $\{\pi_{M,h}^{\mathrm{est}}\}_{M \in \mathcal{M}, h \in [H]}$, Algorithm EXPLORATIVE E2D achieves* $\mathbf{SubOpt} \leqslant \varepsilon$ *within* $\widetilde{\mathcal{O}}(d \cdot \dim(\mathcal{M})H^2L^2/\varepsilon^2)$ *episodes of play.*

*(3)* **(Reward-free learning)** *If $\mathcal{G}$ is a strong Bellman representation, then Algorithm REWARD-FREE E2D achieves* $\mathbf{SubOpt_{rf}} \leqslant \varepsilon$ *within* $\widetilde{\mathcal{O}}(d \cdot \dim(\mathcal{P})H^2L^2/\varepsilon^2)$ *episodes of play.*

### K.3 PROOF OF EXAMPLES

We first present some definitions and properties as preparations. The proofs of Example 13-15 are then presented in Appendix K.3.1-K.3.3 for the regret/PAC bounds, and in Appendix K.3.4 for the reward-free bounds.

**Definition K.11** (Model-based bilinear class, Jin et al. (2021); Foster et al. (2021)). *$\mathcal{G}$ is a bilinear Bellman representation of rank $d$ if there exists maps $X_h(\cdot; \cdot) : \mathcal{M} \times \mathcal{M} \to \mathbb{R}^d$, $W_h(\cdot; \cdot) : \mathcal{M} \times \mathcal{M} \to \mathbb{R}^d$, such that*

$$g_h^{M';\overline{M}}(M) = \langle X_h(M; \overline{M}), W_h(M'; \overline{M}) \rangle, \qquad \forall M, M', \overline{M} \in \mathcal{M}.$$

*Suppose that $\mathcal{G}$ is bilinear with rank d, then by Example K.2,*

$$\mathrm{comp}(\mathcal{G}, \gamma) \leqslant \frac{2H^2L^2 + 2H + 2}{\gamma}.$$

*Similarly, by Proposition K.7 we have* $\mathrm{mlec}_{\gamma,K}(\mathcal{M}) \leqslant \widetilde{\mathcal{O}}(dH^2L^2/\gamma)$.

**Example K.12** (Bellman error as Bellman representation). For a model class $\mathcal{M}$, its Q-type Bellman error is defined as

$$g_h^{M';\overline{M}}(M) := \mathbb{E}^{\overline{M}, \pi_M}\Big[Q^{M',\star}(s_h, a_h) - r_h - V_{h+1}^{M',\star}(s_{h+1})\Big], \qquad \forall M, M', \overline{M} \in \mathcal{M}.$$

Along with $\pi^{\mathrm{est}} = \pi$ and $L = \sqrt{2}$, it gives a Bellman representation of $\mathcal{M}$, which we term as its *QBE*.

Similarly, we can consider the V-type Bellman error

$$g_h^{M';\overline{M}}(M) := \mathbb{E}^{\overline{M}, \pi_M \circ_h \pi_{M'}}\Big[Q^{M',\star}(s_h, a_h) - r_h - V_{h+1}^{M',\star}(s_{h+1})\Big], \qquad \forall M, M', \overline{M} \in \mathcal{M},$$

where $\pi_M \circ_h \pi_{M'}$ stands for the policy which executes $\pi_M$ for the first $h-1$ steps, and then follows $\pi_{M'}$ from the $h$-th step. Along with $\pi_h^{\mathrm{est}} = \pi \circ_h \mathrm{Unif}(\mathcal{A})$ and $L = \sqrt{2A}$, it gives a Bellman representation of $\mathcal{M}$, which we term as its *VBE*.

**Relation to Model-based Bellman-Eluder dimension**  Take Q-type Bellman error for example. Note that the argument of $g_h^{M';\overline{M}}(M)$ corresponds to the roll-in policy $\pi_M$. Therefore, one can check that

$$\mathfrak{e}(\mathcal{G}_h^{M^\star}, \Delta) = \dim_{\mathrm{DE}}(\mathcal{E}_h, \Pi_h, \Delta),$$

where $\mathcal{E}_h = \left\{Q_h^{M,\star} - r_h - \mathbb{P}_h^\star V_{h+1}^{M,\star}\right\}_{M \in \mathcal{M}}$ is the model-induced Bellman residual function class, $\Pi_h = \left\{d_h^{M^\star, \pi}\right\}_{\pi \in \Pi}$ is the collection of distributions over $\mathcal{S} \times \mathcal{A}$ at $h$-step induced by policies. Therefore, $\mathfrak{e}(\mathcal{G}^{M^\star})$ is indeed equivalent to the model-based version of the Q-type (model-induced) Bellman Eluder dimension (Jin et al., 2021).

### K.3.1 PROOF OF EXAMPLE 13, REGRET BOUND

Clearly, for the tabular MDP model class $\mathcal{M}$ of $S$ states and $A$ actions, its estimation complexity $\log \mathcal{N}(\mathcal{M}, \rho) = \widetilde{\mathcal{O}}\left(S^2 A H\right)$ (see Example K.13) and its QBE is bilinear with rank $SA$. Thus by the definition of $\mathrm{comp}$ in (54) and Example K.2, we have $\mathrm{comp}(\mathcal{G}, \gamma) = \mathcal{O}\left(SAH^2/\gamma\right)$, and further by Proposition K.6(1) and Proposition K.10, E2D-TA achieves $\mathbf{Reg_{DM}} \leqslant \widetilde{\mathcal{O}}\left(\sqrt{S^3 A^2 H^3 T}\right)$ regret. $\qquad\square$

We remark that the same regret bound also holds for MOPS and OMLE, by the PSC bound in Proposition K.6(1) and Proposition K.7 combined with Theorem F.1 and Theorem G.3.

In the following, we demonstrate briefly how to construct an optimistic covering of the class of tabular MDPs. Without loss of generality, we only cover the class of transition dynamic P.

**Example K.13** (Optimistic covering of tabular MDP)**.** Consider $\mathcal{M}$, the class of MDPs with $S$ states, $A$ actions, $H$ steps. Fix a $\rho_1 \in (0, 1]$, and $\rho = \rho_1^2/eHS$. For $M \in \mathcal{M}$, we compute its $\rho_1$-optimistic likelihood function as follows: define

$$\widetilde{\mathbb{P}}_h^M(s'|s, a) := \rho \left\lceil \frac{1}{\rho} \mathbb{P}_h^M(s'|s, a) \right\rceil, \qquad \widetilde{\mathbb{P}}_1^M(s) := \rho \left\lceil \frac{1}{\rho} \mathbb{P}_1^M(s) \right\rceil, \qquad (55)$$

and for Markov policy $\pi$, let

$$\widetilde{\mathbb{P}}^{M,\pi}(s_1, a_1, \cdots, s_H, a_H) := \widetilde{\mathbb{P}}_1^M(s_1)\pi_1(a_1|s_1)\widetilde{\mathbb{P}}_1^M(s_2|s_1, a_1)\cdots\widetilde{\mathbb{P}}_{H-1}^M(s_H|s_{H-1}, a_{H-1})\pi(a_H|s_H)$$

$$= \widetilde{\mathbb{P}}_1^M(s_1) \times \prod_{h=1}^{H} \pi_h(a_h|s_h) \times \prod_{h=1}^{H-1} \widetilde{\mathbb{P}}^M(s_{h+1}|s_h, a_h).$$

A direct calculation shows that $\widetilde{\mathbb{P}}^{M,\pi} \geqslant \mathbb{P}^{M,\pi}$ for all $\pi$, and $\|\widetilde{\mathbb{P}}^{M,\pi}(\cdot) - \mathbb{P}^{M,\pi}(\cdot)\|_1 \leqslant \rho_1^2$. Clearly, there are at most $\lceil 1/\rho \rceil^{S^2 AH}$ different optimistic likelihood functions defined by (55), and we can form $\mathcal{M}_0$ by picking a representative in $\mathcal{M}$ for each optimistic likelihood function (if possible). Then, $\log |\mathcal{M}_0| = \mathcal{O}\left(S^2 AH \log(SH/\rho_1)\right)$.

### K.3.2 PROOF OF EXAMPLE 14, REGRET BOUND

We follow the commonly used definition of linear mixture MDPs (Chen et al., 2021), which is slightly more general than the one in Ayoub et al. (2020).

**Definition K.14** (Linear mixture MDPs)**.** *A MDP is called a linear mixture MDP (of rank $d$) if there exists feature maps $\phi_h(\cdot|\cdot, \cdot) : \mathcal{S} \times \mathcal{S} \times \mathcal{A} \to \mathbb{R}^d$ and parameter $(\theta_h)_h \subset \mathbb{R}^d$, such that $\mathbb{P}_h(s'|s, a) = \langle \theta_h, \phi_h(s'|s, a) \rangle$. We further assume that $\|\theta_h\|_2 \leqslant B$ for all $h \in [H]$, and $\|\sum_{s'} \phi_h(s'|s, a)V(s')\|_2 \leqslant 1$ for all $V : \mathcal{S} \to [0, 1]$ and tuple $(s, a, h) \in \mathcal{S} \times \mathcal{A} \times [H]$.*

Suppose that $\mathcal{M}$ is a linear mixture MDP model with the given feature map $\phi$. Then the definition above directly yields the QBE of $\mathcal{M}$ is bilinear with rank $d$. Therefore, as long as $\log \mathcal{N}(\mathcal{M}, \rho) = \widetilde{\mathcal{O}}(dH)$, by Proposition K.10 we can obtain $\widetilde{\mathcal{O}}\left(d\sqrt{H^3 T}\right)$ regret of E2D-TA as claimed.

The following proposition provides an upper bound on $\mathrm{est}(\mathcal{M})$ via a concrete construction. For the simplicity of discussion, we assume that the initial distribution is known, and we also assume the mean reward function is known for no-regret and PAC learning setting.

**Proposition K.15** (Optimistic covering for linear mixture MDPs)**.** *Given feature map $\phi$ of dimension $d$ and constant $B$. Suppose that $\mathcal{M}$ consists of linear mixture MDPs with feature map $\phi$ and parameter bounded by $B$ (without reward component). Then for $\rho$, there exists a $\rho$-optimistic covering $(\widetilde{\mathbb{P}}, \mathcal{M}_0)$ with $\log |\mathcal{M}_0| = \widetilde{\mathcal{O}}(dH)$.*

$\qquad\square$

### K.3.3 PROOF OF EXAMPLE 15, PAC BOUND

We consider the broader class of MDPs with low occupancy rank (Du et al., 2021):

**Definition K.16** (Occupancy rank)**.** *We say a MDP is of occupancy rank $d$ if for all $h \in [H]$, there exists map $\phi_h : \Pi \to \mathbb{R}^d$, $\psi_h : \mathcal{S} \to \mathbb{R}^d$, such that*

$$\mathbb{P}^\pi(s_h = s) = \langle \psi_h(s), \phi_h(\pi) \rangle, \qquad \forall s \in \mathcal{S}, \pi \in \Pi.$$

By definition, low-rank MDP with rank $d$ is of occupancy rank $d$. Furthermore, for model class $\mathcal{M}$ consisting of MDPs with occupancy rank $d$, its VBE is bilinear with rank $d$. Therefore, EXPLO-RATIVE E2D achieves $\mathbf{SubOpt} = \widetilde{\mathcal{O}}\left(H\sqrt{dA\log|\mathcal{M}|/T}\right)$ on such model class as claimed. □

### K.3.4 PROOF OF REWARD-FREE BOUNDS IN EXAMPLE 13-15

**Example K.17** (Model estimation error)**.** For model class $\mathcal{M} = \mathcal{P} \times \mathcal{R}$, its Q-type estimation error is defined as

$$g_h^{\mathsf{P};\overline{\mathsf{P}}}(\pi) := \mathbb{E}^{\overline{\mathsf{P}},\pi}\left[D_{\mathrm{TV}}\left(\mathbb{P}_h^{\mathsf{P}}(\cdot|s_h,a_h), \mathbb{P}_h^{\overline{\mathsf{P}}}(\cdot|s_h,a_h)\right)\right].$$

Along with $\pi^{\mathrm{est}} = \pi$ and $L = \sqrt{2}$, it gives a strong Bellman representation of $\mathcal{M}$, which we term as its *QER*.

Similarly, we can consider the V-type estimation error

$$g_h^{\mathsf{P};\overline{\mathsf{P}}}(\pi) := \mathbb{E}^{\overline{\mathsf{P}},\pi}\left[\max_{a\in\mathcal{A}} D_{\mathrm{TV}}\left(\mathbb{P}_h^{\mathsf{P}}(\cdot|s_h,a), \mathbb{P}_h^{\overline{\mathsf{P}}}(\cdot|s_h,a)\right)\right].$$

Along with $\pi_h^{\mathrm{est}} = \pi \circ_h \mathrm{Unif}(\mathcal{A})$ and $L = \sqrt{2A}$, it gives a strong Bellman representation of $\mathcal{M}$, which we term as its *VER*.

As the following proposition indicates, the two choices of strong Bellman representation above are enough for us to bound the RFDEC for tabular MDP, linear mixture MDP and MDP with low occupancy complexity. The proof of Proposition K.18 is mainly based on the decoupling behavior of linear function classes (cf. Example K.2), and can be found in Appendix K.5.6.

**Proposition K.18.** *For model class $\mathcal{M}$ of linear mixture MDPs (of a given feature $\phi$), by its QER $\mathcal{G}$ and Proposition K.9, we have $\overline{\mathrm{rfdec}}_\gamma(\mathcal{M}) \leqslant 8dH^2/\gamma$.*

*For model class $\mathcal{M}$ of MDPs with occupancy rank $d$, because its VER $\mathcal{G}$ is bilinear, we have $\overline{\mathrm{rfdec}}_\gamma(\mathcal{M}) \leqslant 8dAH^2/\gamma$.*

Combining Proposition K.18 with Theorem I.1 gives that:

1. For tabular MDPs, REWARD-FREE E2D achieves $\mathbf{SubOpt_{rf}} = \widetilde{\mathcal{O}}\left(\sqrt{S^3A^2H^3/T}\right)$;

2. For linear mixture MDPs, REWARD-FREE E2D achieves $\mathbf{SubOpt_{rf}} = \widetilde{\mathcal{O}}\left(\sqrt{dH^3/T}\right)$;

3. For MDPs with occupancy rank at most $d$ (including low-rank MDPs with rank $d$), REWARD-FREE E2D achieves $\mathbf{SubOpt_{rf}} = \widetilde{\mathcal{O}}\left(\sqrt{dAH^2\log|\mathcal{P}|/T}\right)$.

□

### K.4 DISCUSSIONS ABOUT BELLMAN REPRESENTATIONS AND DECOUPLING COEFFICIENT

**Difference in definition of Bellman representation** Our Definition 11 is slightly different from Foster et al. (2021, Definition F.1): They define the function $g_h^{M';\overline{M}}(M)$ in terms of the discrepancy function $\ell_M(M'; s_h, a_h, r_h, s_{h+1})$, while we only require $g_h^{M';\overline{M}}(M)$ can be upper bound by $D_{\mathrm{RL}}$. In general, $g_h^{M';\overline{M}}(M)$ they define can be only upper bound by the Hellinger distance in the full observation $(o, \mathbf{r})$ (which is in general larger than $D_{\mathrm{RL}}$), as the expected discrepancy function may depend on distributional information about the reward that is not captured by the mean. However, when the reward $\mathbf{r}$ is included in the observation $o$, our definition is more general than theirs. More importantly, for the majority of known concrete examples, e.g. those in Du et al. (2021); Jin et al. (2021), the discrepancy function is linear in $r_h$, and hence its expectation can be upper bound by $D_{\mathrm{RL}}$.

**Complexity measure for Bellman representations** In Foster et al. (2021), the complexity of a Bellman representation is measured in terms of *disagreement coefficient*, which can be upper bounded by eluder dimension or star number.

**Definition K.19.** *The disagreement coefficient of a function class $\mathcal{F} \subset (\Pi \to [-1, 1])$ is defined as*

$$\boldsymbol{\theta}\left(\mathcal{F}, \Delta_0, \varepsilon_0; \rho\right) = \sup_{\Delta \geqslant \Delta_0, \varepsilon \geqslant \varepsilon_0} \left\{ \frac{\Delta^2}{\varepsilon^2} \cdot \mathbb{P}_{\pi \sim \rho}\left(\exists f \in \mathcal{F} : |f(\pi)| > \Delta, \mathbb{E}_{\pi \sim \rho}\left[f^2(\pi)\right] \leqslant \varepsilon^2\right) \right\} \vee 1.$$

*By Foster et al. (2021, Lemma 6.1), for $\Delta, \varepsilon > 0, \rho \in \Delta(\Pi)$, it holds that*

$$\boldsymbol{\theta}(\mathcal{F}, \Delta, \varepsilon; \rho) \leqslant 4 \min\left\{\mathfrak{s}^2(\mathcal{F}, \Delta), \mathfrak{e}(\mathcal{F}, \Delta)\right\}.$$

It turns out that our decoupling coefficient can be upper bound by the disagreement coefficient: applying Foster et al. (2021, Lemma E.3), we directly obtain the following result.

**Lemma K.20.** *For function class $\mathcal{F} \subset (\Pi \to [-1, 1])$, we have*

$$\mathrm{dc}(\mathcal{F}, \gamma) \leqslant \inf_{\Delta > 0} \left\{ 2\Delta + 6 \frac{\boldsymbol{\theta}\left(\mathcal{F}, \Delta, \gamma^{-1}\right) \log^2(\gamma \vee e)}{\gamma} \right\},$$

*where $\boldsymbol{\theta}\left(\mathcal{F}, \Delta, \gamma^{-1}\right) := \sup_{\rho \in \Delta(\Pi)} \boldsymbol{\theta}(\mathcal{F}, \Delta, \varepsilon; \rho)$.*

Lemma K.20 also gives Example K.5 directly.

### K.5 PROOF OF PROPOSITIONS

#### K.5.1 PROOF OF EXAMPLE K.2

Under the linearity assumption, we can consider $f \mapsto \theta_f \in \mathbb{R}^d$ such that $f(x) = \langle \theta_f, \phi(x) \rangle \, \forall x \in \mathcal{X}$.

Given a $\nu \in \Delta(\mathcal{F} \times \mathcal{X})$, let us set $\Phi_\lambda := \lambda I_d + \mathbb{E}_{x \sim \nu}\left[\phi(x)\phi(x)^\top\right]$ for $\lambda > 0$. Then

$$\mathbb{E}_{(f,x) \sim \nu}\left[|f(x)|\right] \leqslant \mathbb{E}_{(f,x) \sim \nu}\left[\|\theta_f\|_{\Phi_\lambda} \|\phi(x)\|_{\Phi_\lambda^{-1}}\right] \leqslant \gamma \mathbb{E}_{f \sim \nu}\left[\|\theta_f\|_{\Phi_\lambda}^2\right] + \frac{1}{4\gamma} \mathbb{E}_{x \sim \nu}\left[\|\phi(x)\|_{\Phi_\lambda^{-1}}^2\right].$$

For the first term, we have

$$\begin{aligned}
\mathbb{E}_{f \sim \nu}\left[\|\theta_f\|_{\Phi_\lambda}^2\right] &= \mathbb{E}_{f \sim \nu}\left[\theta_f^\top \left(\mathbb{E}_{x \sim \nu}\left[\phi(x)\phi(x)^\top\right]\right)\theta_f\right] + \lambda \mathbb{E}_{f \sim \nu}\|\theta_f\|^2 \\
&= \mathbb{E}_{f \sim \nu}\mathbb{E}_{x \sim \nu}\left[|f(x)|^2\right] + \lambda \mathbb{E}_{f \sim \nu}\|\theta_f\|^2.
\end{aligned}$$

For the second term, we have

$$\begin{aligned}
\mathbb{E}_{x \sim \nu}\left[\|\phi(x)\|_{\Phi_\lambda^{-1}}^2\right] &= \mathbb{E}_{x \sim \nu}\left[\mathrm{tr}\left(\Phi_\lambda^{-1/2}\phi(x)\phi(x)^\top \Phi_\lambda^{-1/2}\right)\right] \\
&= \mathrm{tr}\left(\Phi_\lambda^{-1/2}\mathbb{E}_{x \sim \nu}\left[\phi(x)\phi(x)^\top\right]\Phi_\lambda^{-1/2}\right) \\
&= \mathrm{tr}\left(\Phi_\lambda^{-1/2}\Phi_0 \Phi_\lambda^{-1/2}\right) \leqslant d.
\end{aligned}$$

Letting $\lambda \to 0^+$ and then taking $\inf_\nu$ completes the proof. □

As a corollary, we have the following result.

**Corollary K.21.** *Suppose that there exists $\phi = (\phi_i : \mathcal{X} \to \mathbb{R}^d)_{i \in \mathcal{I}}$ such that*

$$\mathcal{F} \subset \left\{ f_\theta : x \to \max_i |\langle \theta, \phi_i(x) \rangle| \right\}_{\theta \in \mathbb{R}^d},$$

*then $\mathrm{dc}(\mathcal{F}, \gamma) \leqslant d/\gamma$.*

Corollary K.21 can be obtained similarly to the proof above of Example K.2. However, we believe the following fact is important:

**Proposition K.22.** *Suppose that $\mathcal{W} \subset (\mathcal{X} \times \mathcal{Y} \to \mathbb{R}_{\geqslant 0})$, then for the function class $\mathcal{F}$ defined by*

$$\mathcal{F} := \left\{ f_w : x \mapsto \max_{y \in \mathcal{Y}} w(x, y) \right\}_{w \in \mathcal{W}},$$

*we have $\mathrm{dc}(\mathcal{F}, \gamma) \leqslant \mathrm{dc}(\mathcal{W}, \gamma)$.*

Combining Proposition K.22 and Example K.2 gives Corollary K.21 directly.

**Proof of Proposition K.22**    Fix a $\nu \in \Delta(\mathcal{F} \times \mathcal{X})$. Consider the map $W : \mathcal{F} \to \mathcal{W}$ so that $f(x) = \max_y W(f)(x, y)$ for all $x \in \mathcal{X}$. Further consider $Y : \mathcal{F} \times \mathcal{X} \to \mathcal{Y}$ defined as $Y(f, x) := \arg\max_{y \in \mathcal{Y}} W(f)(x, y)$ (break ties arbitrarily). Then we let $\nu' \in \Delta(\mathcal{F} \times \mathcal{X} \times \mathcal{Y})$ given by $\nu'((w, x, y)) = \nu(\{(f, x) : W(f) = w, y = Y(f, x)\})$, and

$$
\begin{aligned}
\mathbb{E}_{(f,x)\sim\nu}[|f(x)|] &= \mathbb{E}_{(w,x,y)\sim\nu'}[w(x, y)] \\
&\leqslant \mathrm{dc}(\mathcal{W}, \gamma) + \mathbb{E}_{w\sim\nu'}\mathbb{E}_{(x,y)\sim\nu'}\left[w(x, y)^2\right] \\
&\leqslant \mathrm{dc}(\mathcal{W}, \gamma) + \mathbb{E}_{w\sim\nu'}\mathbb{E}_{x\sim\nu'}\left[|f_w(x)|^2\right] \\
&= \mathrm{dc}(\mathcal{W}, \gamma) + \mathbb{E}_{f\sim\nu}\mathbb{E}_{x\sim\nu}\left[|f(x)|^2\right],
\end{aligned}
$$

where the second inequality is due to the definition $f_w(x) = \max_y w(x, y)$, and the last equality is due to the fact that the marginalization of $\nu'$ to $\mathcal{X}$ agrees with marginalization of $\nu$ to $\mathcal{X}$, and for $f \in \mathcal{F}$, $\nu'(\{w : f_w = f\}) = \nu(\{(f', x) : W(f') = w, f_w = f\}) = \nu(f)$. Taking $\inf_\nu$ completes the proof. $\qquad\square$

### K.5.2    PROOF OF PROPOSITION K.6

To simplify the proof, we first introduce the *PSC with estimation policies*.

**Definition K.23.** *For $M \in \mathcal{M}$, let $\pi_M^{\mathrm{est}}$ be the uniform mixture of $\{\pi_{M,0}, \pi_{M,1}^{\mathrm{est}}, \cdots, \pi_{M,H}^{\mathrm{est}}\}$, where we define $\pi_{M,0} = \pi_M$. Let*

$$
\mathrm{psc}_\gamma^{\mathrm{est}}(\mathcal{M}, \overline{M}) := \sup_{\mu\in\Delta(\mathcal{M})} \mathbb{E}_{M\sim\mu}\mathbb{E}_{M'\sim\mu}\left[f^M(\pi_M) - f^{\overline{M}}(\pi_M) - \gamma D_{\mathrm{RL}}^2(\overline{M}(\pi_{M'}^{\mathrm{est}}), M(\pi_{M'}^{\mathrm{est}}))\right],
$$

*where we understand*

$$
D_{\mathrm{RL}}^2(\overline{M}(\pi_{M'}^{\mathrm{est}}), M(\pi_{M'}^{\mathrm{est}})) = \frac{1}{H+1}\sum_{h=0}^{H} D_{\mathrm{RL}}^2(\overline{M}(\pi_{M',h}^{\mathrm{est}}), M(\pi_{M',h}^{\mathrm{est}})).
$$

*We further define $\mathrm{psc}_\gamma^{\mathrm{est}}(\mathcal{M}) = \sup_{\overline{M}\in\mathcal{M}} \mathrm{psc}_\gamma^{\mathrm{est}}(\mathcal{M}, \overline{M})$.*

We can generalize Proposition E.3 to $\mathrm{psc}^{\mathrm{est}}$ by the same argument, as follows.

**Proposition K.24.** *When $\pi^{\mathrm{est}} = \pi$, it holds that*

$$
\overline{\mathrm{dec}}_\gamma(\mathcal{M}, \overline{M}) \leqslant \mathrm{psc}_{\gamma/6}^{\mathrm{est}}(\mathcal{M}) + \frac{2(H+1)}{\gamma}.
$$

*Generally, we always have*

$$
\overline{\mathrm{edec}}_\gamma(\mathcal{M}, \overline{M}) \leqslant \mathrm{psc}_{\gamma/6}^{\mathrm{est}}(\mathcal{M}) + \frac{2(H+1)}{\gamma}.
$$

Therefore, it remains to upper bound $\mathrm{psc}_\gamma^{\mathrm{est}}$ by $\mathrm{comp}(\mathcal{G}, \gamma)$.

**Proposition K.25.** *It holds that*

$$
\mathrm{psc}_\gamma^{\mathrm{est}}(\mathcal{M}, \overline{M}) \leqslant H \max_h \mathrm{dc}(\mathcal{G}_h^{\overline{M}}, \gamma/(H+1)L^2) + \frac{H+1}{4\gamma}.
$$

Combining Proposition K.25 with Proposition K.24 completes the proof of Proposition K.6.

**Proof of Proposition K.25**    Fix a $\overline{M} \in \mathcal{M}$ and $\mu \in \Delta(\mathcal{M})$. Then by the standard performance decomposition using the simulation lemma, we have

$$
\mathbb{E}_{M\sim\mu}\left[f^M(\pi_M) - f^{\overline{M}}(\pi_M)\right]
$$

$$= \mathbb{E}_{M \sim \mu} \left[ \mathbb{E}^M \left[ V_1^{M,\pi_M}(s_1) \right] - \mathbb{E}^{\overline{M}} \left[ V_1^{M,\pi_M}(s_1) \right] + \sum_{h=1}^{H} \mathbb{E}^{\overline{M},\pi_M} \left[ Q^{M,\pi_M}(s_h,a_h) - r_h - V_{h+1}^{M,\pi_M}(s_{h+1}) \right] \right]$$

$$\leqslant \mathbb{E}_{M \sim \mu} \left[ D_{\mathrm{TV}} \left( \mathbb{P}_0^M, \mathbb{P}_0^{\overline{M}} \right) \right] + \sum_{h=1}^{H} \mathbb{E}_{M \sim \mu} \left[ \left| g_h^{M;\overline{M}}(M) \right| \right].$$

Therefore, we bound

$$\sum_{h=1}^{H} \mathbb{E}_{M \sim \mu} \left[ \left| g_h^{M;\overline{M}}(M) \right| \right] = \sum_{h=1}^{H} \underbrace{\mathbb{E}_{M \sim \mu} \left[ \left| g_h^{M;\overline{M}}(M) \right| \right] - \eta \mathbb{E}_{M,M' \sim \mu} \left[ \left| g_h^{M;\overline{M}}(M') \right|^2 \right]}_{\leqslant \mathrm{dc}(\mathcal{G}_h^{\overline{M}},\eta)} + \eta \mathbb{E}_{M,M' \sim \mu} \left[ \left| g_h^{M;\overline{M}}(M') \right|^2 \right]$$

$$\leqslant \sum_{h=1}^{H} \mathrm{dc}(\mathcal{G}_h^{\overline{M}},\eta) + \eta L^2 \mathbb{E}_{M,M' \sim \mu} \left[ D_{\mathrm{RL}}^2 \left( \overline{M}(\pi_{M,h}^{\mathrm{est}}), M(\pi_{M,h}^{\mathrm{est}}) \right) \right],$$

where the inequality is due to the definition of Bellman representation and decoupling coefficient. Furthermore, we have

$$\mathbb{E}_{M \sim \mu} \left[ D_{\mathrm{TV}} \left( \mathbb{P}_0^M, \mathbb{P}_0^{\overline{M}} \right) \right] \leqslant \mathbb{E}_{M,M' \sim \mu} \left[ D_{\mathrm{H}}^2 \left( M(\pi_{M'}), \overline{M}(\pi_{M'}) \right) \right] \leqslant \mathbb{E}_{M,M' \sim \mu} \left[ D_{\mathrm{RL}} \left( M(\pi_{M'}), \overline{M}(\pi_{M'}) \right) \right]$$

$$\leqslant \frac{\gamma}{H+1} \mathbb{E}_{M,M' \sim \mu} \left[ D_{\mathrm{RL}}^2 \left( M(\pi_{M'}), \overline{M}(\pi_{M'}) \right) \right] + \frac{H+1}{4\gamma}.$$

Now taking $\eta = 1/L^2(H+1)$ and combining the above two inequalities above completes the proof. $\qquad\square$

### K.5.3 PROOF OF PROPOSITION K.7

Fix any set of models $\{M^k\}_{k \in [K]} \in \mathcal{M}$. By standard performance decomposition using the simulation lemma, we have

$$\frac{1}{K} \sum_{k=1}^{K} \left[ f^{M^k}(\pi_{M^k}) - f^{M^\star}(\pi_{M^k}) \right]$$

$$= \frac{1}{K} \sum_{k=1}^{K} \sum_{h=1}^{H} \mathbb{E}^{M^\star,\pi_{M^k}} \left[ Q^{M^k,\pi_{M^k}}(s_h,a_h) - r_h - V_{h+1}^{M^k,\pi_{M^k}}(s_{h+1}) \right]$$

$$\leqslant \frac{1}{K} \sum_{h=1}^{H} \sum_{k=1}^{K} \left| g_h^{M^k;M^\star}(M^k) \right|.$$

On the other hand, by the definition of Bellman representation,

$$\sum_{t=1}^{k-1} \left| g_h^{M^k;M^\star}(M^t) \right|^2 \leqslant L^2 \sum_{t=1}^{k-1} D_{\mathrm{RL}}^2(M^k(\pi_{M^t}), M^\star(\pi_{M^t})).$$

Therefore, defining

$$\widetilde{\beta} := \max_{k \in [K]} \sum_{t=1}^{k-1} D_{\mathrm{RL}}^2(M^k(\pi_{M^t}), M^\star(\pi_{M^t})),$$

we have $\sum_{t=1}^{k-1} \left| g_h^{M^k;M^\star}(M^t) \right|^2 \leqslant L^2 \widetilde{\beta}$ for all $k \in [K]$.

Our final step is to use an Eluder dimension argument. By the above precondition, we can apply the Eluder dimension bound in Jin et al. (2021, Lemma 41) to obtain that, for any $h \in [H]$ and $\Delta > 0$,

$$\frac{1}{K} \sum_{k=1}^{K} \left| g_h^{M^k;M^\star}(M^k) \right| \leqslant \frac{\mathcal{O}(1)}{K} \cdot \left( \sqrt{\mathfrak{e}(\mathcal{G}_h,\Delta) L^2 \widetilde{\beta} K} + \min \left\{ \mathfrak{e}(\mathcal{G}_h,\Delta), K \right\} + K \cdot \Delta \right)$$

$$\leqslant \frac{\mathcal{O}\left(1\right)L}{K} \cdot \left(\sqrt{\mathfrak{e}(\mathcal{G}_h, \Delta) \max\left\{\widetilde{\beta}, 1\right\} K} + K\Delta\right),$$

where $\mathcal{O}\left(1\right)$ hides the universal constant. Summing over $h \in [H]$ gives

$$\frac{1}{K} \sum_{h=1}^{H} \sum_{k=1}^{K} \left|g_h^{M^k;M^\star}(M^k)\right| \leqslant \frac{\mathcal{O}\left(1\right)L}{K}\left(H\sqrt{\max_{h\in[H]} \mathfrak{e}(\mathcal{G}_h, \Delta) \max\left\{\widetilde{\beta}, 1\right\}K} + KH\Delta\right)$$

$$\leqslant \frac{\gamma}{K} \cdot \max\left\{\widetilde{\beta}, 1\right\} + \frac{\mathcal{O}\left(1\right)}{K}\left(\frac{H^2L^2 \max_{h\in[H]} \mathfrak{e}(\mathcal{G}_h, \Delta)K}{\gamma} + KHL\Delta\right),$$

where the last inequality uses AM-GM. Therefore, we have shown that for any $\left\{M^k\right\}_{k\in[K]}$,

$$\frac{1}{K} \sum_{k=1}^{K} \left[f^{M^k}(\pi_{M^k}) - f^{M^\star}(\pi_{M^k})\right] - \frac{\gamma}{K} \cdot \max\left\{\widetilde{\beta}, 1\right\}$$

$$\leqslant \mathcal{O}\left(1\right) \cdot \left(\frac{H^2L^2 \max_{h\in[H]} \mathfrak{e}(\mathcal{G}_h, \Delta)}{\gamma} + HL\Delta\right)$$

$$\leqslant \mathcal{O}\left(1\right) \cdot H^2L^2\left(\frac{\max_{h\in[H]} \mathfrak{e}(\mathcal{G}_h, \Delta)}{\gamma} + \Delta\right).$$

By definition of $\mathrm{mlec}_{\gamma,K}$, taking $\inf_{\Delta>0}$ completes the proof. $\qquad\square$

### K.5.4 PROOF OF PROPOSITION K.9

By Proposition I.3 and the strong duality, we only need to upper bound the following quantity in terms of the decoupling coefficient:

$$\mathrm{rrec}_\gamma(\mathcal{M}, \overline{\mu}) = \sup_{\nu\in\Delta(\mathcal{P}\times\mathcal{R}\times\Pi)} \inf_{p_e\in\Delta(\Pi)} \mathbb{E}_{(\mathsf{P},R,\pi)\sim\nu} \left|\mathbb{E}_{\overline{\mathsf{P}}\sim\overline{\mu}}\left[f^{\overline{\mathsf{P}},R}(\pi') - f^{\mathsf{P},R}(\pi')\right]\right|$$

$$- \gamma\mathbb{E}_{\mathsf{P}\sim\nu}\mathbb{E}_{\pi\sim p_e}\mathbb{E}_{\overline{\mathsf{P}}\sim\overline{\mu}}\left[D_{\mathrm{H}}^2(\mathsf{P}(\pi), \overline{\mathsf{P}}(\pi))\right]$$

Then,

$$\mathbb{E}_{(\mathsf{P},R,\pi)\sim\nu}\mathbb{E}_{\overline{\mathsf{P}}\sim\mu}\left[\left|f^{\overline{\mathsf{P}},R}(\pi) - f^{\mathsf{P},R}(\pi)\right|\right]$$

$$\overset{(i)}{\leqslant} \mathbb{E}_{(\mathsf{P},R,\pi)\sim\nu}\mathbb{E}_{\overline{\mathsf{P}}\sim\mu}\left|\left(\mathbb{E}^{\mathsf{P}} - \mathbb{E}^{\overline{\mathsf{P}}}\right)\left[V_1^{(P,R),\pi}(s_1)\right]\right|$$

$$+ \sum_{h=1}^{H} \mathbb{E}_{(\mathsf{P},R,\pi)\sim\nu}\mathbb{E}_{\overline{\mathsf{P}}\sim\mu}\left|\mathbb{E}^{(\overline{\mathsf{P}},R),\pi}\left[Q_h^{(P,R),\pi}(s_h, a_h) - R_h - V_{h+1}^{(P,R),\pi}(s_{h+1})\right]\right|$$

$$\overset{(ii)}{\leqslant} \mathbb{E}_{(\mathsf{P},R,\pi)\sim\nu}\mathbb{E}_{\overline{\mathsf{P}}\sim\mu}\left[D_{\mathrm{TV}}\left(\mathsf{P}_0, \overline{\mathsf{P}}_0\right)\right] + \sum_{h=1}^{H} \mathbb{E}_{(\mathsf{P},R,\pi)\sim\nu}\mathbb{E}_{\overline{\mathsf{P}}\sim\mu}\left[\left|g_h^{(P,R);\overline{\mathsf{P}}}(\pi)\right|\right],$$

where (i) is because we use the performance decomposition lemma, (ii) is due to the definition of strong Bellman representation. Similar to the proof of Proposition K.25, we have

$$\sum_{h=1}^{H} \mathbb{E}_{(\mathsf{P},R,\pi)\sim\nu}\mathbb{E}_{\overline{\mathsf{P}}\sim\mu}\left[\left|g_h^{(P,R);\overline{\mathsf{P}}}(\pi)\right|\right] \leqslant \sum_{h=1}^{H} \left\{\mathrm{dc}(\mathcal{G}_h^{\overline{\mathsf{P}}}, \eta) + \eta\mathbb{E}_{(\mathsf{P},R)\sim\nu}\mathbb{E}_{\pi\sim\nu}\mathbb{E}_{\overline{\mathsf{P}}\sim\mu}\left[\left|g_h^{(P,R);\overline{\mathsf{P}}}(\pi)\right|^2\right]\right\}$$

$$\leqslant \sum_{h=1}^{H} \mathrm{dc}(\mathcal{G}_h^{\overline{\mathsf{P}}}, \eta) + L^2\eta \sum_{h=1}^{H} \mathbb{E}_{\mathsf{P}\sim\nu,\pi\sim\nu}\mathbb{E}_{\overline{\mathsf{P}}\sim\mu}\left[D_{\mathrm{H}}^2(\mathsf{P}(\pi_h^{\mathrm{est}}), \overline{\mathsf{P}}(\pi_h^{\mathrm{est}}))\right],$$

and

$$\mathbb{E}_{M\sim\mu}\left[D_{\mathrm{TV}}\left(\mathbb{P}_0^M, \mathbb{P}_0^{\overline{M}}\right)\right] \leqslant \frac{\gamma}{H+1}\mathbb{E}_{\mathsf{P}\sim\nu,\pi\sim\nu}\left[D_{\mathrm{H}}^2\left(\mathsf{P}(\pi), \mathsf{P}(\pi)\right)\right] + \frac{H+1}{4\gamma}.$$

We just need to choose $p_{e,h} \in \Delta(\Pi)$ as $p_{e,h}(\pi') = \nu(\{(\mathsf{P}, R, \pi) : \pi_h^{\mathrm{est}} = \pi'\})$ (here $\pi_0^{\mathrm{est}} = \pi$), $p_e = \frac{1}{H+1} \sum_{h=0}^{H} p_{e,h} \in \Delta(\Pi)$, and let $\eta = \frac{\gamma}{(H+1)L^2}$ to obtain

$$\mathrm{rrec}_\gamma(\mathcal{M}, \overline{\mu}) \leqslant \sum_{h=1}^{H} \mathrm{dc}(\mathcal{G}_h^{\overline{\mathsf{P}}}, \gamma/(H+1)L^2) + \frac{H+1}{4\gamma}$$

Applying Proposition I.3 completes the proof. $\qquad\square$

### K.5.5 PROOF OF PROPOSITION K.15

We construct such a covering directly, which is a generalization of the construction in Example K.13. An important observation is that, by the definition of feature map (cf. Definition K.14), it must hold that

$$\sum_{s'} \left\| \phi(s'|s,a) \right\|_1 \leqslant 2d, \qquad \forall (s,a) \in \mathcal{S} \times \mathcal{A}.$$

Then, we set $N = \lceil B/\rho \rceil$ and let $B' = N\rho$. For $\theta \in [-B', B']^d$, we define the $\rho$-neighborhood of $\theta$ as $\mathcal{B}(\theta, \rho) := \rho \lfloor \theta/\rho \rfloor + [0, \rho]^d$, and let

$$\widetilde{\mathbb{P}}_\theta(s'|s,a) := \max_{\theta' \in \mathcal{B}(\theta, \rho)} \left\langle \theta', \phi(s'|s,a) \right\rangle.$$

Then, if $\theta$ induces a transition dynamic $\mathbb{P}_\theta$, then $\widetilde{\mathbb{P}}_\theta \geqslant \mathbb{P}_\theta$, and

$$\sum_{s'} \left| \widetilde{\mathbb{P}}_\theta(s'|s,a) - \mathbb{P}_\theta(s'|s,a) \right| = \sum_{s'} \max_{\theta' \in \mathcal{B}(\theta, \rho)} \left| \left\langle \theta' - \theta, \phi(s'|s,a) \right\rangle \right| \leqslant \rho \sum_{s'} \left\| \phi(s'|s,a) \right\|_1 \leqslant 2\rho d.$$

Now, for $\Theta = (\theta_h)_h \in (\mathbb{R}^d)^{H-1}$, we define

$$\widetilde{\mathbb{P}}_\Theta^\pi(s_1, a_1, \cdots, s_H, a_H) := \mathbb{P}_1(s_1) \pi_1(a_1|s_1) \widetilde{\mathbb{P}}_{\theta_1}(s_2|s_1, a_1) \cdots \widetilde{\mathbb{P}}_{\theta_{H-1}}(s_H|s_{H-1}, a_{H-1}) \pi(a_H|s_H)$$

$$= \mathbb{P}_1(s_1) \cdot \prod_{h=1}^{H} \pi_h(a_h|s_h) \times \prod_{h=1}^{H-1} \widetilde{\mathbb{P}}_{\theta_h}(s_{h+1}|s_h, a_h).$$

Suppose that $\rho \leqslant 1/(2Hd)$, then a simple calculation shows that when $\Theta$ induces an MDP,

$$\left\| \widetilde{\mathbb{P}}_\Theta^\pi - \mathbb{P}_\Theta^\pi \right\|_1 \leqslant 2eHd\rho.$$

Therefore, let $\rho_1 = \sqrt{2eHd\rho}$, then by picking representative in each $\ell_\infty$-$\rho$-ball, we can construct a $\rho_1$-optimistic covering with $|\mathcal{M}_0| \leqslant (2N)^{Hd} = (2\lceil B/\rho \rceil)^{Hd} = \left( 2 \lceil 2eHdB/\rho_1^2 \rceil \right)^{Hd}$, which implies that $\log |\mathcal{M}_0| \leqslant \mathcal{O}\left( dH \log(dHB/\rho_1) \right)$. $\qquad \square$

### K.5.6 PROOF OF PROPOSITION K.18

We deal with linear mixture MDP first. Suppose that for each $\mathsf{P} \in \mathcal{P}$, $\mathsf{P}$ is parameterized by $\theta^\mathsf{P} = (\theta_h^\mathsf{P})_h$. Then the QER of $\mathcal{P}$ is given by

$$g_h^{\mathsf{P};\overline{\mathsf{P}}}(\pi) = \mathbb{E}^{\overline{\mathsf{P}},\pi} \left[ D_{\mathrm{TV}} \left( \mathbb{P}_h^\mathsf{P}(\cdot|s_h, a_h), \mathbb{P}_h^{\overline{\mathsf{P}}}(\cdot|s_h, a_h) \right) \right]$$

$$= \frac{1}{2} \max_{V: \mathcal{S} \times \mathcal{S} \times \mathcal{A} \to [-1,1]} \mathbb{E}^{\overline{\mathsf{P}},\pi} \left[ \sum_{s_{h+1}} \left( \mathbb{P}_h^\mathsf{P}(s_{h+1}|s_h, a_h) - \mathbb{P}_h^{\overline{\mathsf{P}}}(s_{h+1}|s_h, a_h) \right) V(s_{h+1}|s_h, a_h) \right]$$

$$= \frac{1}{2} \max_{V: \mathcal{S} \times \mathcal{S} \times \mathcal{A} \to [-1,1]} \mathbb{E}^{\overline{\mathsf{P}},\pi} \left[ \sum_{s_{h+1}} \left\langle \theta_h^\mathsf{P} - \theta_h^{\overline{\mathsf{P}}}, \phi_h(s_{h+1}|s_h, a_h) \right\rangle V(s_{h+1}|s_h, a_h) \right]$$

$$= \frac{1}{2} \max_{V: \mathcal{S} \times \mathcal{S} \times \mathcal{A} \to [-1,1]} \left\langle \theta_h^\mathsf{P} - \theta_h^{\overline{\mathsf{P}}}, \mathbb{E}^{\overline{\mathsf{P}},\pi} \left[ \sum_{s_{h+1}} V(s_{h+1}|s_h, a_h) \phi_h(s_{h+1}|s_h, a_h) \right] \right\rangle$$

$$= \frac{1}{2} \max_{V: \mathcal{S} \times \mathcal{S} \times \mathcal{A} \to [-1,1]} \left| \left\langle \theta_h^\mathsf{P} - \theta_h^{\overline{\mathsf{P}}}, \mathbb{E}^{\overline{\mathsf{P}},\pi} \left[ \sum_{s_{h+1}} V(s_{h+1}|s_h, a_h) \phi_h(s_{h+1}|s_h, a_h) \right] \right\rangle \right|.$$

Applying Proposition K.9 and Corollary K.21 completes the proof of this case.

We next deal with the class $\mathcal{P}$ of dynamic of MDPs with occupancy rank at most $d$. For $\mathsf{P} \in \mathcal{P}$, we consider $\phi^\mathsf{P} = \left( \phi_h^\mathsf{P} : \Pi \to \mathbb{R}^d \right)$, $\psi^\mathsf{P} = \left( \psi_h^\mathsf{P} : \mathcal{S} \to \mathbb{R}^d \right)$, such that $\mathbb{P}^{\mathsf{P},\pi}(s_h = s) = \left\langle \psi_h^\mathsf{P}(s), \phi_h^\mathsf{P}(\pi) \right\rangle$. Then the VER of $\mathcal{P}$ is given by

$$g_h^{\mathsf{P};\overline{\mathsf{P}}}(\pi) = \mathbb{E}^{\overline{\mathsf{P}},\pi} \left[ \max_{a \in \mathcal{A}} D_{\mathrm{TV}} \left( \mathbb{P}_h^\mathsf{P}(\cdot|s_h, a), \mathbb{P}_h^{\overline{\mathsf{P}}}(\cdot|s_h, a) \right) \right]$$

$$= \left\langle \sum_s \psi_h^{\overline{\mathsf{P}}}(s) \max_{a \in \mathcal{A}} D_{\mathrm{TV}} \left( \mathbb{P}_h^\mathsf{P}(\cdot|s, a), \mathbb{P}_h^{\overline{\mathsf{P}}}(\cdot|s, a) \right), \phi_h^{\overline{\mathsf{P}}}(\pi) \right\rangle.$$

Applying Proposition K.9 and Example K.2 completes the proof of this case. $\qquad \square$

