# OpenReview forum: "Unified Algorithms for RL with Decision-Estimation Coefficients: No-Regret, PAC, and Reward-Free Learning"
_ICLR.cc/2023/Conference — Submitted to ICLR 2023_

### Official Review · Reviewer_tzo7 · 2022-10-20

**Confidence:** 4
**Clarity, Quality, Novelty And Reproducibility:** Proof is nice and structured. Novelty…
**Correctness:** 3
**Technical Novelty And Significance:** 2
**Empirical Novelty And Significance:** Not applicable
**Recommendation:** 5

**Strength And Weaknesses:**

I appreciate the authors' great effort to extend DEC to PAC setting and reward-free setting with a lot of tedious calculation. However, I feel this work is a bit incremental. Instead of covering material as much as possible, I suggest the author focuses on one problem and thinks deeper. There are some important problems left for DEC and this paper inherits them to other settings.

1. It's better to explain clearly what do you mean "sufficient and necessary"? Clearly the upper bound (Thm 5) and lower bound (Proposition 6) differ by a log(|M|) term.

2. The algorithm is far from computational efficient. There are several minimax operators involved such that there is little useful guidance for practitioners. The algorithm is not implementable and of course no experiments.

3. The guarantee in Example 13 is far from optimal. The dependency for S is very bad. It is unclear this is due to analysis or due to the complexity measure. Again, then why you can claim "sufficient and necessary"? "Unified algorithm" should not be an excuse. That means this unified algorithm cannot pass the sanity check of tabular MDPs.

4. It's very unusual for ICLR to have a 58 page submission. That means it is impossible for reviewers to check the correctness of the proof. And a lot of proofs are repeated from Foster et al. (2021) (for example, Section B.3). Such a paper is more suitable for journal submission in my mind.


**Summary Of The Paper:**

This paper extended DEC from regret minimization to PAC guarantee and reward-free case. Similar algorithms, lower bounds and upper bounds are established.

**Summary Of The Review:**

A solid theoretical paper but a bit incremental. Computational efficiency is a big limitation.

---

> ### Author Response · Authors · 2022-11-09
> **Response to Reviewer tzo7**
>
> We thank the reviewer for the valuable feedback to our paper. We respond to the main concerns as follows.
>
> *--- “It's better to explain clearly what do you mean "sufficient and necessary"? Clearly the upper bound (Theorem 5) and lower bound (Proposition 6) differ by a $\log|\mathcal{M}|$ term.”*
>
> By “necessary and sufficient”, we indeed meant modulo the $\log|\mathcal{M}|$ difference. This claim follows Foster et al. (2021) but we agree that this could be imprecise. To avoid confusions, we have properly edited all appearances of “necessary and sufficient” in our revision by either changing the statement or adding additional explanations (for example the edits in the abstract and Section 1 marked in red).
>
> *--- “The algorithm is far from computationally efficient. There are several minimax operators involved, such that there is little useful guidance for practitioners.”*
>
> We agree that our main algorithm E2D-TA is not computationally efficient and minimizes a somewhat involved risk function with several min-max operators. Our alternative algorithms Model-Based Posterior Sampling (MOPS; Algorithm 5) and Optimistic Maximum Likelihood Estimation (OMLE; Algorithm 6) make small steps towards computational efficiency—They are *slightly* more amenable to approximate practical implementation, with simpler risk functions/update rules that do not involve min-max operators. And they enjoy similar regret guarantees as the main algorithm E2D-TA (Appendix E).
>
> However, we believe the question of computational efficiency is beyond the scope of this work, given that our algorithms achieve sample-efficiency with general function approximation (with general model classes), where computational efficiency is in general a challenging and largely unsolved open problem (see, e.g. Du et al. 2021, Jin et al. 2021, Foster et al. 2021). We agree though that designing fully computationally efficient algorithms is an important question for the entire line of work on sample-efficient RL.
>
>
> *--- “The dependency for $S$ is very bad [...] this unified algorithm cannot pass the sanity check of tabular MDPs.”*
>
> We would first like to emphasize that achieving optimal dependence on $S$ for tabular MDPs is not the focus of this work, as our algorithms are aimed to achieve sample efficiency for RL with general function approximation (in the form of an arbitrary model class), where this kind of suboptimal behavior (on a specific subclass of problems) for algorithms that work for general function approximation is common and mostly expected. For example, minimax optimal algorithms for linear mixture MDPs result in suboptimal $S,A$ dependencies when specialized to the subclass of tabular MDPs (e.g. Remark 5.3, Zhou et al. 2021).
>
> We agree with the point though that our algorithm does not achieve optimal dependence on $S$ for tabular MDPs. Sharpening our algorithm (or any other general algorithms alike) on tabular MDPs, while maintaining their bounds on broader classes is an important open question, which we’d like to leave as future work.
>
> [Reference] Zhou, Dongruo, Jiafan He, and Quanquan Gu. "Provably efficient reinforcement learning for discounted mdps with feature mapping." In International Conference on Machine Learning, pp. 12793-12802. PMLR, 2021.
>
> ---
> We thank the reviewer again for reading our response. We would sincerely appreciate it if the reviewer could let us know of any additional issues, or consider improving the rating if we have addressed the concerns.

---

> > ### Author Response · Authors · 2022-12-09
> > **Following Up**
> >
> > Dear reviewer,
> >
> > We are following up to check whether our rebuttal has addressed your concerns. Please let us know if you have any further questions. If your concern is addressed, we would appreciate if you would reconsider your score in light of our clarification.
> >
> > Thank you,
> >
> > Authors

---

### Official Review · Reviewer_Tm8H · 2022-10-26

**Confidence:** 3
**Correctness:** 3
**Technical Novelty And Significance:** 2
**Empirical Novelty And Significance:** Not applicable
**Recommendation:** 5

**Clarity, Quality, Novelty And Reproducibility:**

The presentation should be improved. In particular, the comprison with Foster et al. 2021 is not sufficient. Since E2D-TA is very similar to E2D, the authors may also present E2D to show the difference. There is no numerical experiments.

**Strength And Weaknesses:**

Strength: The authors proposed E2D-TA by combining E2D with tempered aggregation to help avoid estimating the transition model.
In technique, the authors propose a new definition for $dec$ and manage to derive a stronger bound for error term $Est_{RL}$.


Weaknesses: The motivation is not strong enough. The authors compare this work with Foster et al., 2021 in the third paragraph in Section 1.  As stated above, the main merit of the proposed algorithm is to avoid estimating the transition model. However, the proposed algorithm is still computational intractable given this improvement.  Even assuming the tabular state-action space (where we can estimate the model efficiently), E2D-TA seems still computational inefficient.



**Summary Of The Paper:**

The paper follows the framework of decision-estimation coefficient (DEC) proposed by (Foster et al. 2021). The major contribution of this work is to combine DEC with tempered aggregation to design algorithms for several problems (e.g., regret minimization, PAC-learning and reward-free learning) in DMSO (decision making under structured observation).

**Summary Of The Review:**

Given the consideration above, I think the motivation is still not clear and tend to reject this paper. It would be helpful if the authors could give some meaningful examples where E2D-TA is computational efficient while E2D is not.

---

> ### Author Response · Authors · 2022-11-09
> **Response to Reviewer Tm8H**
>
> We thank the reviewer for the valuable feedback to our paper. We respond to the main concerns as follows.
>
> *--- “The motivation is not strong enough. [...] the main merit of the proposed algorithm is to avoid estimating the transition model.”*
>
> Our main algorithm E2D-TA (and the PAC/reward-free algorithms) does not avoid estimating the transition model. Instead, the TA (Tempered Aggregation) subroutine achieves a stronger model estimation guarantee than Vovk’s aggregating algorithm in Foster et al. (2021), which in turn allows us to give the first truly unified E2D style algorithm for sample-efficient model-based RL.
>
> *--- “the comparison with Foster et al. 2021 is not sufficient.”*
>
> We believe the improvement of our E2D-TA algorithm over Foster et al. (2021) is a key contribution of this work. Please find a more detailed discussion about this point in our “Response to all reviewers”.
>
> *--- “E2D-TA seems still computational inefficient.”*
>
> We agree that E2D-TA are computationally inefficient, and we believe computational efficiency is indeed an important question for the entire line of work on sample-efficient RL. However, we believe this is beyond the scope of this work, given that our algorithms are aimed for sample-efficiency with general function approximation (with general model classes), where computational efficiency is in general a challenging and largely unsolved open problem (see, e.g. Du et al. 2021, Jin et al. 2021, Foster et al. 2021).
>
>
> ---
> We thank the reviewer again for reading our response. We would sincerely appreciate it if the reviewer could let us know of any additional issues, or consider improving the rating if we have addressed the concerns.

---

> > ### Author Response · Authors · 2022-12-09
> > **Following Up**
> >
> > Dear reviewer,
> >
> > We are following up to check whether our rebuttal has addressed your concerns. Please let us know if you have any further questions. If your concern is addressed, we would appreciate if you would reconsider your score in light of our clarification.
> >
> > Thank you,
> >
> > Authors

---

### Official Review · Reviewer_oWcL · 2022-11-03

**Confidence:** 3
**Correctness:** 3
**Technical Novelty And Significance:** 2
**Empirical Novelty And Significance:** Not applicable
**Recommendation:** 5

**Clarity, Quality, Novelty And Reproducibility:**

The paper is written in good language and the notations are easy to follow. Since this is a theoretical machine learning paper there is no experiment and reproducibility is not applicable.

**Strength And Weaknesses:**

The strengths of the paper according to me are the following:
1. The authors prove the upper bound of DEC, EDEC and RFDEC under a uniform framework, which is not covered by previous works.
2. The authors design an algorithm for reward-free and PAC learning case, with finite class and low-bellman dimension models as examples. This is not covered in previous works under E2D framework.

The weaknesses of the paper (according to me) are the following:
1. In my humble optinion, it is hard to justify the paper's novelty. For me, the main algorithm E2D-TA seems to be a modification of Algorithm 1 in (Foster et al.,2021) only by changing the method of learning the underlying model.  Foster et al. first estimate the underlying model using methods such as Vovk aggregation which only incurs an extra estimation error of \gamma\log(|M|/\delta). This paper estimates the underlying model by computing posterior distribution on the model set and claim to be a tighter bound justified by Jensen's inequality, but its contribution to the overall regret is still of scale \gamma\log(|M|/\delta) and therefore this technique, as the main novelty, doesn't seem to optimize the regret bound. Meanwhile, this paper adopts different definition of estimation error i.e. EST. Therefore the authors need to further justify the importance of using this method.

2. Forster et al. provide an analysis for infinite model class case in Section 3.2.3 with a proof base on covering number and further show its application in bilinear RL settings and covers low BE dimension results in Appendix F.   It is hard to see the main difference between the analyses in those two paper. The results in reward-free cases and PAC learning under those settings also seem to be a straightforward generalization.

3. The necessity of using TA as the model estimator is not obvious. In my humble opinion, it would be better if the authors can justify why simply using other estimators (e.g. Vovk's aggregation) can not be generalized to reward-free/PAC cases.



**Summary Of The Paper:**

The authors introduce a new model-based meta-algorithm called E2D-TA under the framework of E2D(Forster et al.,2021), which updates the model estimator by Tempered Aggregation. Under this main algorithm, the authors further introduced two complexity parameters, EDEC and RFDEC, and correspondingly designed two algorithms for PAC and reward-free learning tasks. The authors also provide the rate of growth for E2D, EDEC and RFDEC, and prove that with delicate choice of the parameters, these algorithms enjoy a sublinear regret bound, which also meets the lower bound. Moreover, the authors provide sublinear regret bounds for RL with Bellman representability under E2DTA framework.

**Summary Of The Review:**

This paper generalizes the E2D framework in Foster et al. to PAC learning and reward-free learning by substituting the model estimation method with Tempered Aggregation and new complexity metrics. It further provides minimax optimal regret bound, which is also proved to be sublinear under many cases, including finite model class and low BE dimension cases. However, there are still difficulties to grasp its true novelty. In my opinion, it would be better if the authors can further highlight the contribution of their techniques in more concrete ways, especially:
1. The generalization from simple E2D to their reward-free and PAC learning edition, and
2. The connection between this generalization and using TA as the model estimator.

---

> ### Author Response · Authors · 2022-11-09
> **Response to Reviewer oWcL**
>
> We thank the reviewer for the valuable feedback to our paper. We respond to the main concerns as follows.
>
> *--- “it is hard to justify the paper's novelty [...] Vovk aggregation which only incurs an extra estimation error of \gamma\log(|M|/\delta) [...] but its contribution to the overall regret is still of scale \gamma\log(|M|/\delta) and therefore this technique, as the main novelty, doesn't seem to optimize the regret bound”*
>
> We believe this question is about a key contribution of this work. Please find a detailed comparison between our E2D-TA and Foster et al. (2021)’s result in our “Response to all reviewers”.
>
> As a short answer: We agree that, for the “model estimation term” within the regret bounds, our new model estimation subroutine (Tempered Aggregation; TA) achieves appearingly the same $\log(|\mathcal{M}|/\delta)$ bound as Vovk’s aggregation (though with different meanings, as we explained in the “Proof overview” in Section 3). However, TA gives an important improvement on the “DEC term”—The regret bound of E2D with Vovk’s aggregation scales with a generally intractable version of the DEC that does not admit polynomial bounds for most RL problems. By contrast, the regret bound of E2D-TA scales with a much milder DEC that admits polynomial bounds for most known tractable RL problems, which makes E2D-TA the first truly unified sample-efficient E2D algorithm for RL.
>
> We have also revised our writings in Section 3 & Introduction/Abstract to highlight this comparison.
>
> *--- “The results in reward-free cases and PAC learning under those settings also seem to be a straightforward generalization.”*
>
> We agree that our DEC definitions (EDEC/RFDEC) in these settings are similar to the DEC for the no-regret setting. However, we emphasize that the resulting algorithms (Explorative / Reward-Free E2D) are new and conceptually very different from existing PAC/reward-free algorithms, which makes them interesting contributions in our opinion.
>
> For example, Explorative E2D automatically *learns* good exploration policies instead of manually setting them as in most existing PAC algorithms. Reward-Free E2D is different from existing reward-free algorithms using optimistic model estimation (Liu et al. 2021) or reaching all relevant states (Jin et al. 2020); Instead, it directly chooses an exploration policy that minimizes a RFDEC style min-max risk. In addition, we handle all three settings (no-regret, PAC, reward-free) with similar E2D algorithms that only differ in the risk functions, which is the first such unification to our best knowledge.
>
> [Reference]
> Chi Jin, Akshay Krishnamurthy, Max Simchowitz, and Tiancheng Yu. Reward-free exploration for reinforcement learning. In International Conference on Machine Learning, pp. 4870–4879. PMLR, 2020a.
> Qinghua Liu, Tiancheng Yu, Yu Bai, and Chi Jin. A sharp analysis of model-based reinforce-
> ment learning with self-play. In International Conference on Machine Learning, pp. 7001–7010.
> PMLR, 2021.
>
> *--- “The necessity of using TA as the model estimator is not obvious [...] why simply using other estimators (e.g. Vovk's aggregation) can not be generalized to reward-free/PAC cases.”*
>
> As we argued in the first point, TA was indeed necessary to fix the issue about Vovk’s aggregating algorithm and yield polynomial regret bounds. This is also the case for the generalizations to PAC/reward-free settings, where TA was necessary in order to obtain unified polynomial sample complexity guarantees.
>
> ---
> We thank the reviewer again for reading our response. We would sincerely appreciate it if the reviewer could let us know of any additional issues, or consider improving the rating if we have addressed the concerns.

---

> > ### Author Response · Authors · 2022-12-09
> > **Following Up**
> >
> > Dear reviewer,
> >
> > We are following up to check whether our rebuttal has addressed your concerns. Please let us know if you have any further questions. If your concern is addressed, we would appreciate if you would reconsider your score in light of our clarification.
> >
> > Thank you,
> >
> > Authors

---

### Author Response · Authors · 2022-11-09
**Response to all reviewers on core improvement over Foster et al. (2021) & Revision Uploaded**

We thank all reviewers again for their valuable feedback to our work.

We would like to humbly point out that, our core improvement over Foster et al. (2021), the regret bound of the E2D-TA algorithm (Section 3), seems to be misunderstood by the reviewers. We would like to clarify this point, as we believe this is a key contribution of our work. We are happy to engage / discuss further if the reviewers have additional questions about this point.

The main improvement of E2D-TA over E2D with Vovk’s aggregating algorithm (henceforth *E2D-Vovk*) of Foster et al. (2021) is an improved regret upper bound that scales with **a much milder version of the DEC**, in comparison with E2D-Vovk which scales with a **potentially prohibitive variant of the DEC** that do not admit polynomial bounds for most known tractable RL problems. The mild version of the DEC is necessary for the regret bound to be turned to desired polynomial sample complexities for large classes of tractable RL problems.

More concretely, the main upper bound of Foster et al. (2021, Theorem 3.3 & 4.1) shows that E2D-Vovk achieves regret
$$
\mathcal{O}( T \times \sup_{\overline{M}\in{\color{red} {\rm co}(\mathcal{M})}} {{\rm dec}_\gamma}(\mathcal{M}, \delta^{\overline{M}}) + \gamma \log(|\mathcal{M}|/\delta) ),
$$

where ${\color{red} {\rm co}(\mathcal{M})}$ denotes the set of all *mixtures* of models in $\mathcal{M}$, and $\delta^{\overline{M}}$ denotes the point mass at ${\overline{M}}$. By contrast, our E2D-TA (our Theorem 2) achieves regret
$$
\mathcal{O}( T \times \overline{\rm dec}_\gamma(\mathcal{M}) + \gamma \log(|\mathcal{M}|/\delta) ).
$$

For most tractable RL problems, the quantity $\sup_{\overline{M}\in{\color{red} {\rm co}(\mathcal{M})}} {\rm dec}_\gamma(\mathcal{M}, \overline{M})$ does not admit polynomial bounds (except for bandit problems), as noted in Foster et al. (2021, Section 7.1.3). The intuition is that $\overline{M}\in{\rm co}(\mathcal{M})$ may in general be a *mixture* of MDPs, and hence no longer admits a Markov structure. To address that issue, Foster et al. developed alternative algorithms with better regret bounds, but those require problem-specific model estimation subroutines (cf. their Section 7.1.3) which they can only design for restricted problem classes such as tabular MDPs (or any problem satisfying their additional "layer-wise convex" assumption). By contrast, $\overline{\rm dec}_\gamma(\mathcal{M})$ admits polynomial bounds for most known tractable RL problems, which implies desired sample complexity guarantees (cf. our Section 5).

In conclusion, the change from Vovk’s aggregating algorithm to TA was a **necessary** change to make E2D a truly unified algorithm for RL problems with desired sample complexity guarantees. We believe this makes it an important contribution to both the DEC framework, and the theory of sample-efficient RL at large.

We understand the reviewers’ concerns about the similarity (and thus seemingly incremental contribution) between TA and Vovk’s aggregating algorithm, as well as the two versions of the DEC. We have revised our writings (Section 3 & Appendix C.1) and the abstract/introduction to further clarify this, with changes marked in red.

---

> ### Comment · Reviewer_tzo7 · 2022-11-11
> **Thanks for the response.**
>
> Can you clarify what do you mean "they can only design for certain restricted problem classes such as tabular MDPs"? I believe Dylan's paper can cover bilinear classes which is a general class of models.

---

> > ### Author Response · Authors · 2022-11-11
> > **Response**
> >
> > Thanks for pointing this out. By “certain restricted problem classes such as tabular MDPs”, we meant the layer-wise convex transition assumption (Assumption 7.2) in Dylan’s paper, and we were thinking about tabular MDPs as a primary example of this assumption. We have edited the response with some additional explanations accordingly.
> >
> > Regarding bilinear classes, their work does contain two sets of results for bilinear classes, but those require either additional assumptions or special algorithms:
> >
> > * “Basic results” (their Section 7.1) require bilinear class + the layer-wise convex assumption. Importantly, layer-wise convex assumption is not implied by bilinear class, and many important problems within bilinear class are not layer-wise convex in general, such as low-rank MDPs.
> >
> > * “Refined guarantees” (their Section 7.2) are obtained by a modified algorithm whose risk function is tailored to the Bilinear class structure, and hence is no longer a generic RL algorithm. Further, that algorithm only achieves low Bayes regret instead of frequentist regret.
> >
> > Overall, we believe their guarantees in the “Basic results” part could work in general under low DEC + the layer-wise convex assumption, the latter being quite an uncommon assumption for RL with general function approximation. Our E2D-TA improves over this as it works under just low DEC.

---

### Decision · Program_Chairs · 2023-01-20

**Decision:**

Reject

**Justification For Why Not Higher Score:**

No new tractable MDP instances.

The presentation is distracting.

**Justification For Why Not Lower Score:**

N/A

**Metareview: Summary, Strengths And Weaknesses:**

This paper follows up along the DEC framework proposed recently by [Foster et al. 2021], and proposed two similar complexity metrics Explorative DEC (EDEC) and Reward-Free DEC (RFDEC), for the PAC learning and reward-free exploration tasks, respectively. However, such extensions are considered by all reviewers to be incremental, and indeed much of the proofs are directly borrowed from the original DEC paper.

Yet, it is emphasized by the authors in their response that even in the regret-minimization setting, the proposed algorithm achieves tighter regret bounds than the original DEC paper. In fact, it is the first algorithm that simultaneously achieves polynomial regret bound for all known tractable instances, whereas the original E2D algorithm fails without task-specific modifications (thus not a unified algorithm anymore).

In my opinion, if this were the first DEC paper, I would vote for acceptance with no hesitation. However, the situation is tricky here because the original DEC paper has yet to be published, and the contribution of this paper on top of the original manuscript is minor. In particular, there are two main drawbacks:
1. The extension to PAC learning and reward-free exploration are very incremental and technically not challenging. Such results, in fact, serve as a distraction and make readers/reviewers feel the paper is trivial.
2. While this paper fixes some issues suffered by the original DEC paper, e.g., the original DEC measure is exponential for some tractable RL instances., it does not include any new RL instances that are previously unsolved, and this has been one of the most important metrics to judge the contribution of an RL theory paper which claims to provide a more general solution. The "unified algorithm" seems minor to me, because I can always design a "model-selection" type meta-algorithm, say by combining bilinear-UCB and GOLF. Then it will technically be a "unified algorithm" that solves both Bilinear Class and low Bellman-Eluder dimension.

I want to suggest the following changes to the authors:
1. Downweight the emphasis on PAC and reward-free learning. Maybe move them completely to the appendix.
2. In the main paper, focus on the regret-minimization setting and elaborate precisely on what is lacking in the original DEC paper and how you fix it.
3. Find new MDP instances that are solvable with the new algorithm but are unsolvable under any previous framework, e.g, bilinear class and bellman-eluder dimension.